# The Seasonal Cycle of $pCO_2$ and $CO_2$ fluxes in the Southern Ocean: Diagnosing Anomalies in CMIP5 Earth System Models

**Precious N. Mongwe [1,2]., Marcello Vichi[2,3] & Pedro M.S. Monteiro[1,2]**

**[1]Southern Ocean Carbon-Climate Observatory (SOCCO), CSIR, Cape Town, South Africa**

**[2]Department of Oceanography, University of Cape Town, Cape Town, South Africa**

**[3]Marine Research Institute, University of Cape Town, Cape Town, South Africa**

pmongwe@csir.co.za

## Abstract

The Southern Ocean forms an important component of the earth system as a major sink of $CO_2$ and heat. Recent studies based on the Coupled Model Intercomparison Project version 5 (CMIP5) Earth System Models (ESMs) show that CMIP5 models disagree on the phasing of the seasonal cycle of the $CO_2$ flux ($FCO_2$) and compare poorly with available observation products for the Southern Ocean. Because the seasonal cycle is the dominant mode of $CO_2$ variability in the Southern Ocean, its simulation is a rigorous test for models and their long-term projections. Here we examine the competing roles of temperature and dissolved inorganic carbon (DIC) as drivers of the seasonal cycle of $pCO_2$ in the Southern Ocean to explain the mechanistic basis for the seasonal biases in CMIP5 models. We find that despite significant differences in the spatial characteristics of the mean annual fluxes, the intra-model homogeneity in the seasonal cycle of $FCO_2$ is greater than observational products. $FCO_2$ biases in CMIP5 models can be grouped into two main categories i.e. group-SST and group-DIC. Group-SST models show an exaggeration of the seasonal rates of change of sea surface temperature (SST) in autumn and spring during the cooling and warming peaks. These higher-than-observed rates of change of SST tip the control of the seasonal cycle of $pCO_2$ and $FCO_2$ towards SST and result in a divergence between the observed and modelled seasonal cycles, particularly in the Sub-Antarctic Zone. While almost all analyzed models (9 out of 10) show these SST-driven biases, 3 out of 10 (namely NorESM1-ME, HadGEM-ES and MPI-ESM, collectively the group-DIC models) compensate the solubility bias because of their overly exaggerated primary production, such that biologically-driven DIC changes mainly regulate the seasonal cycle of $FCO_2$.

## 1. Introduction

The Southern Ocean (south of $30^oS$) takes up about a third of the total oceanic $CO_2$ uptake, slowing down the accumulation of $CO_2$ in the atmosphere (Fung et al., 2005; Le Quere et al., 2016; Takahashi et al., 2012). The combination of upwelling deep ocean circumpolar waters (which are rich in carbon and nutrients) and the subduction of fresh colder mid-latitude waters makes it a key region in the role of sea-air gas exchange and heat uptake(Barbero et al., 2011; Gruber et al., 2009; Sallée et al., 2013). The Southern Ocean supplies about a third of the total nutrients responsible for biological production north of $30^oS$ (Sarmiento et al., 2004), and accounts for about 75% of total ocean heat uptake (Frölicher et al., 2015). Recent studies suggest that the Southern Ocean $CO_2$ sink is expected to change as a result of anthropogenic warming, however, the sign and magnitude of the change is still disputed (Leung et al., 2015; Roy et al., 2011; Sarmiento et al., 1998; Segschneider and Bendtsen, 2013). While some studies suggest that the Southern Ocean $CO_2$ sink is weakening and will continue to do so (e.g. Le Quéré et al., 2007; Son et al, 2010; Thompson et al., 2011), other recent studies infer an increasing $CO_2$ sink (Landschutzer et al., 2015; Takahashi et al., 2012; Zickfeld et al., 2008).

Although the Southern Ocean plays a crucial role as a $CO_2$ reservoir and regulator of nutrients and heat, it remains under-sampled, especially during the winter season (JJA) (seasonal cycle in the Southern Hemisphere) (Bakker et al., 2014; Monteiro et al., 2010). Consequently, we largely rely on Earth System Models (ESM), inversions and ocean models for both process understanding and future simulation of $CO_2$ processes in the Southern Ocean. The Coupled Model Intercomparison Project (CMIP) provides an example of such a globally organized platform (Taylor et al., 2012). Although recent studies based on CMIP5 ESMs, forward and inversions models show that CMIP5 models agree on the $CO_2$ annual mean sink, they disagree with available observations on the phasing of the seasonal cycle of sea-air $CO_2$ flux (FCO$_2$) in the Southern Ocean (e.g. Anav et al., 2013; Lenton et al., 2013).

The seasonal cycle is a major mode of variability for chlorophyll (Thomalla et al., 2011) and $CO_2$ in the Southern Ocean (Monteiro et al., 2010; Lenton et al., 2013). The large-scale seasonal states of sea-air $CO_2$ fluxes (FCO$_2$) in the Southern Ocean comprise of extremes of strong summer in-gassing with a weaker in-gassing or even out-gassing in winter (Metzl et al., 2006). These extremes are linked by the autumn and spring transitions. In autumn $CO_2$ in-gassing weakens linked to the increasing entrainment of sub-surface waters, which are rich in dissolved inorganic carbon (DIC), (Lenton et al., 2013; Metzl et al., 2006; Sarmiento and Gruber, 2006). During spring, the increase of primary production consumes DIC at the surface and increases the ocean's capacity to take up atmospheric $CO_2$ (Gruber et al., 2009; Le Quéré and Saltzman, 2013; Pasquer et al., 2015; Gregor et al., 2017). The increase of sea surface temperature (SST) in summer reduces surface $CO_2$ solubility, which counteracts

the biological uptake and reduces the $CO_2$ flux from the atmosphere (Takahashi et al., 2002; Lenton et
al., 2013).

$FCO_2$ is also spatially variable in the Southern Ocean at the seasonal scale. North of 50°S is generally
the main $CO_2$ uptake zone (Hauck et al., 2015; Sabine et al., 2004). This region forms a major part of
the sub-Antarctic Zone and is characterized by the confluence of upwelled, colder and nutrient-rich
deep circumpolar water and mid-latitudes warm water (McNeil et al., 2007; Sallée et al., 2006). It is
characterized by enhanced biological uptake during spring and solubility-driven $CO_2$ uptake due to
cool surface waters (Marinov et al., 2006; Metzl, 2009; Takahashi et al., 2012). South of 60°S towards
the marginal ice Zone, $CO_2$ fluxes are largely dominated by out-gassing, driven by the upwelling of
circumpolar waters, which are rich in DIC (Matear and Lenton, 2008; McNeil et al., 2007).

The inability of CMIP5 ESM to simulate a comparable $FCO_2$ seasonal cycle with available observations
estimates in the Southern Ocean has been the subject of recent literature (e.g. Anav et al., 2013;
Kessler and Tjiputra, 2016) and the mechanisms associated with these biases are still not well
understood. This model-observations disagreement highlights that the current ESMs might not
adequately capture the dominant seasonal processes driving the $FCO_2$ in the Southern Ocean. It also
questions the sensitivity of models to adequately simulate the Southern Ocean century-scale $CO_2$ sink
and its sensitivity to climate change feedbacks (Lenton et al., 2013). Efforts to improve simulations of
$CO_2$ properties with respect to observations in the Southern Ocean are ongoing using forced ocean
models (e.g. Pasquer et al., 2015; Rodgers et al., 2014; Visinelli et al., 2016; Rosso et al., 2017).
However, it remains a challenge for fully coupled simulations. In a previous study, we developed a
diagnostic framework to evaluate the seasonal characteristics of the drivers of $FCO_2$ in ocean
biogeochemical models (Mongwe et al., 2016). We here apply this approach to 10 CMIP5 models
against observation product estimates in the Southern Ocean. The subsequent analysis is divided as
follows; the methods section (section 2) explains our methodological approach, followed by results
(section 3), which comprise four subjections. Section 3.1 explores the spatial variability of the annual
mean representation of $FCO_2$ in the 10 CMIP5 models against observation product estimates; section
3.2 quantifies the biases in the $FCO_2$ seasonal cycles in the 10 models. Section 3.3 investigates surface
ocean drivers of $FCO_2$ changes (temperature driven solubility and primary production), and finally,
section 3.4 examines the source terms in the DIC surface budget (primary production, entrainment
rates and vertical gradients) and their role in surface $pCO_2$ changes. The discussion (section 4) is an
examination of the mechanisms behind the $pCO_2$ and $FCO_2$ biases in the models.  We conclude with a
synthesis of the main findings and their implications.


## 2. Methods

The Southern Ocean is here defined as the ocean south of the Sub-Tropical Front (STF, defined according to Orsi et al., (1995), 11.3$^{o}$C isotherm at 100m). It is divided into two main domains, the Sub-Antarctic Zone; between the STF and the Polar Front (PF: 2$^{o}$C isotherm at 200m) and the Antarctic Zone, south of the PF. Within the Sub-Antarctic Zone and Antarctic Zone, we further partition the domain into the three main basins of the Southern Ocean i.e. Pacific, Atlantic and the Indian zone.

### 2.1 Observations datasets

We used the Landschützer et al. (2014) data product (FCO$_2$ and partial pressure of CO$_2$ (pCO$_2$) as the main suite of observations-based estimates against which to compare the models throughout the analysis. Landschützer et al. (2014) dataset is synthesized from Surface Ocean CO$_2$ Atlas version 2 (SOCAT2) observations and high resolution winds using a Self-Organizing Map (SOM) through a Feed Forward Neural Network (FNN) approach (Landschützer et al., 2013). While Landschützer et al. (2014) dataset is based on more *in situ* observations (SOCAT2, 15 million source measurements) (Bakker et al., 2014) in comparison to Takahashi et al., (2009) (3 million surface measurements), used in Mongwe et al., (2016). We are nevertheless mindful that due to paucity of observations in the Southern Ocean, this data product is still subject to significant uncertainties, as discussed in Ritter et al. (2018). To evaluate the uncertainty between data products we compare the Landschützer et al. (2014) data with Gregor et al. (2017) data product, which is based on two independent empirical models: Support Vector Regression (SVR) and Random Forest Regression (RFR) as well as against Takahashi et al. (2009) for pCO$_2$ in the Southern Ocean. We compare pCO$_2$ instead of FCO$_2$ firstly, because Gregor et al., (2017) only provided fugacity and pCO$_2$, and being mindful that the choice of wind product and transfer velocity constant in computing FCO$_2$ would increase the level of uncertainty (Swart et al., 2014). Secondly, while the focus of the paper is on the examination biases in the air-sea fluxes of CO$_2$, the major part of our analysis is based on pCO$_2$, which primarily determines the direction and part of the magnitude of the fluxes. We find that the three data products agree on the seasonal phasing of pCO$_2$ in the Sub-Antarctic Zone, but they show differences in the magnitudes (Fig. S1). In the Antarctic Zone, all three datasets agree in both phasing and amplitude (Fig. S1). At this stage it is not clear whether this agreement is due to all the methods converging even with the sparse data or the reason for agreement is the lack of observations. Nevertheless, more independent in situ observations will be helpful to resolve this issue. In this regard float observations from the SOCCOM program (Johnson et al., 2017) and glider observations (Monteiro et al., 2015), for example, are likely to become

helpful in resolving these data uncertainties in addition to ongoing ship-based measurements.

We also used the Takahashi et al. (2009) in situ $FCO_2$ dataset as a complementary source for
comparison of spatial $FCO_2$ properties in the Southern Ocean. Takahashi et al. (2009) data estimates
are comprised of a compilation of about 3 million surface measurements globally, obtained from 1970
– 2000 and corrected for reference year 2000. This dataset is used, as provided, on a $4^o$ (latitude) x $5^o$
(longitude) resolution. Using monthly mean sea surface temperature (SST) and salinity from the World
Ocean Atlas 2013 (WOA13) dataset (Locarnini et al., 2013), we reconstructed total alkalinity (TAlk)
using the Lee et al. (2006) formulation. We also use this dataset as the main observations platform in
section 2.3. To calculate the uncertainty of the computed TAlk, we compared the calculated total
alkalinity ($TAlk_{calc}$) based on ship measurements of SST and surface salinity dataset with actual
observed $TAlk_{obs}$ of the same measurements for a set of winter (August) data collected in the Southern
Ocean. We found that $TAlk_{calc}$ compares well with $TAlk_{obs}$ ($R^2$ = 0.79) (Fig. S2, Supplementary). We
then used this computed monthly TAlk and $pCO_2$ from Landschützer et al. (2014) to compute DIC
using CO2SYS (Pierrot et al., 2006, http://cdiac.ornl.gov/ftp/co2sys/CO2SYS_calc_XLS_v2.1), using K1,
K2 from Mehrbach et al. (1973) refitted by Dickson and Millero (1987). For interior ocean DIC, we
used the Global Ocean Data Analysis Project version 2 (GLODAP2) annual means dataset (Lauvset et
al., 2016). The Mixed Layer Depth (MLD) data was taken from de Boyer Montégut et al. (2004), on a $1^o$
x $1^o$ grid, the data is provided as monthly means climatology and was used as provided. We also use
satellite chlorophyll dataset from Johnson et al. (2013).

**2.2 CMIP5 Model data**

We used 10 models from the Coupled Model Intercomparison Project version 5 (CMIP5) Earth System
Models (ESM) shown in Table 1. The selection criterion for the models was based on the availability of
essential variables for the analysis in the CMIP5 data portal (http://pcmdi9.llnl.gov) at the time of
writing: i.e. monthly $FCO_2$, $pCO_2$, chlorophyll, net primary production (NPP), surface oxygen, surface
Dissolved Inorganic Carbon (DIC), MLD, Sea Surface Temperature (SST), vertical temperature fields
and annual DIC for the historical scenario. The analysis is primarily based on the climatology over
1995 – 2005, which was selected to match a period closest to the available observational data product
(Landschützer et al. (2014), 1998 – 2011). However, we do examine the consistency of the seasonality
of $FCO_2$ over periods longer than 10 years by comparing the seasonal cycle of $FCO_2$ and temporal
standard deviation of 30 years (1975 – 2005) vs 10 years (1995 – 2005) for HadGEM2-ES and
CanESM2. We find that the seasonal cycle of $FCO_2$ remains consistent (R = 0.99) in both HadGEM2-ES
and CanESM2 over 30 years (Fig. S3). All CMIP5 model outputs were regridded into a common $1^{\circ}$x$1^{\circ}$
regular grid throughout the analysis, except for annual $CO_2$ mean fluxes, which were computed on the
original grid for each model.

**Table 1:** A description of the 10 CMIP5 ESMs that were used in this analysis. It shows the ocean
resolution, atmospheric resolution, and available nutrients for the biogeochemical component, sea-ice
model, vertical levels and the marine biogeochemical component for each ESM.

| Full name and Source | Model Name | Ocean Resolution | Atmospheric Resolution | Nutrients | Sea ice model | Veridical Coordinate & Levels | Ocean Biology | Reference |
|---|---|---|---|---|---|---|---|---|
| **Canadian Centre for Climate Modelling and Analysis, Cananda** | CanESM2 | CanOM4 $0.9^{\circ}$x$1.4^{\circ}$ | $2.8125^{\circ}$ x $2.8125^{\circ}$ | N (accounts for Fe limitation) | CanSIM1 | z 40 levels | NPZD | Zahariev et al., 2008 |
| **Centro Euro-Mediterraneo Sui Cambiamenti Climatici, Italy** | CMCC-CESM | OPA8.2 $0.5$-$2^{\circ}$x$2^{\circ}$ | $3.8^{\circ}$ x $3.7^{\circ}$ | P, N, Fe, Si | CICE4 | z 21 levels | PELAGOS | Vichi et al., 2007 |
| **Centre National de Recherches Météorologiqu es-Centre Européen de Recherche et de Formation Avancée en Calcul Scientifique, France** | CNRM-CM5 | NEMOv3.3 $1^{\circ}$ | $1.4^{\circ}$ | P, N, Fe, Si | GELATO5 | z 42 levels | PISCES | Séférian et al., 2013 |
| **Institut Pierre-Simon Laplace, France** | IPSL-CM5A-MR | NEMO2.3 $0.5$-$2^{\circ}$ x $2^{\circ}$ | $2.58^{\circ}$ x $1.25^{\circ}$ | P, N, Fe, Si | LIM2 | z 31 levels | PISCES | Séférian et al., 2013 |
| **Max Plank Institute for Meteorology, Germany** | MPI-ESM-MR | MPIOM $1.41^{\circ}$×$0.89^{\circ}$ | $1.875^{\circ}$ x $1.875^{\circ}$ | P, N, Fe, Si | MPIOM | z 40 levels | HAMOCC 5.2 | Ilyina et al., 2013 |

| Community Earth System Model, USA | CESM1-BGC | 0.3° x1° | 0.9° x 1.25° | (P), N, Fe, Si | | z 60 levels | BEC | Moore et al., 2004 |
|---|---|---|---|---|---|---|---|---|
| Norwegian Earth System Model, Norway | NorESM1-ME | MICOM 0.5° x 0.9° | 2.5° x 1.9°, | P, N, Fe, Si | CICE4.1 | ρ 53 levels | HAMOCC | Tjiputra et al., 2013 |
| Geophysical Fluid Dynamics Laboratory Earth System Model, USA | GFDL-ESM2M | 0.3° x 1° | 2.5° x 2.0° | N, P, SiO4, Fe | SISp2 | z 50 levels | TOPAZ2 | Dunne et al., 2013 |
| Meteorological Research Institute-Earth System Model Version 1, Japan | MRI-ESM | 0.5° x 1° | | P,N | MRI.COM3 | σ-z 51 levels | NPZD | Adachi et al., 2013 |
| Hadley Global Environment Model 2 – Earth System, UK | HadGEM-ES | 0.3° x 1° | 2.5° x 2.0° | N,Fe,S | | 40 levels | Diat-HadOCC | Palmer and Totterdell, 2001 |



**2.3 Sea-Air CO$_2$ Flux Drivers: The Seasonal Cycle Diagnostic Framework**

The seasonal cycle of the ocean-atmosphere pCO$_2$ gradient ($\Delta$pCO$_2$) is the main driver of the variability
of FCO$_2$ over comparable periods (Sarmiento and Gruber, 2006; Wanninkhof et al., 2009; Mongwe et
al., 2016). Wind speed plays a dual role as a driver of FCO$_2$: it drives the seasonal evolution of
buoyancy-mixing dynamics, which influences the biogeochemistry and upper water column physics
(but these processes are incorporated into the variability of the DIC), as well as the rate of gas
exchange across the air-sea interface (Wanninkof et al., 2013). However, because winds in the
Southern Ocean do not have large seasonal variation (Young, 1999), for this analysis, we neglect the
role of wind as a secondary driver of the seasonal cycle of FCO$_2$. Consequently, the seasonal cycle of
FCO$_2$ is directly linked to surface pCO$_2$ variability, influenced by changes in temperature, salinity, TAlk
and DIC and macronutrients (Sarmiento and Gruber, 2006; Wanninkhof et al., 2009). In this analysis
we use this assumption as a basis to explore how the seasonal variability of temperature and DIC
regulate the seasonal cycle of pCO$_2$ in CMIP5 models relative to observational product estimates.

The seasonal cycle diagnostic framework was developed as a way of scaling the relative contributions
from the rates of change of SST- and the total DIC-driven changes to the seasonal cycle of $pCO_2$ on to a
common DIC scale (Mongwe et al., 2016). We use the framework to explore how understanding
differences emerging from the temperature- and DIC-driven $CO_2$ variability could be helpful as a
diagnostic of the apparent observations –model seasonal cycle biases in the Southern Ocean.

The total rate of change of DIC in the surface layer consists of the contribution of air-sea exchanges,
biological, vertical and horizontal transport-driven changes (Eq. 1).

$$\left(\frac{\partial DIC}{\partial t}\right)_{Tot} = \left(\frac{\partial DIC}{\partial t}\right)_{air-sea} + \left(\frac{\partial DIC}{\partial t}\right)_{Bio} + \left(\frac{\partial DIC}{\partial t}\right)_{Vert} + \left(\frac{\partial DIC}{\partial t}\right)_{Hor} \qquad (1)$$
Because we used zonal means from medium resolution models, we assume that the horizontal terms
are negligible, though mindful that there could be a seasonal cycle in the divergence of the horizontal
transport due to a latitudinal gradient in DIC perturbed by Ekman flow in some regions of the Sub-
Antarctic Zone (Rosso et al., 2017). This leaves air-sea exchange, vertical fluxes (advection and
diffusion) and biological processes as the dominant drivers of DIC.
Since temperature does not affect DIC changes directly, but only $pCO_2$ through solubility, it was
necessary to scale the influence of temperature into equivalent DIC units in order to compare the
influence of temperature vs DIC control of surface $pCO_2$ variability. Thus in order to constrain the
contribution of temperature on the seasonal variability of $pCO_2$ and $FCO_2$ we derived a new synthetic
temperature-linked term "DIC equivalent" ($DIC_T$) defined as: *the magnitude of DIC change that would*
*correspond to a change in $pCO_2$ driven by a particular temperature change.* In this way the $\Delta pCO_2$
driven solely by modelled or observed temperature change, is converted into equivalent DIC units,
which allows its contribution to be scaled against the observed or modelled total surface DIC change
(Eq.1). Shifts between temperature and DIC control of $pCO_2$ are in effect tipping points because they
reflect major shifts in the mechanisms that drive $pCO_2$ variability. We use this as the basis to
investigate the possible mechanisms behind model biases in the seasonal cycle of $pCO_2$.
This calculation of $DIC_T$ is done in two steps: firstly, the temperature impact on $pCO_2$ is calculated
using the Takahashi et al. (1993) empirical expression that linearizes the temperature dependence of
the equilibrium constants.
$$\left(\frac{\partial p_{O_2}}{\partial t}\right)_{SST} = 0.0423 \times pCO_2 \times \left(\frac{\partial pCO_2}{\partial SST}\right) \qquad (2)$$
Though this relationship between dSST and $dpCO_2$ is based on a linear assumption (Takahashi et al.,
1993), this formulation has been shown to hold and has been widely used in literature (e.g. Bakker et
al., 2014; Feely et al., 2004; Marinov and Gnanadesikan, 2011; Takahashi et al., 2002; Wanninkhof et
al., 2009; Landschützer et al., 2018). We show in the supplementary material that the extension of this
expression into polar temperature ranges (SST < 2°C) only introduces a minor additional uncertainty
of 4 -5% (SM Fig. S4).
Secondly, the temperature-driven change in $pCO_2$ is converted to an equivalent $DIC_T$ using the Revelle
factor.
$$\left(\frac{\partial DIC_T}{\partial t}\right)_{SST} = \frac{DIC}{\gamma_{DIC} \times pCO_2}\left(\frac{\partial pCO_2}{\partial t}\right)_{SST} \tag{3}$$
Here we also used a fixed value for the Revelle Factor ($\gamma_{DIC}=14$), typical of polar waters in the Southern
Ocean in order to assess the error linked to this assumption. We recomputed the Revelle factor in the
Sub-Antarctic and Antarctic Zones using annual mean climatologies of TAlk, salinity, sea surface
temperature and nutrients. Firstly, we examined DIC changes for the nominal range of $pCO_2$ change
(340 – 399 µatm:1 µatm intervals) and then used this dataset to derive the Revelle factor. The range of
calculated Revelle factors in the Southern Ocean was between $\gamma_{DIC} \sim 12 - 15.5$ with an average of $\gamma_{DIC} =$
$13.9\pm1.3$. This justifies our use of $\gamma_{DIC} = 14$ for the conversion of the solubility-driven $pCO_2$ change to
an equivalent DIC ($DIC_T$) throughout the analysis. We have provided the uncertainty that this
conversion makes into the temperature constraint $DIC_T$, by using the upper and lower limits of the
Revelle factor ($\gamma_{DIC} = 12 - 15.5$) in the model framework. In the Supplementary Material (Fig. S5) we
show examples for observations in the Sub-Antarctic and Antarctic Zones, which indicate that the
extremes of the Revelle factor values ($\gamma_{DIC} = 12 - 15.5$) do not alter the phasing or magnitude of the
relative controls of temperature or DIC on the seasonal cycle of $pCO_2$.
The rate of change of DIC was discretized on a monthly mean as follows:

$$\left(\frac{\partial DIC_T}{\partial t}\right)_{SST} \approx \left(\frac{\Delta DIC}{\Delta t}\right)_{n,l} = \frac{DIC_{n+1,l}-DIC_{n,l}}{1\, month} \tag{4}$$

Where n is time in month, l is vertical level (in this case the surface, l=1). We here take the forward
derivative such that November rate is the difference between 15 November and 15 December, thus
being centered at the interval between the months.
Finally, to characterize periods of temperature or DIC dominance as main drivers of the instantaneous
(monthly) $pCO_2$ change we subtract Eq. 1 from Eq. 4, which yields a residual indicator $M_{T-DIC}$ Eq. 5. $M_{T-}$

DIC is then used as indicator of the dominant driver of instantaneous $pCO_2$ changes in this scale monthly time scale.

$$M_{T-DIC} = \left|\left(\frac{\partial DIC_T}{\partial t}\right)_{SST}\right| - \left|\left(\frac{\partial DIC}{\partial t}\right)_{Tot}\right| \qquad (5)$$

$M_{T-DIC} > 0$ indicates that the $pCO_2$ variability is dominated by the temperature-driven solubility and when $M_{T-DIC} < 0$, it indicates that $pCO_2$ changes are mainly modulated by DIC processes (i.e. Biological $CO_2$ changes and vertical scale physical DIC mechanisms). We also examine the following DIC processes; i.) Biological DIC changes using chlorophyll, NPP, export carbon, surface oxygen, and ii.). Physical DIC mechanisms using estimated entrainment rates at the base of the mixed layer. Details of this calculation are in section 2.4.

In the Southern Ocean, salinity and TAlk are considered lower-order drivers of the seasonal cycle of $pCO_2$ (Takahashi et al., 1993). In the supplementary material (Fig. S6), we show that salinity and TAlk do not play a major role as drivers of the local seasonal cycle of $pCO_2$. We do so by computing the equivalent rate of change of DIC resulting from seasonal variability of salinity and TAlk as done for temperature (Eq. 2), i.e. still assuming empirical linear relationships from Takahashi et al. (1993): $\left(\frac{\ln (pCO_2)}{\ln (TAlk)} \approx -9.4\right)$ and $\left(\frac{\ln (pCO2)}{\ln (Sal)} = 0.94\right)$. By applying these relationships to the model data, we confirmed that indeed salinity and TAlk are secondary drivers of $pCO_2$ changes i.e. $\left[\left(\frac{\partial DIC}{\partial t}\right)_{Tot}\right]_{average} \approx$ 5 µmol kg$^{-1}$ month$^{-1}$, while $\left[\left(\frac{\partial DIC}{\partial t}\right)_{Tot}\right]_{average} \approx 0.6$ µmol kg$^{-1}$ month$^{-1}$ and $\left[\left(\frac{\partial DIC}{\partial t}\right)_{TAlk}\right]_{maximum} \approx 0.4$ µmol kg$^{-1}$ month$^{-1}$.

## 2.4 Entrainment mixing

$CO_2$ uptake by the Southern Ocean has been shown to weaken during winter linked to the entrainment of sub-surface DIC as the MLD deepens (e.g. Lenton et al., 2013; Metzl et al., 2006; Takahashi et al., 2009). Here we estimate this rate of entrainment (RE) using Eq. 6, which estimates the advection of preformed DIC at the base of the mixed layer:

$$RE = U_e \left(\frac{\partial DIC}{\partial z}\right)_{MLD} \qquad (6)$$

$$RE_n = \left(\frac{\Delta MLD_n}{\Delta t}\right)\left(\frac{\Delta DIC}{\Delta z}\right)_{n,MLD} \qquad (7)$$

$$\left(\frac{\Delta DIC}{\Delta z}\right)_{n,MLD} = \frac{DIC_{n,MLD_{n+1}} - DIC_{n,MLD_n}}{\Delta z} \qquad (8)$$

In which $U_e$ is an equivalent entrainment velocity based on the rate of change of the MLD and n is the
time in months. This approximation of vertical entrainment is necessary as it is not possible to
compute this term from the CMIP5 data because the vertical DIC distribution is only available as an
annual means. We use the entrainment rates to estimate the influence of subsurface/bottom DIC
changes on surface DIC changes and subsequently $pCO_2$ and $FCO_2$. Because we are mainly interested in
the period autumn- winter, where the MLD ≥ 60 m in the Sub-Antarctic Zone and ≥ 40 m in the
Antarctic Zone at this depth seasonal variations in DIC are anticipated to be minimal – these estimates
can be used. The monthly and annual mean DIC from a NEMO PISCES 0.5 x 0.5° model output were
used to estimate the uncertainty by comparing RE computed from both (Dufour et al., 2013). We found
the annual and monthly estimates to be indeed comparable with minimal differences (not shown). It is
noted as a caveat that this rate of entrainment is only a coarse estimate because we were using annual
means, and is intended only for the autumn-winter period when MLDs are deepened.

## 3. Results

### 3.1 Annual climatological sea-air $CO_2$ fluxes

The annual mean climatological distribution of $FCO_2$ in the Southern Ocean obtained from
observational products is spatially variable, but mainly characterized by two key features: (i) $CO_2$ in-
gassing north of 50°-55°S (Polar Frontal Zone, PFZ) within and north of the Sub-Antarctic Zone, and
(ii), $CO_2$ out-gassing between the PF (~ 58°S) and the Marginal Ice Zone (MIZ, ~ 60° - 68°S) (Fig. 1a-b).
Most CMIP5 models broadly capture these features, however, they also show significant differences in
space and magnitude between the basins of the Southern Ocean (Fig. 1).  With the exception of CMCC-
CESM, which shows a northerly-extended $CO_2$ out-gassing band between about 40°S and 50°S, CMIP5
models generally show the $CO_2$ out-gassing zone between 50°S- 70°S in agreement with observational
estimates (Fig. 1).

The analyzed 10 CMIP5 models show a large spatial dispersion in the spatial representation of the
magnitudes of $FCO_2$ with respect to observations (Fig. 1, Table 2).  They generally overestimate the
upwelling-driven $CO_2$ out-gassing (55°S -70°S) in some basins relative to observations. IPSL-CM5A,
CanESM2, MPI-ESM, GFDL-ESM2M and MRI-ESM, for example, show $CO_2$ out-gassing fluxes reaching
up to 25 g m$^{-2}$ yr$^{-1}$, while observations only show a maximum of 8 g m$^{-2}$ yr$^{-1}$ (Fig. 1).  Between 40°S-
56°S (Sub-Antarctic Zone), observations and CMIP5 models largely agree, showing a $CO_2$ in-gassing
feature, which is mainly attributable to biological processes (McNeil et al., 2007; Takahashi et al.,
2012). South of 65°S, in the MIZ, models generally show an excessive $CO_2$ in-gassing with respect to

observations (with the exception of CanESM2, IPSL-CM5A-MR and CNRM-CM5). Note that as much as this bias south of the MIZ might be a true divergence of CMIP5 models from the observed ocean, it is also possibly due to the lack of observations in this region, especially during the winter season (Bakker et al., 2014; Monteiro, 2010).

Table 2 shows the Pattern Correlation Coefficient (PCC) and the Root Mean Square Error (RMSE), which are here used to quantify the model spatial and magnitude performances against Landschützer et al. (2014) data product.  Out of the 10 models, six show a moderate spatial correlation with Landschützer et al. (2014)  (PCC = 0.40 – 0.60), i.e. CNRM-CM5, GFDL-ESM2M, HadGEM2-ES, IPSL-CM5A-MR, CESM1-BGC, NorESME-ME and CanESM2. While MPI-ESM-MR (PCC = 0.37), MRI-ESM (PCC = 0.36) and CMCC-CESM (PCC = -0.09) show a weak to null spatial correlation with observations, the latter is mainly due to the overestimated out-gassing region. Spatially, GFDL-ESM2M and NorESM1-ME are the most comparable to Landschützer et al. (2014), (RMSE < 9), while CCMC-CESM, CanESM2, MRI-ESM and CNRM-CM5 shows the most differences (REMSE > 15). The rest of the models show a modest comparison (RSME 9 – 11).

NorESM1-ME and CESM1-BGC are the only two of the 10 models showing a consistent spatial (REMSE < 9) and magnitude (PCC ≈ 0.50) performance. From Table 2, it is evident that an appropriate representation of the spatial properties of $FCO_2$ with respect to observations does not always correspond to comparable magnitudes. CanESM2 for example, shows a good spatial comparison (PCC = 0.54), yet a poor estimation of the magnitudes (RMSE = 19.5). In this case this is caused by an overestimation of $CO_2$ uptake north of 55$^o$S ($\approx$ - 28 g m$^{-2}$ yr$^{-1}$) and $CO_2$ out-gassing (> 25 g m$^{-2}$ yr$^{-1}$) in the Antarctic zone, resulting in a net total Southern Ocean annual weak sink (-0.05 Pg C m$^{-2}$ yr$^{-1}$).

## 3.2 Sea-Air CO$_2$ Flux Seasonal Cycle Variability and Biases

The seasonal cycle of $FCO_2$ is shown in Fig. 2. The seasonality of $FCO_2$ in the 10 CMIP5 models shows a large dispersion in both phasing and amplitude, but mostly disagrees with observations in the phase of the seasonal cycle as well as disagreeing with each other. More quantitatively, CMIP5 models show weak to negative correlations with the Landschützer et al. (2014) data product in the Sub-Antarctic Zone and have slightly higher correlations in the Antarctic Zone (see supplementary Fig. S7).  This discrepancy is consistent with the findings of Anav et al. (2013), who however used fixed latitude criteria. Based on the phasing, the seasonality of $FCO_2$ in CMIP5 models can be a priori divided in two main groups: group-DIC models, comprising of MPI-ESM, HadGEM-ES and NorESM1-ME, and group-SST models, the remainder i.e. GFDL-ESM2M, CMCC-CESM, CNRM-CERFACS, IPSL-CM5A-MR, CESM1-BGC, MRI-ESM and CanESM2. The naming convention is suggestive of the mechanism driving the

seasonal cycle, as will be clarified further on. A similar grouping was also identified by Kessler and
Tjiputra (2016) using a different criterion. Fig. 3 shows the seasonal cycle of $FCO_2$ of an equally-
weighted ensemble of the two groups compared to observations; the shaded area shows the decadal
standard deviation for the models and the Landschützer et al. (2014) data product for 1998-2014
standard deviation in the various regions.

In the Sub-Antarctic Zone, the observational products show a weakening of $CO_2$ uptake during winter
(less negative values in June-August) with values close to the zero at the onset of spring (September)
in all three basins. Similarly, during the spring season, all three basins are seen to maintain a steady
increase of $CO_2$ uptake until mid-summer (December), while they differ during autumn (March-May).
The Pacific basin shows an increase in $CO_2$ uptake during autumn that is not observed in the other
basins (only marginally in the Indian zone). In the Antarctic zone, the observed $FCO_2$ seasonal cycle is
mostly similar in all three basins (Fig. 3d-f). While this seasonal cycle consistency may suggest a
spatial uniformity of the mechanisms of $FCO_2$ at the Antarctic, we are also mindful that this may be due
to a result of the paucity of observations in this area. In the Antarctic Zone, all three basins show a
weakening of uptake or increasing of out-gassing from the onset of autumn (March) until mid-winter
(June–July). The winter $CO_2$ out-gassing is followed by a strengthening of the $CO_2$ uptake throughout
spring to summer, when it reaches a $CO_2$ in-gassing peak.

The differences in the seasonal cycle of $FCO_2$ across the three basins of the Sub-Antarctic Zone found in
the observational product (Fig. 2) are likely a consequence of spatial differences seen in Fig. 1. To
verify this, we calculated the correlation between the seasonal cycles from the Landschützer et al.
(2014) observational product in the three basins (Fig. 4). The $FCO_2$ seasonal cycle in the Sub-Antarctic
Atlantic and Indian basins are similar (R = 0.8), while the other basins are quite different to one
another  (R = -0.1 for Pacific – Atlantic and R ~ 0.4 for Pacific – Indian). Contrary to the observational
product, CMIP5 models show the same seasonal cycle phasing across all three basins in the Sub-
Antarctic Zone (basin – basin correlation coefficients are always larger than 0.50 in Fig. 4 despite the
spatial differences in Fig. 2, with the exception of three models (i.e. CMCC-CESM, CESM-BGC1 and
GFDL-ESM2M)). Thus, contrary to Landschützer et al. (2014), CMIP5 models shows a zonal
homogeneity in the seasonal cycle of $FCO_2$, which may suggest that the drivers of $CO_2$ are less regional.
In the Antarctic Zone, CMIP5 models agree with observations in the spatial uniformity of the seasonal
cycle of $FCO_2$ across the three basins.

Group-DIC models are characterized by an exaggerated $CO_2$ uptake during spring-summer (Fig. 3) with
respect to observation estimates and $CO_2$ out-gassing during winter. These models generally agree
with observations in the phasing of $CO_2$ uptake during spring, but overestimate the magnitudes. It is
worth noting that the seasonal characteristics of group-DIC models are mostly in agreement with the
observations in the Atlantic and Indian basin in Sub-Antarctic Zone (R > 0.5 in Fig. 4). The large
standard deviation ($\sim$ 0.01 g C m$^{-2}$ day$^{-1}$) during the winter and spring-summer seasons in the Atlantic
basin shows that though group-DIC models agree in the phase, magnitudes vary considerably (Fig. 3b).
For example MPI-ESM reaches up to 0.06 g C m$^{-2}$ day$^{-1}$ out-gassing during winter, while HadESM2-ES
and NorESM2 peak only at $\sim$ 0.03 g C m$^{-2}$ day$^{-1}$. Group-SST models on the other hand are characterized
by a $CO_2$ out-gassing peak in summer (Dec-Feb) and a $CO_2$ in-gassing peak at the end of autumn (May),
and their phase is opposite to the observational estimates in the Atlantic and Indian basins (Fig. 3b,c).
Group-SST models only show a strengthening of $CO_2$ uptake during spring in the Indian basin.
Interestingly, group-SST models compare relatively well with the observed $FCO_2$ seasonal cycle in the
Pacific basin, whereas group-DIC models disagree the most with the observed estimates (Fig. 3a). This
phasing difference within models and against observed estimates probably suggests that the
disagreement of CMIP5 models $FCO_2$ with observations is not a matter of a relative error/constant
magnitude offset, but most likely points to differences in the seasonal drivers of $FCO_2$.

In the Antarctic Zone (Fig. 3d-f), both group-DIC and group-SST models perform better than in the
Sub-Antarctic, in respect of phasing and amplitude in as shown by the correlation analysis in Fig. S7.
Models reflect comparable $pCO_2$ seasonality in the different basins of the AZ to the observational
products (Fig. 4, with the exception of MRI-ESM and CanESM2 where R < 0 for all three basins). Here
$FCO_2$ magnitudes oscillate around zero with the largest disagreements occurring during mid-summer,
where observation estimates show a weak $CO_2$ sink ($\approx$ - 0.03 gC m$^{-2}$ day$^{-1}$), and group-SST show a zero
net $CO_2$ flux and a strong uptake in group DIC (e.g. $\approx$ -0.12 gC m$^{-2}$ day$^{-1}$ in the Pacific basin). The large
standard deviation ($\approx$ 0.01 gC m$^{-2}$ day$^{-1}$) here indicates considerable differences among models (Fig.
3d-f).

**3.3 Seasonal Scale Drivers of Sea-Air $CO_2$ Flux**

We now examine how changes in temperature and DIC regulate $FCO_2$ variability at the seasonal scale
following the method described in Sec. 2.3. Fig. 5 shows the monthly rates of change of SST (dSST/dt)
for the 10 models compared with WOA13 SST. CMIP5 generally shows agreement in the timing of the
switch from surface cooling (dSST/dt < 0) to warming (dSST/dt > 0) and vice versa; i.e. March
(summer to autumn), and September (winter to spring) respectively. In both the Sub-Antarctic and
Antarctic Zone CMIP5 models agree with observations in this timing (Fig. 5). However, while they
agree in phasing, the amplitude of these warming and cooling rates are overestimated with respect to
the WOA13 dataset with the exception of NorESM1-ME. Subsequently these differences in the
magnitude of dSST/dt have important implications for the solubility of $CO_2$ in seawater; larger
magnitudes of |dSST/dt| are likely to enhance the response of the $pCO_2$ to temperature through $CO_2$
solubility changes. For example, because the observations in the Indian basin show a warming rate of
about 0.5°C month$^{-1}$ lower compared to the other two basins, we expect a relatively weaker role of
surface temperature in this basin.

As described in sec. 2.3, the computed dSSt/dt magnitudes were used to estimate the equivalent rate
of change of DIC driven by $CO_2$ solubility using Eq. 2. The seasonal cycle of $| (dDIC_T/dt)_{SST} |$ vs
$|(dDIC/dt)_{Tot} |$, for the 10 models and observations is presented in the supplementary material (Fig.
S8) where we show the seasonal mean of $M_{T-DIC}$ from (Eq. 3). As articulated in sec. 2.3, $M_{T-DIC}$ (Fig. 6) is
the difference between the total surface DIC rate of change of DIC (Eq. 1) and the estimated equivalent
temperature-driven solubility DIC changes Eq. 3, such that when $| (dDIC_T/dt)_{SST} | > |(dDIC/dt)_{Tot} |$,
temperature is the dominant driver of the instantaneous $pCO_2$ changes, and conversely when |
$(dDIC_T/dt)_{SST} | < |(dDIC/dt)_{Tot} |$,DIC processes are the dominant mode in the instantaneous $pCO_2$
variability.  The models showing the former feature are SST-driven and belong to group-SST, while the
models showing the latter are DIC-driven and belong to group-SST.

According to the $M_{T-DIC}$ magnitudes in Fig. 6, the seasonal cycle of $pCO_2$ in the observational estimates
is predominantly DIC-driven most of the year in both the Sub-Antarctic and Antarctic Zone. Note that,
however, during periods of high |dSST/dt|, i.e. autumn and spring, observations show a moderate to
weak DIC control ($M_{T-DIC} \approx 0$). The Antarctic Zone is mostly characterized by a stronger DIC control
(mean Annual $M_{T-DIC} > 3$) except for during the spring season (Fig. 6). Consistent with the similarity
analysis presented in Fig. 4, the Antarctic Zone shows coherence in the sign of the temperature –DIC
indicator ($M_{T-DIC} > 0$) within the three basins.


**3.4 Source terms in the DIC surface budget**

To further constrain the surface DIC budget in Eq. 1, we examine the role of the biological source term
using chlorophyll and Net Primary Production (NPP) as proxies. Fig. 8 shows the seasonal cycle of
chlorophyll, NPP and the rate of surface DIC changes (dDIC/dt). The observed seasonal cycle of
chlorophyll (Johnson et al., 2013) shows a similar seasonal cycle within the three basins during the
spring-summer seasons (autumn-winter data are removed due to the satellite limitation) in both the
Sub-Antarctic and Antarctic Zone. Magnitudes are however different in the Sub-Antarctic Zone; the
Atlantic basin shows larger chlorophyll magnitudes (Chlorophyll reach up to 1.0 mg m$^{-3}$) compared to
the Pacific and Indian basins (Chl < 1 mg m$^{-3}$).

CMIP5 models here show a clear partition between group-DIC and group-SST models. While they
mostly maintain the same phase, group-DIC shows larger amplitudes of chlorophyll relative to group-
SST and observed estimates in the Sub-Antarctic Zone. This difference is even clearer in NPP
magnitudes, where group-DIC models show a maximum of NPP > 1 mmol m$^{-2}$ s$^{-1}$ in summer, while
group-SST magnitudes shows about half of it. Except for CESM1-BGC and CMCC-CESM (and NorESM1-
ME for NPP), each CMIP5 model generally maintains a similar chlorophyll seasonal cycle (phase and
magnitude) in all three basins of the Southern Ocean. This is contrary to the observations, which show
differences in the magnitude. Consistent with the observational product, CESM1-BGC simulates larger
amplitude in the Atlantic basin. While CMCC-CESM also has this feature, it also shows an
overestimated chlorophyll peak in the Indian basin. In the Antarctic Zone both observations and
CMIP5 models generally agree in both phase and magnitude (except for CanESM2) of the seasonal
cycle of chlorophyll in all three basins.
We now examine the influence of the vertical DIC rate in Eq. 1, using estimated entrainment rates (RE,
Eq. 5) based on MLD and vertical DIC gradients (see sec. 2.3).  Fig. 7 shows the seasonal changes of
MLD compared with the rate from the observational product. CMIP5 models largely agree on the
timing of the onset of MLD deepening (February in the Pacific basin, and March for the Atlantic and
Indian basin) and shoaling (September) in the Sub-Antarctic Zone (with the exception of NorESM1-ME
and IPSL-CM5A in the Pacific basin). The Indian basin generally shows deeper winter MLD in both
observations and CMIP5 models in the Sub-Antarctic Zone. Note that while CMIP5 models generally
show the observed deeper MLDs in the Indian basin, they show a large variation; for example, the
winter maximum depth ranges from 100 m (CMCC-CESM, pacific basin) to 350 m (CanESM2, Indian
basin) in the Sub-Antarctic Zone.  In the Antarctic Zone CMIP5 models are largely in agreement on the
timing of the onset of MLD deepening (February), but also variable in their winter maximum depth. It
is worth noting that the observed MLD seasonal cycle might be biased due to limited in situ
observations particularly in the Antarctic Zone (de Boyer Montégut et al., 2004).
The estimated RE values in Fig. 10 show that almost all CMIP5 (with the exception of NorESM1-ME)
entrain subsurface DIC into the mixed layer during autumn–winter in agreement with the
observational estimates. In the Sub-Antarctic Zone, the estimates using the observational products
show the strongest entrainment in the Atlantic basin in May (RE reaches up to 10 µmol kg$^{-1}$ month$^{-1}$),
while it is lower in the other basins. In the Antarctic Zone, observed RE conversely shows stronger
entrainment rates in the Pacific and Indian basin (RE > 15 µmol kg$^{-1}$ month$^{-1}$) in comparison to the
Atlantic basin (RE = 11 µmol kg$^{-1}$ month$^{-1}$). CMIP5 models entrainment rates are variable but not
showing any particular deficiency when compared with the observational estimates. Also, the group-
DIC and group-SST models show no clear distinction, the major striking features being the relatively

stronger entrainment in MPI-ESM and CanESM2 across the three basins in the Sub-Antarctic Zone in mid to late winter (RE = 15 µmol kg$^{-1}$ month$^{-1}$), and the large winter entrainment in IPSL-CM5A-MR in the Antarctic Pacific basin. The supply of DIC to the surface due to vertical entrainment is therefore generally comparable between model simulations and the available estimate.

However, our RE estimates are estimated at the base of the mixed layer, which is not necessarily a complete measure of the vertical flux of DIC at the surface. We therefore investigate the annual mean vertical DIC gradients in Fig. 10 as an indicator of where the surface uptake processes occur. The simulated CMIP5 profiles are similar to GLODAP2, but some differences arise. In the Sub-Antarctic Zone, GLODAP2 shows a shallower surface maximum in the Atlantic basin consistent with higher biomass in this basin (Fig. 8) ($(dDIC/dz)_{smax}$ = 0.55 µmol kg$^{-1}$ m$^{-1}$, at 50 m) compared to the Pacific ($(dDIC/dz)_{smax}$ = 0.60 µmol kg$^{-1}$ m$^{-1}$, at 80 m) and Indian basin ($(dDIC/dz)_{smax}$ = 0.40 µmol kg$^{-1}$ m$^{-1}$, at 80 m). CMIP5 models generally do not show this feature in the Sub-Antarctic Zone, except for CESM1-BGC1 ($(dDIC/dz)_{smax}$ = 0.50 µmol kg$^{-1}$ m$^{-1}$, at 50 m). Instead, they show the surface maxima at the same depth in all three basins. In the Antarctic Zone both CMIP5 models and observations show larger $(dDIC/dz)_{smax}$ magnitudes and nearer surface maxima (with the exception of CanESM2 and CESM1-BGC). This difference in the position and magnitude of the DIC maxima between the Sub-Antarctic and Antarctic Zone has important implications for surface DIC changes and subsequently pCO$_2$ seasonal variability. Because of the nearer surface DIC maxima in the Antarctic Zone, surface DIC changes are mostly influenced by these strong near-surface vertical gradients than MLD changes. This implies that even if the entrainment rates at the base of the MLD are comparable between the Sub-Antarctic and the Antarctic, the surface supply of DIC may be larger in the Antarctic Zone.

# 4. Discussion

Recent studies have highlighted that important differences exist between the seasonal cycle of pCO2 in models and observations in the Southern Ocean (Lenton et al., 2013; Anav et al., 2015; Mongwe, 2016). Paradoxically, although the models may be in relative agreement for the mean annual flux, they diverge in the phasing and magnitude of the seasonal cycle (Lenton et al., 2013; Anav et al., 2015; Mongwe, 2016). These differences in the seasonal cycle raise questions about the climate sensitivity of the carbon cycle in these models because they may reflect differences in the process sensitivities to drivers that are themselves climate sensitive.

In this study we expand on the framework proposed by Mongwe et al. (2016), which examined the
competing roles of temperature and DIC as drivers of $pCO_2$ variability and the seasonal cycle of $pCO_2$ in
the Southern Ocean, to explain the mechanistic basis for seasonal biases of $pCO_2$ and $FCO_2$ between
observational products and CMIP5 models.  This analysis of 10 CMIP5 models and one observational
product (Landschutzer et al., 2014) highlighted that although the models showed different seasonal
cycles (Fig. 2), they could be grouped into two categories (SST- and DIC-driven) according to their
mean seasonal bias of temperature or DIC control (Fig. 3 & 6).

A few general insights emerge from this analysis. Firstly, despite significant differences in the spatial
characteristics of the mean annual fluxes (Fig. 1), models show unexpectedly greater inter-basin
coherence in the phasing seasonal cycle of $FCO_2$ and SST-DIC control than observational products (Fig.
3 & 6). Clear inter-basin differences have been highlighted in studies on the climatology and
interannual variability that examined $pCO_2$ and $CO_2$ fluxes based on data products (Landschutzer et al.,
2015; Gregor et al., 2017), as well as phytoplankton chlorophyll based on remote sensing (Thomalla et
al., 2011; Carranza et al., 2016).   Briefly, the Atlantic basin shows the highest mean primary
production in contrast to the Pacific basin, which has the lowest (Thomalla et al., 2011).  Similarly,
strong inter-basin differences for $pCO_2$ and $FCO_2$ have been highlighted and ascribed to SST control
(Landschützer et al., 2016) and wind stress - mixed layer depth (Gregor et al., 2017).  The combined
effect of these regional differences in forcing of $pCO_2$ and $FCO_2$ would be expected to be reflected in the
CMIP5 models as well.  A quantitative analysis of the correlation of the phasing of the seasonal cycle of
$FCO_2$ between basins for different models shows that all the models except three (CMCC-CESM, GFDL-
ESM2M CESM1-CESM) are characterized by strong inter-basin correlation in both the SAZ and the AZ
(Fig. 4).  This suggests that the carbon cycle in these CMIP5 models is not sensitive to inter-basin
differences in the drivers as is the case for observations. This most likely implies that CMIP5 models
are not sensitive to regional $FCO_2$ variability at the basin scale, so $FCO_2$ seasonal biases are zonally
uniform.

Secondly, an important part of this analysis is based on the assumption that the observational
products that are used to constrain the spatial and temporal variability of $pCO_2$ and $FCO_2$ reflect the
correct seasonal cycles of the Southern Ocean.  This assumption requires significant caution not only
due to the limitations in the sparseness of the *in situ* observations but also due to limitations of the
empirical techniques in overcoming these data gaps (Landschutzer et al., 2014; Rödenbeck et al., 2015;
Gregor et al., 2017a, b; Ritter et al., 2018).  The uncertainty analysis from these studies suggests that,
while the seasonal bias in observations may be less in the SAZ and PFZ, it is the highest in the AZ
where access is limited mostly to summer, and winter ice cover results in uncertainties that may limit
the significance of the data-model comparisons.   It is important to note that though the observation
product that we use here (Landschützer et al., (2014) is based on more surface measurement (10
millions, SOCAT v3) compared to previous datasets (e.g. Tahakahashi et al., 2009, 3 millions), the data
are still sparse in time and space in the Southern Ocean. Thus, in using this data product as our main
observational estimates for this analysis we are mindful of the limitations in the discussion below.

Thirdly, the seasonal cycle of $\Delta pCO_2$ is the dominant mode of variability in $FCO_2$ (Mongwe et al., 2016;
Wanninkhof et al., 2009). Though winds provide the kinematic forcing for air-sea fluxes of $CO_2$ and
indirectly affect $FCO_2$ through mixed layer dynamics and associated biogeochemical responses
(Mahadevan et al., 2012; du Plessis et al., 2017), $\Delta pCO_2$ sets the direction of the flux. Surface $pCO_2$
changes are mainly driven by DIC and SST (Hauck et al., 2015; Takahashi et al., 1993). Subsequently
the sensitivity of CMIP5 models to how changes in DIC and SST regulate the seasonal cycle of $FCO_2$ is
fundamental to the model's ability to resolve the observed $FCO_2$ seasonal cycle. Thus, here we
examined the influence of DIC and SST on $FCO_2$ at seasonal scale for 10 CMIP5 models with respect to
observed estimates. Because temperature does not directly affect DIC changes, we first scaled up the
impact of SST changes on $pCO_2$ through surface $CO_2$ solubility to equivalent DIC units using the Revelle
factor (section 2.3). In this way, we can distinguish the influence of surface solubility and DIC changes
(i.e. biological and physical) on $pCO_2$ and hence on $FCO_2$.

Fourthly, using this analysis framework (sec 2.3, summarized in Fig. 6) we found that CMIP5 models
$FCO_2$ biases cluster in two groups, namely group-DIC ($M_{T\text{-}DIC}$ <0) and group-SST ($M_{T\text{-}DIC}$ > 0). Group-DIC
models are characterized by an overestimation of the influence of DIC on $pCO_2$ with respect to
observations estimates, which instead indicate that physical and biogeochemical changes in the DIC
concentration mostly regulate the seasonal cycle of $FCO_2$ (in short, DIC control). Group-SST models
show an excessive temperature influence on $pCO_2$; here surface $CO_2$ solubility biases are mainly
responsible for the departure of modelled $FCO_2$ from the observational products. While CMIP5 models
mostly show a singular dominant influence of these extremes, observations show a modest influence
of both, with a dominance of DIC changes as the main driver of seasonal $FCO_2$ variability. Below we
discuss the seasonal cycle characteristics and possible mechanisms for these two groups of CMIP5
models in the Sub-Antarctic and Antarctic Zones of the Southern Ocean.

## 608  4.1 Sub-Antarctic Zone (SAZ)


Our diagnostic analysis indicates that the seasonal cycle of $pCO_2$ in the observational product
(Landschützer et al., 2014) is mostly DIC controlled across all three basins of the SAZ ($M_{T\text{-}DIC}$ < 0 in Fig.
6). The Atlantic basin shows a stronger DIC control (Annual mean $M_{T\text{-}DIC} \geq 2$) compared to the Pacific
and Indian basin (Annual mean $M_{T\text{-}DIC} \approx 1$). This stronger influence of DIC on $pCO_2$ in the Atlantic basin
is consistent with higher primary production in this basin (Graham et al., 2015; Thomalla et al., 2011),
here shown by the larger mean seasonal chlorophyll from remote sensing in the Atlantic basin with
respect to the Pacific and Indian basin (Fig. 8). This significant basin difference is most likely linked to
the fact that the Atlantic basin has longer periods of shallow MLD compared to the Pacific and Indian
basins (Fig. 7a-c, Nov – Mar & Nov - Feb respectively) and has been shown to have higher supplies of
continental shelves and land-based iron (Boyd and Ellwood, 2010; Tagliabue et al., 2012; 2014). These
conditions are more likely to enhance primary production that translates into a higher rate of change
of surface DIC (Fig. 8), which becomes the major driver of $FCO_2$ variability. In contrast, shorter periods
of shallow MLD and lower iron inputs in the Pacific basin (Tagliabue et al., 2012), likely account for a
lower chlorophyll biomass and hence the weaker DIC control evidenced in our analysis ($M_{T\text{-}DIC} \approx 0$ in
Fig. 6). In the Indian basin, the winter mixed layer is deeper than in the Atlantic and deepens earlier in
the season (Fig. 7c). These conditions limit chlorophyll concentration (Fig. 8) and possibly contribute
to the lower rates of surface temperature change because of the enhanced mixing (cf Fig. 5a-c). As a
consequence, the resulting net driver in the Indian and Pacific basins is a weaker DIC control, because
both biological DIC and solubility changes are relatively weaker and they oppose each other. Because
of this, when the magnitudes of the rate of change of SST are larger during cooling and warming
seasonal peaks (autumn and spring respectively), DIC control is weaker ($M_{T\text{-}DIC} \approx 0$) during these
seasons.

CMIP5 models do not capture these basin-specific features as demonstrated with the correlation
analysis in Fig. 4, with the exception of three group-SST models (i.e. CESM1-BGC, GFDL-ESM2M and
CMCC-CESM). These, in contrast, mostly show comparable $FCO_2$ phasing in the three basins. The
seasonal cycle of $CO_2$ flux in the Southern Ocean (3,4) is both zonally and meridionally uniform for
most CMIP5 models, in contrast to observational data product (Fig. 3). This suggests that CMIP5
models show equal sensitivity to basin scale $FCO_2$ drivers, suggesting that $pCO_2$ and $FCO_2$ driving
mechanisms are less local than for observations. Thus the understanding of fine-scale (mesoscale and
sub-mesoscale) processes responsible for basin-scale $FCO_2$ variability will be an important
contribution to the next generation of ESM. Studies based on new available data from higher
resolution autonomous platforms like Monteiro et al., (2015), Williams et al., (2017). Briggs et al.,
(2018) and Rosso et al., (2017) may be useful constraints to these dynamics in ESMs.

The major feature of group-SST models in the SAZ is the out-gassing during summer and in-gassing
mid-autumn to winter  (Fig. 3a-c, Apr-Aug), which our diagnostics in Fig. 6 attribute to temperature
(solubility) control. The summer period coincides with the highest warming rates (dSST/dt, Fig 5a-c),
and associated reduction in solubility of $CO_2$. Similarly, exaggerated cooling rates at the onset of
autumn (Fig. 5a-c) enhance $CO_2$ solubility causing a change in the direction of $FCO_2$ into strengthening
$CO_2$ in-gassing (Fig 3a-c). Thus, while group-SST models have a seasonal amplitude of $FCO_2$
comparable to observations, they are out of phase (Fig. 3) as was the case in a previous analysis of a
forced ocean model (Mongwe et al., 2016).

In addition to increasing $CO_2$ solubility, the rapid cooling at the onset of autumn also deepens the MLD
(March-June, Fig. 7), which induces entrainment of DIC, increasing surface $CO_2$ concentration and
weakening the ocean-atmosphere gradient, and, in some instances, reversing the air-sea flux to out-
gassing (Lenton et al., 2013a; Mahadevan et al., 2011; Metzl et al., 2006). While these processes
(cooling and DIC entrainment) are likely to co-occur in the Southern Ocean, in CMIP5 models they are
characterized by their extremes: temperature impact of solubility exceeds the rate of entrainment
(Fig. 6 & 10). Because of the dominance of the solubility effect in group-SST models, the impact of DIC
entrainment on surface $pCO_2$ changes, the weakening of $CO_2$ in-gassing / out-gassing only happens in
mid-late winter (June-July -August) when entrainment fluxes peak (Fig. 10) and the SST rate
approaches zero (Fig. 5).

In the spring-summer transition, primary production is expected to enhance the net $CO_2$ uptake
(Thomalla et al., 2011; Le Quéré and Saltzman, 2013). However, the elevated surface warming rates
during spring reduces $CO_2$ solubility in group-SST models and overwhelms the role of primary
production in the seasonal cycle of $pCO_2$ and $FCO_2$ (atmospheric $CO_2$ uptake). As a consequence, these
group-SST models mostly show a constant or weakening net $CO_2$ uptake flux during spring in the
Pacific and Atlantic basin even though primary production is occurring and is relatively elevated (Fig.
3 & 8). Though some models show chlorophyll concentrations comparable to observations (e.g. GFDL-
ESM2M, CNRM-CM5, CanESM2), and sometimes greater (e.g. MRI-ESM), the impact of temperature-
driven solubility still dominates due to the phasing of the rates of the two drivers (Fig. 2a-c). The
Indian basin however shows the only exception to this phenomenon. Here, the amplitude of the
seasonal surface warming is relatively smaller ($\sim 0.5$ $^oC^{-1}$ month$^{-1}$ lower than the Pacific and Atlantic
basins), and the biologically-driven $CO_2$ uptake becomes notable and shows a net strengthening of the
sink of $CO_2$ during spring (Fig. 3c).

Though almost all analyzed CMIP5 models (with the exception of NorESM1-ME) exaggerate the
warming and cooling rates in autumn and spring, group-DIC models do not manifest the expected
temperature-driven solubility impact on $pCO_2$ and $FCO_2$ (Fig. 2). Instead, the seasonal cycle of $pCO_2$ and
$FCO_2$ are controlled by DIC changes, which are driven by an overestimated seasonal primary
production and the associated export carbon (Fig. 8). It is striking how in these models the seasonal
cycle of chlorophyll and $FCO_2$ are in phase (Fig 3a-c, 8a-c, with linear correlation coefficients always
larger than 0.9 not shown) but, as we discuss below, this is not because the temperature rates of
change are correctly scaled but because the biogeochemical process rates are exaggerated (Fig. 8).

Because of the particularly enhanced production in group-DIC models, the $CO_2$ sink is stronger (Fig. 8)
with respect to observation estimates during spring. This is visible in the reduction of surface DIC
(negative dDIC/dt in Fig. 8a, g-i), which can only be explained by drawdown due to the formation and
export of organic matter (Le Quéré and Saltzman, 2013). However, note that in the same way, after the
December production peak, both CMIP5 models and observations show an increase of surface DIC
concentrations (positive dDIC/dt) until March (Fig. 8, g-i). These DIC growth rates are particularly
enhanced in group-DIC models compared to some group-SST and observations (Fig. S9). The onset of
these DIC increases also coincides with the depletion of surface oxygen (Fig. S9), which we speculate is
due to the remineralization of organic matter to DIC through respiration. Unfortunately, only a few
models have stored the respiration rates, therefore the full reason for this DIC rebound remains to be
examined at a later stage. We would however tend to exclude other processes, because the onset of
$CO_2$ out-gassing seen in March in group-DIC models occurs prior to significant MLD deepening (Fig. 7)
and entrainment fluxes, therefore remineralization is likely be a key process here (Fig. 8).

## 702 4.2 Antarctic Zone (AZ)


The seasonal cycle framework summarized in Fig. 6 shows that the variability of $FCO_2$ and $pCO_2$ in the
Landschützer et al. (2014) product is characterized by a stronger DIC control (annual mean $M_{T-DIC} < -2$)
relative to the Sub-Antarctic ($M_{T-DIC} \approx -1$), except in the spring season ($M_{T-DIC} > -1$). This DIC control is
spatially uniform in the Antarctic Zone across all three basins (Fig. 4). The available datasets indicate
that the combination of weaker SST rates due to lower solar heating fluxes (Fig. 5), and stronger
shallower vertical DIC maxima (Fig. 10) favour a stronger DIC control through larger surface DIC rates.
The spatial uniformity in the seasonality of $FCO_2$ is also evident in the satellite chlorophyll and
calculated dDIC/dt from GLODAP2 in Fig. 9. Contrary to the Sub-Antarctic this might be suggesting
that $FCO_2$ mechanisms here are less local. It could be hypothesized that the seasonal extent of sea-ice,
deeper mixing and heat balance differences affect this region more uniformly compared to the Sub-
Antarctic Zone, and hence the mechanisms of $FCO_2$ are spatially homogeneous. However, we cannot
forget that sparseness of observations in this region is a key limitation to data products (Bakker et al.,
2014; Gregor et al., 2017; Monteiro et al., 2010; Rödenbeck et al., 2013) that might hamper the
emergence of basin-specific features. Consequently, this highlights the importance and need to
prioritize independent observations in the Southern Ocean south of the polar front and in the Marginal
Ice Zone. Increased observational efforts should also include a variety of platforms such as
autonomous vehicles like gliders (Monteiro et al., 2015) and biogeochemical floats (Johnson et al.,
2017) in addition to ongoing ship-based measurements.

In general terms, CMIP5 models are mostly in agreement (with an exception of MRI-ESM) with the
observational product on the dominant role of DIC to regulating the seasonal cycle of $FCO_2$ (Fig. 6d-f),
though not all models agree in the phase of the seasonal cycle of $FCO_2$ (e.g. CanESM2, Fig. 2). Though
CMIP5 models still mostly show the SST rates biases in autumn and spring with respect to observed
estimates, the stronger and near-surface vertical DIC maxima (Fig. 10), likely favor DIC as a dominant
driver of $FCO_2$ changes. Differences between group-SST and group-DIC models are only evident in mid-
summer when SST rates heighten and primary production peaks (Fig. 3 & 9). Probably because of sea
ice presence, the onset of SST warming is a month later (November) here in comparison to the Sub-
Antarctic (October). This subsequently allows the onset of primary production before the surface
warming, which then permits the biological $CO_2$ uptake to be notable in group-SST models. Thus the
two model groups here agree in the $FCO_2$ in-gassing during spring with group-SST models being the
closest to the observational product. The MRI-ESM is the only model showing anomalous solubility
dominance during autumn and spring as in the Sub-Antarctic Zone.

This coherence of CMIP5 models and observations in the Antarctic Zone may suggest that CMIP5
models compare better to observations in this region (Fig. 4). However, because CMIP5 models also
show this spatial homogeneity in the Sub-Antarctic Zone (contrary to observational estimates), it is
not clear whether this indicates an improved skill in CMIP5 model to the mechanisms of $FCO_2$ in this
region, or both CMIP5 models and observational product lacks spatial sensitivity to the drivers of
$FCO_2$. The sparseness of observations in the AZ points to the latter.

The cause of differences in the seasonal rates of SST change in group-SST models remains a subject of
ongoing research. The Southern Ocean is a part of the global ocean (upwelling) where earth systems
models show a persistent warming  SST bias (Hirahara et al., 2014). Several studies point to highlight
potential explanations but the main reasons remains uncertain.   For example, CMIP5 models
differences in the magnitude and meridional location of the peak of wind speeds in the Southern Ocean
(Bracegirdle et al., 2013) and MLD differences (Meijers, 2014; Sallée et al., 2013) may be such that the
net effect of change on surface turbulence and mixing leads to these amplified surface temperature
rates. Other known CMIP5 modells' biases thatwhich may contribute includes; heat fluxes and storage
(Frölicher et al., 2015) as well as sea-ice dynamics (Turner et al., 2013). Notwithstanding these,
investigation of the reasons for sources of these dSST/dt biases is out of the scope of this study. Our
aim here is to show that understanding biases in the drivers of $pCO_2$ (DIC and SST) at the seasonal
scale is necessary to understand differences in the seasonal cycle of $FCO_2$ between models and
observational products. However we recommend that the mechanistic basis for the differences the
seasonal rates of warming and cooling be a matter of urgent investigated further

758 .





# 5. Synthesis

We used a seasonal cycle framework to highlight and examine two major biases in respect of $pCO_2$ and
$FCO_2$ in 10 CMIP5 models in the Southern Ocean.

Firstly, we examined the general exaggeration of the seasonal rates of change of SST in autumn and
spring seasons during peak cooling and warming respectively with respect to available observations.
These elevated rates of SST change tip the control of the seasonal cycle of $pCO_2$ and $FCO_2$ towards SST
from DIC and result in a divergence between the observed and modelled seasonal cycles, particularly
in the Sub-Antarctic Zone. While almost all analyzed models (9 of 10) show these SST-driven biases, 3
of the 10 (namely NorESM1-ME, HadGEM-ES and MPI-ESM) don't show these solubility biases because
of their overly exaggerated primary production (and remineralization) rates such that biologically-
driven DIC changes mainly regulate the seasonal cycle of $FCO_2$. These models reproduce the observed
phasing of $FCO_2$ as a result of an incorrect scaling of the biogeochemical fluxes. In the Antarctic Zone,
CMIP5 models compare better with observations relative to the Sub-Antarctic Zone. This is mostly
because both CMIP5 models and observational product estimates show a spatial and temporal
uniformity in the characteristics of $FCO_2$ in the Antarctic Zone. However, it is not certain if this is
because model process dynamics perform better in this high latitude zone or that the observational
products variability is itself limited by the lack of *in situ data.* This remains an open question that
needs to be explored further and highlights the need for increased scale-sensitive and independent
observations south of the Polar Front and into the sea-ice zone.

The second major bias is that contrary to observational products estimates, CMIP5 models generally
show an equal sensitivity to basin scale $FCO_2$ drivers (except for CMCC-ESM, GFDL-ESM2M and
CESM1-BGC) and hence the seasonal cycle of $FCO_2$ has similar phasing in all three basins of the Sub-
Antarctic Zone. This is in contrast to observational and remote sensing products that highlight strong
seasonal and interannually varying basin contrasts in both $pCO_2$ and phytoplankton biomass. It is not
clear if this is due to inadequate carbon process parameterization or improper representation of the
dynamics of the physics.  This should be investigated further with CMIP6 models and our analysis
framework is proposed as a useful tool to diagnose the dominant drivers.   Contrary to observed
estimates, CMIP5 models simulate $FCO_2$ seasonal dynamics that are zonally homogeneous and we
suggest that any investigation of local (basin-scale) mechanisms, dynamics and long term trends of
$FCO_2$ using CMIP5 models must remain tentative and should be treated with caution. This highlights a
key area of development for the next generation of models such those planned to be used for CMIP6.



## Acknowledgements


This work was undertaken with financial support from the following South African institutions: CSIR
Parliamentary Grant, National Research Foundation (NRF SANAP programme), Department of Science
and Technology South Africa (DST), and the Applied Centre for Climate and Earth Systems Science
(ACCESS). We thank the CSIR Centre for High Performance Computing (CHPC) for providing the
resources for doing this analysis. We also want to thank Peter Landschützer, Taro Takahashi and Luke
Gregor for making their data products available as well as the three reviewers for their productive
comments that we think have strengthened the paper

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

L184

L185

L186

L187

## Figures

L189

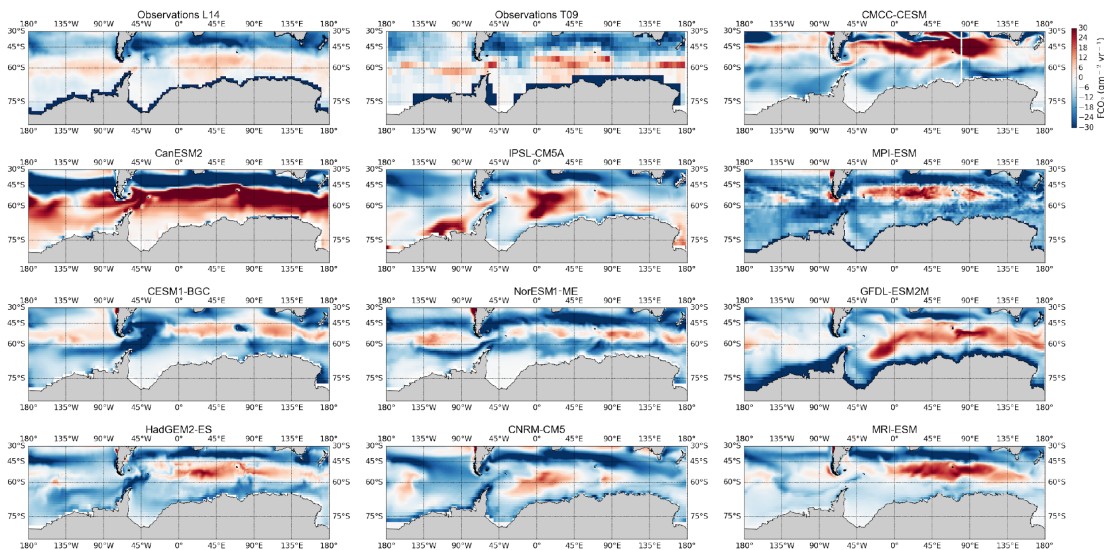

L190

**Fig. 1**: The annual mean climatological distribution Sea-Air $CO_2$ Flux ($FCO_2$, in gC m$^{-2}$ yr$^{-1}$) for observations
(L14: Landschützer et al., 2014 and T09: Takahashi et al., 2009) and 10 CMIP5 models over 1995 – 2005.
CMIP5 models broadly capture the spatial distribution of FCO2 with respect to L14 and T09, however, they
also show significant differences in space and magnitude between the basins of the Southern Ocean with a
few exceptions.

L196

L197

L198

L199

L200

L201

L202

L203

L204

L205

L206

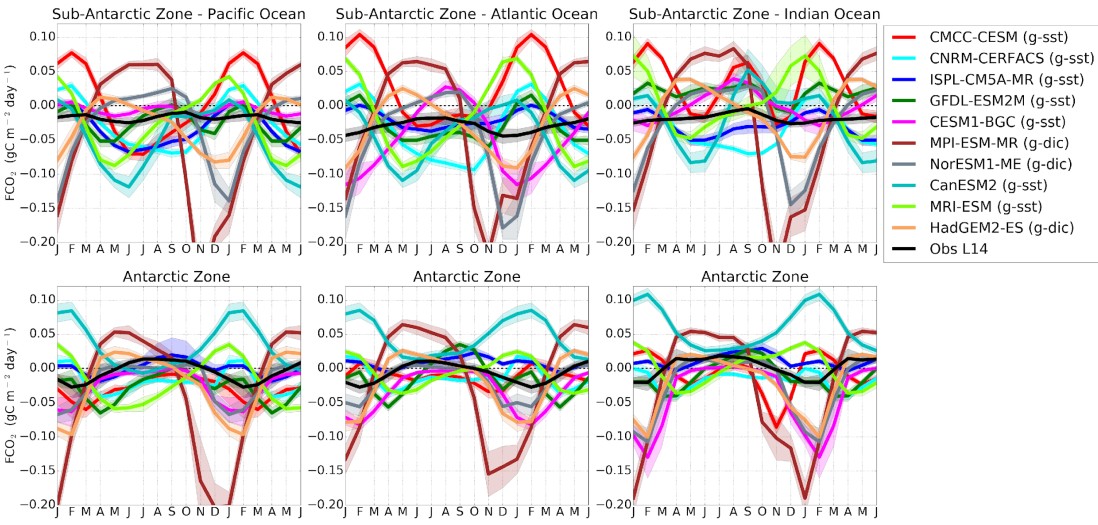

**Fig. 2**: Seasonal cycle of Sea-Air $CO_2$ Flux ($FCO_2$, in $gC\ m^{-2}\ yr^{-1}$) in observations and 10 CMIP5 models in the Sub-Antarctic and Antarctic zones of the Pacific Ocean (first column), Atlantic Ocean (second column) and Indian Ocean (third column). The shaded area shows the temporal standard deviation over the considered period (1995 – 2005), g-sst and g-dic shows the clustering of CMIP5 models into group-SST and group-DIC as shown in Fig. 3 (section 3.2).

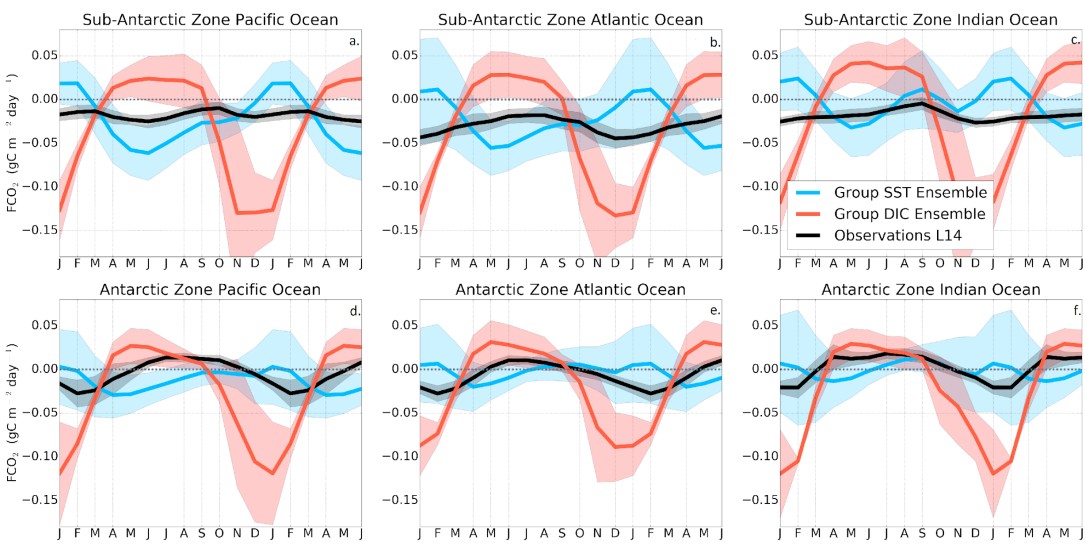

**Fig. 3.** Seasonal cycle of the equally-weighted ensemble means of FCO$_2$ (gC m$^{-2}$ yr$^{-1}$) from Fig. 2 for group

DIC models (MPI-ESM, HadGEM-ES and NorESM) and group SST models (GFDL-ESM2M, CMCC-CESM,

CNRM-CERFACS, IPSL-CM5A-MR, CESM1-BGC, NorESM2, MRI-ESM and CanESM2). The shaded areas show

the ensemble standard deviation. The black line is the Landschützer et al. (2014) observations.

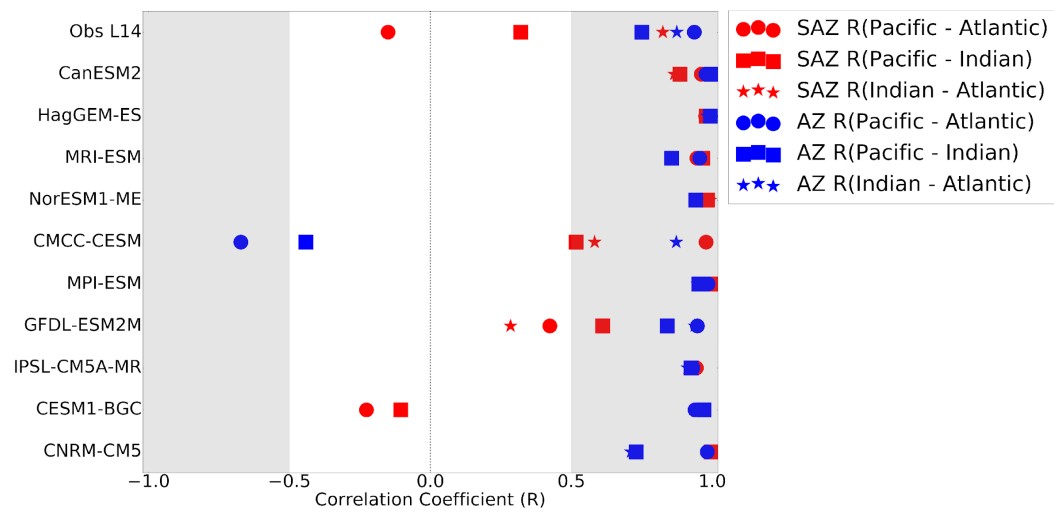

**Fig. 4**: The correlation coefficients (R) of basin – basin seasonal cycles of $FCO_2$ for observations (Landschützer et al., 2014) and 10 CMIP5 models in the three basins of the Southern Ocean i.e. Pacific, Atlantic and Indian basin.

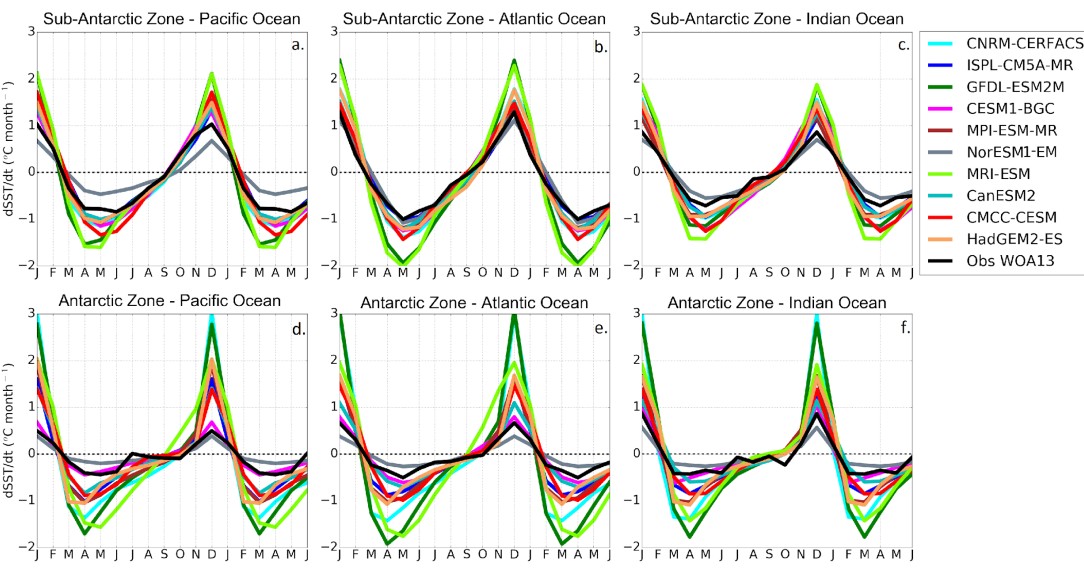

**Fig. 5**: Mean seasonal cycle of the estimated rate of change of sea-surface temperature (dSST/dt, $^{o}$C month$^{-1}$) for the Sub-Antarctic and Antarctic zones of the Pacific Ocean (first column), Atlantic Ocean (second column) and Indian Ocean (third column).

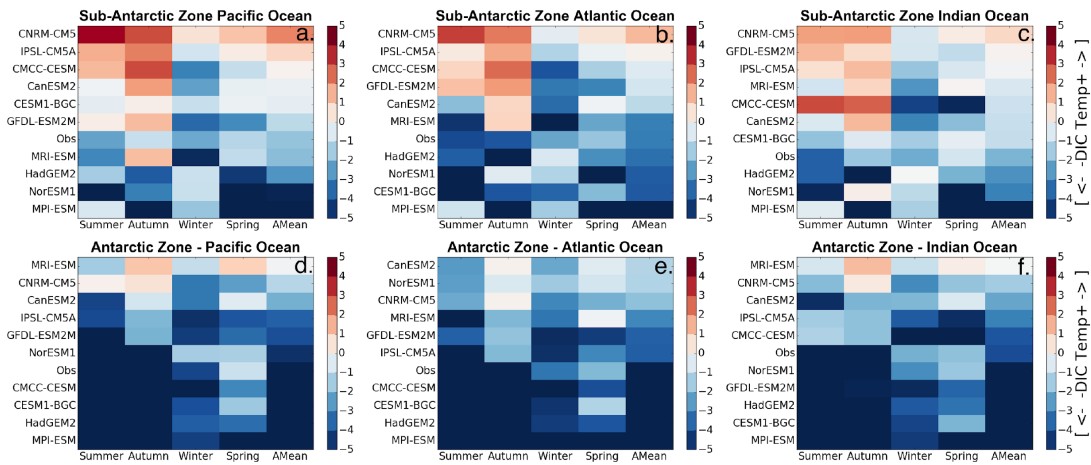

**Fig. 6**: Mean seasonal and annual values of the DIC–temperature control index ($M_{T\text{-}DIC}$). The increase in the red color intensity indicates increase in the strength of the temperature driver and the blue intensity shows the strength of the DIC driver. The models are sorted according to the annual mean value of the indicator presented in the last column (Amean).

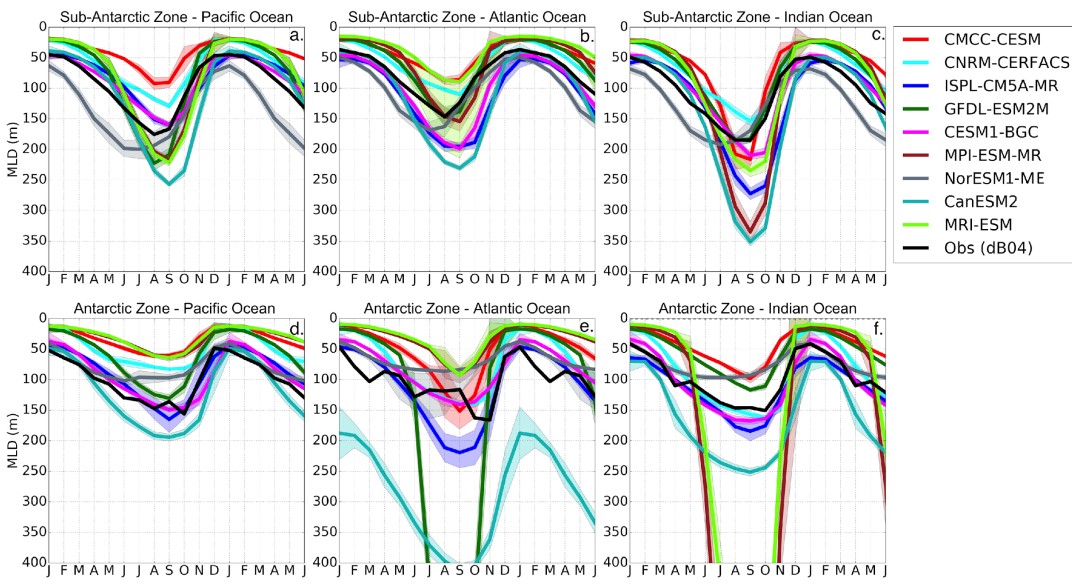

**Fig. 7**: Seasonal cycle of the Mixed Layer Depth (MLD) in the Sub-Antarctic and Antarctic zones of the Pacific Ocean (first column), Atlantic Ocean (second column) and Indian Ocean (third column).

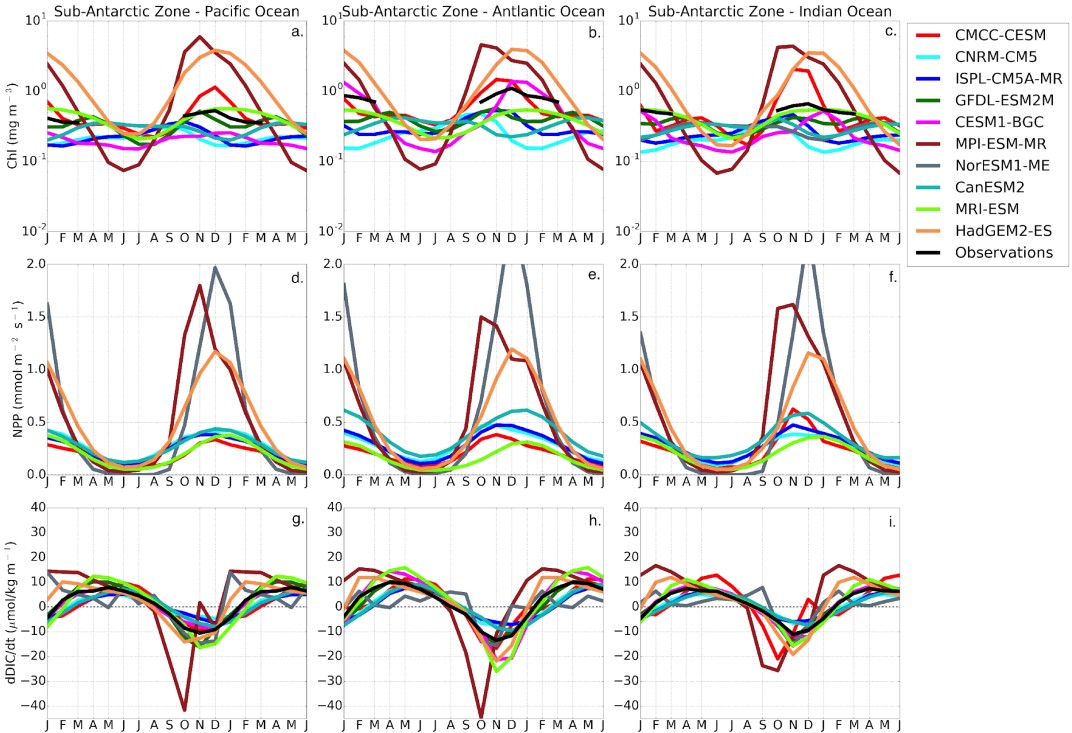

316

**Fig. 8**: The seasonal cycle of chlorophyll (mg m$^{-3}$), Net Primary Production (mmol m$^{-2}$ s$^{-1}$) and the surface

rate of change of DIC (μmol kg$^{-1}$ month$^{-1}$) in the Sub-Antarctic zone of the Pacific Ocean (first column),

Atlantic Ocean (second column) and Indian Ocean (third column).

320

321

322

323

324

325

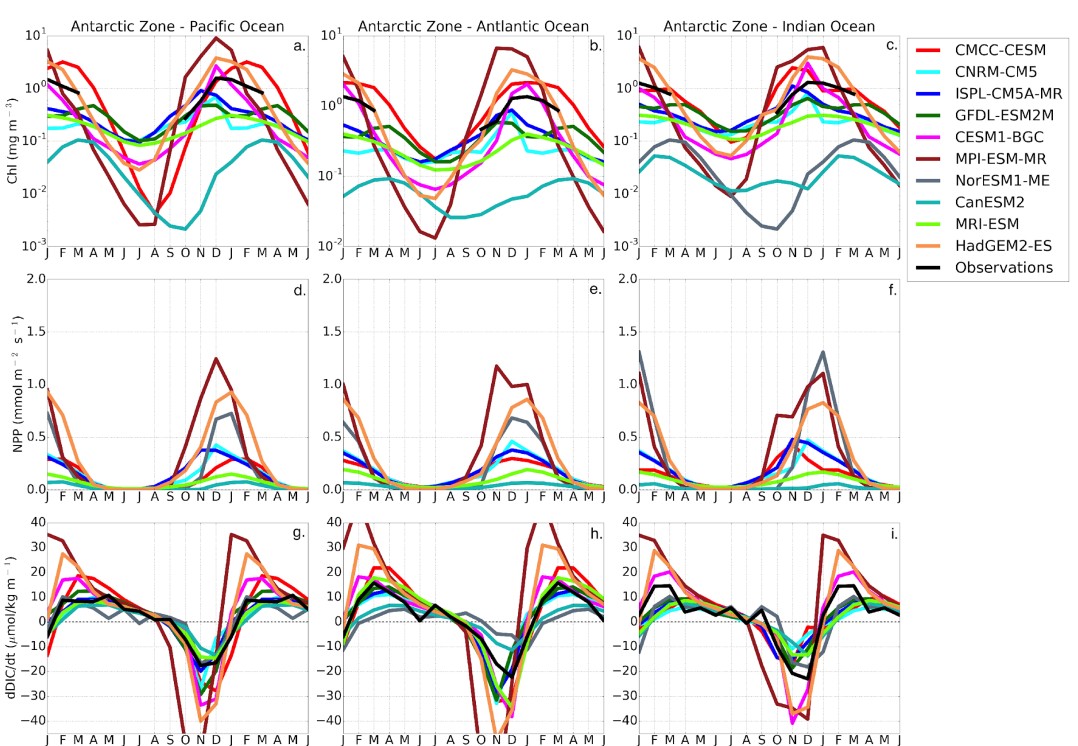

L326

**Fig. 9** Same as Fig. 8 for the Antarctic zone.

L327

L328

L329

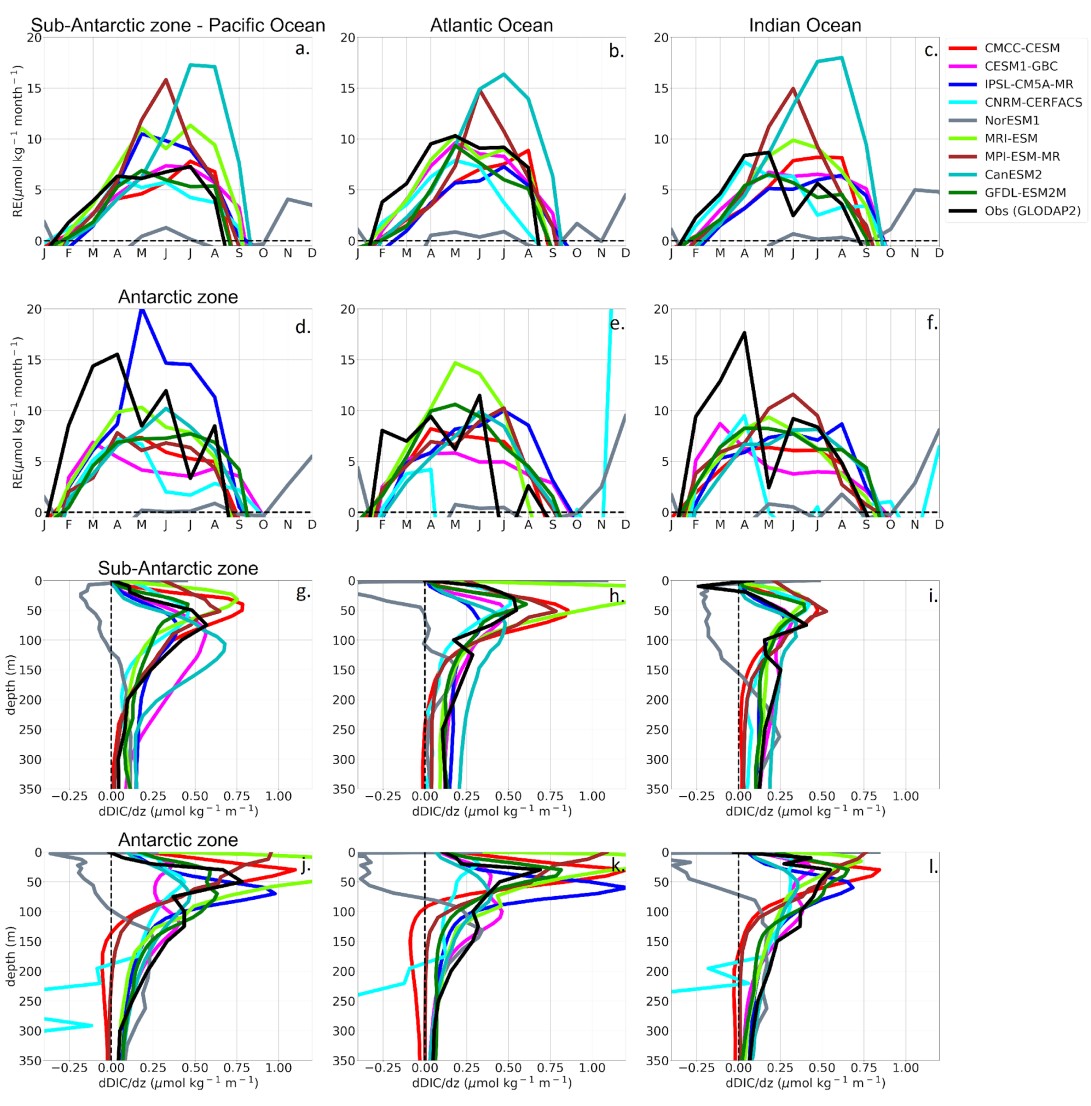

**Fig. 10**: (a-f) Estimated DIC entrainment fluxes (mol kg month⁻¹) at the base of the mixed layer and (g-i) vertical DIC gradients (μmol kg⁻¹ m⁻¹) in the Sub-Antarctic and Antarctic zone of the Pacific Ocean (first column), Atlantic Ocean (second column) and Indian Ocean (third column).

⌊342   **Table 2**: Sea-Air CO₂ fluxes (Pg C yr⁻¹) annual mean uptake in the Southern Ocean (first column), here

⌊343   defined as south of the Sub-tropical front, Sub-Antarctic zone (second column) and Antarctic zone (third

⌊344   column). The third and fourth column shows the Pattern Correlation Coefficient (PCC) and Root Mean

⌊345   Square Error (RMSE) for the whole Southern Ocean for each model. Observations here refer to

⌊346   Landschützer et al., 2014.

Table 2: Sea-Air CO$_2$ Fluxes Mean Annual Uptake, PCC and RMSE

| Model | Southern Ocean | Sub-Antarctic zone | Antarctic zone | PCC | RMSE |
|---|---|---|---|---|---|
| CNRM-CM5 | -0.823 ± 0.003 | -0.682 ± 0.002 | -0.122 ± 0.001 | 0.44 | 17.9 |
| GFDL-ESM2M | -0.161 ± 0.005 | -0.074 ± 0.004 | -0.077 ± 0.002 | 0.43 | 8.47 |
| HadGEM2-ES | -0.489 ± 0.005 | -0.284 ± 0.003 | -0.197 ± 0.001 | 0.55 | 10.9 |
| IPSL-CM5A-MR | -0.496 ± 0.003 | -0.582 ± 0.006 | 0.101 ± 0.003 | 0.53 | 10.5 |
| MPI-ESM-MR | -0.870 ± 0.006 | -0.530 ± 0.002 | -0.326 ± 0.002 | 0.37 | 9.87 |
| MRI-ESM | -0.048 ± 0.002 | 0.022 ± 0.003 | -0.070 ± 0.001 | 0.36 | 15.6 |
| NorESM1 | -0.699 ± 0.004 | -0.412 ± 0.003 | -0.270 ± 0.002 | 0.60 | 8.96 |
| CESM1-BGC | -0.532 ± 0.006 | -0.132 ± 0.003 | -0.385 ± 0.004 | 0.47 | 9.15 |
| CMCC-CESM | 0.121 ± 0.006 | 0.367 ± 0.004 | -0.225 ± 0.003 | -0.09 | 17.9 |
| CanESM2 | -0.058 ± 0.008 | -0.720 ± 0.006 | 0.661 ± 0.004 | 0.54 | 19.5 |
| Observations | -0.253 ± 0.3 | -0.296 ± 0.3 | 0.053 ± 0.3 | | |

⌊347