# Peer review of "The Seasonal Cycle of pCO2 and CO2 fluxes in the Southern Ocean: Diagnosing"

_Biogeosciences, 2017_

## Referee Comment (RC1) · Anonymous Referee #1 · 27 Oct 2017

The Manuscript "Mechanisms of the Sea-Air CO2 Flux Seasonal Cycle biases in CMIP5 Earth Systems Models in the Southern Ocean" by Precious et al provides an empirical partitioning of flux between correlations with sea surface temperature change (presumably due to solubility) and influences DIC concentration including biological, wind, circulation terms in determining air-sea fluxes in the Southern Ocean from CMIP5 models. The authors find that the CMIP5 generation carbon cycle models can be classified into two groups – one whose phasing of the seasonal cycle agrees with an observationally constrained estimate but is overly strong, and another that has the opposite phasing. The manuscript is thus helpful in this empirical anecdote in spurring further assessment of the mechanistic commonalities and differences between there models.

[Figure]

The manuscript is also valuable in contrasting the representation of Antarctic and Subantactic zones in these models, and there relative zonal symmetry between the three basins compared to the observationally constrained analysis.

Unfortunately, I find the title beginning "Mechanisms..." to be a disappointing overreach as rather than including quantification of the solubility, transport, and biological pump mechanisms, the authors rely entirely on the empirical seasonal relationship correlation of dSST/dt and cursory analysis of mixed layer entrainment as metrics of model mechanisms. Throughout the manuscript, the authors assert that their SST correlation demonstrates that the seasonal cycle "is determined by the role of temperature". Correlation isn't causation... SST correlates with many other things – including mixed layer depth and productivity, the mechanistic significance of correlations with SST is complex. For example, the authors could calculate the change in equilibrium DIC under temperature change and convert their dSST/dt metric into a dDIC_solubility/dt metric to quantitatively assess the role of solubility. Of course, this would involve the assumption of infinite wind and neglect the year-timescale of DIC equilibration, and including the role of wind in modulating the timing of CO2 flux would complicate the authors simplistic interpretation. Beyond that, interior budgets could be constructed, leading the authors to developing a simple box model of mixed layer DIC to be able to reproduce the various GCM results through the combination of gas exchange, thermal, transport, and biological mechanisms. While such a more mechanistically based box model analysis could prove very valuable in uncovering the mechanistic differences between the models, it is probably outside the scope of the present manuscript.

The current reliance on qualitative correlation leaves the analysis with a modest utility in describing basic model results as a repackaging of previously published analyses, but of marginal utility in providing novel insight into the causes of model CO2 flux seasonality and implications for future Southern Ocean carbon uptake. At a minimum, I suggest the authors quantify the role of SST change on DIC solubility to be able to confidently assess whether the role of temperature in the temperature correlated

group can be attributed to solubility, Beyond that, the authors should include some quantification of model biases relative to observational uncertainty, and include several of the available observationally constrained products to assess that uncertainty.

Specific comments

14 – "has" should be "may" here as this is an assertion/hypothesis, not a conclusion

16 – Why "specialized" suggest removing or replacing with a more specific word.

24 – I think "with a dominance of DIC regulation" should be "with a dominance of other factors driving DIC regulation"

28 - "resolve" which way, higher or lower?

37-38 – What latitude criterion is used for this "third" estimate?

43-44 – "Evolution of the Southern Ocean CO2 sink is expected to change..." is confusing as evolution is already a change, and doesn't specify how – suggest something like "The Southern Ocean CO2 sink is expected to diminish under anthropogenic warming"

66 – Remove "from September,"

82 - Two very relevant additional manuscripts discussion these term balances include:

Nevison, C.D., Manizza, M., Keeling, R.F., Stephens, B.B., Bent, J.D., Dunne, J., Ilyina, T., Long, M., Resplandy, L., Tjiputra, J. and Yukimoto, S., 2016. Evaluating CMIP5 ocean biogeochemistry and Southern Ocean carbon uptake using atmospheric potential oxygen: Present‐day performance and future projection. Geophysical Research Letters, 43(5), pp.2077-2085.

Jiang, C., Gille, S.T., Sprintall, J. and Sweeney, C., 2014. Drake Passage Oceanic p CO2: Evaluating CMIP5 Coupled Carbon–Climate Models Using in situ Observations. Journal of Climate, 27(1), pp.76-100.

135 – The assertion that "The ocean-atmosphere CO2 gradient is known to be the main driver of FCO2 variability" Is true in a regional sense but is certainly not try in a temporal sense in most regions where wind variability can dominate like in the equatorial Pacific:

http://onlinelibrary.wiley.com/doi/10.1029/2005JC003129/full

The delta pCO2 argument was that is you average over large enough scales, the mixed layer equilibration time of CO2 was short enough (about a year) that CO2 fluxes were determined by the net balance of biolgy and thermal factors rather than the wind. On a seasonal scale, ignoring the role of wind seems like a fatal flaw. Rather, the authors should argue that the wind variability in this region is small before disregarding it. This is likely true in the Southern Ocean where winds are strong in all seasons.

181 – While I am glad the authors are considering mixed layer entrainment, it seems remiss here to ignore the biological and other circulation terms such as upwelling and consider them all lumped together as "DIC" terms.

184 – A brief description of the Orsi definition should be provided here.

249 – The statement that the models "do not capture any of the basin-specific features" is a fairly strong, but non-quantitative statement. This should be much more specific – like, the observational reanalysis shows a stronger flux in the Atlantic than the Indian and Pacific while the models show similar fluxes in each basin.

255, 267, 269 –in all three cases, I ask the same question, "How much". It is not sufficient to anecdotally say that there are "some differences", "an overestimation", "large standard deviation" – all of these are relative statements that mean nothing if not quantified. The major question is if they are large relative to the uncertainty in the observations.

256 – Please quantify the time span – 2 months, 3 months?

259 – "mid-summer (Dec-Feb)" does not make sense as Dec-Feb is the entire summer... what is the time span?

260 – "end of autumn (March)" does not make sense as March is the beginning of Autumn.

278 – I do not understand the phrase, "the overall model biases are not consistent with the seasons"

282 –again, "less differences" and "small negative biases" should be quantified to be meaningful.

319 – "justifies our a priori separation" comes across as inappropriate self-congratulation. The salient point is that the separation quantifies the separation between the two classes of models in terms of the relative dominance of SST-Flux correlation.

370-371 – Unless the authors are going to quantitatively explore the mechanisms of SST drivers on CO2 flux including solubility, phytoplankton growth rates, etc, the statement "the seasonal cycle of FCO2 is determined by the role of temperature" should rather be "the seasonal cycle of FCO2 is correlated with temperature"

380-391 – This paragraph pertaining to the role of solubility is missing quantification of the role of solubility changes. The simplest

---

## Referee Comment (RC2) · Anonymous Referee #2 · 31 Oct 2017

I think there is the nucleus of a solid paper here, but major revision of the present manuscript is required. The English is generally good although there are some quirks of usage. The Discussion/Conclusion is somewhat repetitive; I would consider merging Discussion and Conclusions.

Major points (1) I think the methodology could be better explained. Perhaps the logic of equations 1-3 is explained more thoroughly in the previous paper by Mongwe et al 2016, but this critical reference is missing from the reference list. Assuming they mean the paper in Ocean Modelling 106: 90, I will agree that equations 1 and 2 can be derived from

equations 3.2-3.4 in that paper. But the LHS of equation 2, which does not appear in the previous work, is physically speaking, a fairly nebulous quantity. The Takahashi et al 1993 estimate of 0.0423 Kˆ-1 mainly expresses the change in partial pressure due to changing temperature for a given concentration of CO2 ([CO2*]), with a small contribution from the partitioning of DIC among CO2/HCO3-/CO3– due to the temperature dependence of the equilibrium constants. DIC does not change as a result of changes in temperature, except indirectly through gas exchange.

So what we have here is an observed change of pCO2 with changing temperature, convoluted into a change of DIC by application of several highly empirical conversion factors (more about this below), as an estimate of the changes in DIC not attributable to biological uptake/remineralization (and therefore primarily attributable to gas exchange). This in itself might be inoffensive, but I would prefer if its relationship to actual physical processes were better explained. The equations that are taken as a starting point are highly empirical, and we should not invest rearrangements of these with an outsized significance.

0.0423 Kˆ-1 is intended to be an average value for a broad range of ocean conditions, but it is stated to be valid for salinities from 34-36 and temperatures from 2-28 C (Takahashi et al 1993). In the Southern Ocean one will encounter conditions outside, or on the far edges of, these ranges. What are the implications of this for the analysis shown here? This seems like something that could be evaluated. Similarly, the calculations assume a constant Revelle factor, but it should be quite straightforward to calculate Revelle factors from the model outputs, giving a range for the range of environmental conditions characteristic of the study area. The conclusions are probably robust to these assumptions, but I see no reason why this can not be tested.

Finally, isn't the total DIC variability by definition the sum of the various components? So I'm not clear why the temperature driven component should ever be larger than the total. I find equation 3 and the discussion on 304-311 to be the most confusing part. We have an observed rate of change of DIC (which is never actually defined), which one would think would be the sum of the contributions from gas exchange, biological uptake/remineralization and entrainment. But in this case, the index that is considered is that the total is either greater or less than one of these three components (whose physical meaning is nebulous). To confuse matters worse, we have a reference to "the total DIC seasonal cycle (dDIC/dt)" (306-307). Doesn't dX/dt imply an instantaneous rate-of-change that will itself vary over the annual cycle? I really do not understand what is being asserted here. (Also, the text should say something about exactly what sort of discretization was used in calculation of trends, e.g., does delta-X/delta-t for November represent a value for Nov. 1 based on a difference of October and November means, or is it something else? If this is the case, figure axes should indicate that calculated values are for the first of the month and not the mid-month.)

The discussion of entrainment is also confusing and poorly connected to actual physical processes. Equation 4 does not have the units of a flux, but rather of a rate of change within the surface layer. The proper quantity here is not DIC concentration at MLD(T+1) but rather the difference between DIC at MLD(T+1) and at MLD(T). A simple example: we have an 80 m mixed layer that deepens to 100 m over a 24 h period, with an initial linear gradient of DIC concentration from 2.0 mol mˆ-3 at 80 m to 2.1 at 100 m. The total amount of DIC entrained is 0.5 * (20 m) * (100 mmol mˆ-3), for a net vertical flux of 1000 mmol mˆ-2 dˆ-1 or a surface rate of change of 10 mmol mˆ-3 dˆ-1 based on the 'new' MLD as per equation 4. According to equations 4 and 6 in this paper (5 is redundant and unnecessary), this rate would be 210 mmol mˆ-3 dˆ-1.

(2) There is no discussion of the effects of SST and and SSS on CO2 solubility, other than changes in temperature arising from air-sea heat flux. When the mixed layer deepens in autumn, changes in surface T+S caused by entrainment of subsurface water could affect solubility. This may be a second order effect, but it wouldn't hurt to discuss it, and at times the text is ambiguous as to what exactly is being considered. For example, on 395-396 it is stated that "While surface cooling strengthens CO2 solubility in autumn, the concurrent MLD deepening has an opposing effect". So at first it appears that what is being referred to is the effect of entrainment on solubility via T+S. But when you read a bit further it seems that they are actually talking about DIC entrainment, which of course has no effect on solubility. I think entrainment effects on solubility via SST+SSS should be discussed. If the authors do not wish to address this topic, at least they should modify this passage to make clear that this effect was not considered.

(3) Similarly, the authors could discuss bias in alkalinity estimates as a source of potential error. Upwelled deep waters contain excess alkalinity as a result of accumulated dissolution of biogenic carbonates. Whenever deeper waters are entrained, the Lee formula will generally underestimate the alkalinity. This bias may be small but it is systematic and should be evaluated. (The authors should list in the text the regression coefficients from Figure S1 as well as the correlation coefficient. It is possible to have a very strong correlation but a large systematic bias. In this case the bias is small, and this should be stated in the main text.)

(4) The use of chlorophyll as a proxy is not really explained, when primary production and export production are generally available as model output fields. One might justify this by saying that observations are available only for chlorophyll, but this should be stated explicitly. There are also observation-based estimates of primary production available (see below Terminology).

(5) I would like to see some discussion of the possibility that the apparently greater temperature control in the Pacific sector (259-263) is a real effect that arises from iron limitation. Because terrestrial sources of iron are much greater in Atlantic sector and the western half of the Indian sector (see e.g., Graham et al 2015 DSR I 104: 9; Tagliabue et al., 2012 Biogeosciences 9: 2333), it seems logical that the effect of seasonal biological drawdown on pCO2 would be greater than in the Pacific and in the eastern half of the Indian sector. These regions also overlap the regions where the wind speed and the amplitude of its annual cycle are greatest (e.g., Trenberth et al., 1990, JPO 20: 1742), which will also tend to reduce the influence of biological uptake relative to temperature. Note also that the wind speed peaks in the spring and fall transition periods, particularly over the Pacific sector (Trenberth op cit their Fig. 4).

Some more questions about methodology

Why use only ten years of model output (124)? The results could be biased by internal variability; the more usual averaging period would be 20 or even 30 years. With a reference year of 2000, this would require using emissions scenarios, which is perhaps a reason not to do it, but the differences among scenarios are very small in 2005-2015 (because the scenarios are constructed precisely around the assumption that there is some inertia in human societies and abrupt changes are unlikely). I think the authors should (a) pick one model, recalculate the results for 1990-2010 and 1985-2015, and estimate the potential error associated with aliasing of internal variability. And (b) if this error turns out to be large, repeat the calculation for the full suite of models.

Why use GLODAP1 data for DIC? GLODAP2 has been available for almost two years. There are also more up to date data sets available for MLD. ARGO has made the Southern Ocean a much less undersampled region than it was in 2004. (The abbreviation MLD is not defined at first use (117)).

Some stylistic advice

Sir Peter Medawar is said to have advised authors of scientific papers to never begin with "a resounding banality". We all do this. "CO2 is one of the most important greenhouse gases." "The ocean is a critical component of the climate system." But it's actually good advice. The abstract to this paper starts weak, but ends even weaker. The final sentence is very poorly worded and bordering on incomprehensible. What is in between is generally good. But if we want people to read the paper, we have to end the abstract in a way that generates interest.

Similarly, the main text (Conclusion) ends by saying that "the inability of the CMIP5 ESMs to resolve CO2 biological uptake during spring might be crucially related to the sensitivity of the pCO2 to temperature". This again leaves the reader at a loss as to what exactly the authors are trying to say, at a critical point. Is the problem here model errors in SST, or the way that solubility is calculated from T+S? Almost certainly the former, but it's hard to tell from the way this is phrased.

Terminology

I think the authors should acknowledge that the FCO2 data product is not really 'observed' in the sense that pCO2 is. I think they should compare modelled and observed pCO2, and then discuss what this means for modelled estimates of CO2 flux, without referring to the Landschuter FCO2 estimates as observations. CO2 fluxes in models and data products like this are actually quite different conceptually. When you estimate CO2 flux from observed pCO2, the errors in the flux are a linear function of errors in wind speed (or u^2, assuming a quadratic parameterization) and the piston velocity. In numerical models, pCO2 and DIC self-regulate to dampen these errors when monthly averaged fluxes are considered, e.g., if both the wind speed and the DIC are too large, the enhanced outgassing flux will reduce the pCO2 and DIC error. Higher wind speed or piston velocity will tend to drive pCO2 towards atmospheric, and not necessarily towards the 'correct' value if over- or undersaturation exists, so there is no straightforward way to correct for this difference. But I think that the authors should acknowledge that it exists, and that in comparing modelled and 'observed' fluxes they are to some degree comparing apples and oranges. (With regard to point (4) above, if there is no observed primary production, there is no observed CO2 flux either: both of these are extrapolated from the primary observed field using models of unknown accuracy.)

I think it is stretch to call the Takahashi et al (2009) flux estimates a 'dataset' (103). I think it would be more accurate to refer to CO2 flux estimates of Takahashi et al, or CO2 flux estimated by the methods of Takahashi et al (as do Takahashi et al themselves). Note that the description of this data product (104-106) refers only to pCO2 measurements and says nothing about how flux was estimated.

Don't refer to climate model projections as predictions (15, 85, 496). Similarly, I would remove the term "mode" (60, 488) as the intended meaning deviates from the usual meaning of this term in climate research. (Rule of thumb for young authors: don't coin neologisms without some compelling reason, and don't appropriate existing terms and give them new meanings.)

Similarly, the intended meaning of the word "phase" is unclear, and the usage of this term is not usual. (The same is true of "coherent": see below 252, 339.) This begins in the Introduction, where it is stated that the models "disagree on the seasonal cycle of CO2 flux and they are out of phase with observations" (58). Does this mean the phasing of the seasonal cycle in the models differs from that in the observations? This would seem to be the most plausible explanation, but the present wording does not communicate this (or anything else) effectively. The later invocations (e.g., 231-232, 260-266) are better but could still benefit from the being a bit more explicit about exactly what is being discussed. Instead of referring to the models and observations being out of phase with each other, one could refer to actual physical processes: e.g., that the modelled seasonal cycle of SST is out of phase with that in the WOA climatology. Assuming, of course, that this is actually an accurate characterization. Or is what is meant here simply that the models do not reproduce the phase of the annual cycle accurately? When we say that two cycles are out of phase, we usually mean offset by 180 degrees.

Similarly, "the onset of primary production" could be better phrased. Primary production is not zero in the winter, although clearly seasonality is large in high-latitude environments. I would probably say something like "the increase in biological uptake in spring". More importantly, primary production per se has little or no effect on DIC: in principle it could be entirely respired within the surface layer. It is true that net community production is usually positively created with primary production, especially in highly seasonal high latitude environments. But since the focus of this paper is DIC and pCO2 I don't see what is gained by referring to primary production rather than simply biological uptake (e.g., 482).

Details

"ingassing" is variously spelled as ingassing, in gassing, in-gassing (e.g., 62, 64, 205)

Model names appear to be misspelled in places (e.g., 203, 233, 283, 319, 321, 401, 431, 466, Figures 3, 4, and 6)

remove the phrase "used a specialized diagnostic analysis"

change "contradicting" to "counteracting"

change "oceanic" to "ocean"

43-44 "The century scale evolution of the Southern Ocean CO2 sink is expected to change as a result of anthropogenic warming, however the anticipated change is still disputed." The century scale evolution of the Southern Ocean CO2 sink is expected to change as a result of anthropogenic warming, however, the sign and magnitude of the change is still disputed.

"especially during the winter season" especially during winter missing "Ref"

62-63 "with a weaker in-gassing or even outgassing state in winter" with weaker in-gassing or even outgassing in winter add a comma after "weakens"

"The increase of sea surface temperature (SST) with summer weakens the surface CO2 solubility" The increase of sea surface temperature (SST) in summer reduces surface CO2 solubility

86-89 "Efforts to .... coupled simulations". Break this into two sentences at "however".

"exploring the mechanisms of the observed model biases" exploring the mechanisms underlying the model biases delete "the" before CO2SYS

delete "in the Southern Ocean"

delete "the" before F_DIC

Fig. 6 appears to be cited out of order here not clear what "ensemble" means in this context not clear what "decadal" means in this context

"In the Sub-Antarctic zone for all three basins, observed FCO2 show a weakening of CO2 uptake during winter (less negative values in JJA) with values close to the zero flux at the onset of spring" In the Subantarctic zone, in all three basins, there is a weakening of CO2 uptake during winter (less negative values in JJA) with values close to zero at the onset of spring

"In the Antarctic zone, the observed FCO2 seasonal cycle is similar in all three basins (Fig. 3d-f), possibly resulting from the limited number of observations." In the Antarctic zone, the observed FCO2 seasonal cycle is similar in all three basins (Fig. 3d-f). This apparent uniformity may result from the limited number of observations, and additional interbasin differences may arise as new observations are collected. (I am guessing at the intended meaning here.)

change "coherent" to "consistent"

"though they agree in the phasing, they have some differences in magnitudes" although they agree on the phase, the magnitude varies considerably

"Group A shows an overestimation of the CO2 uptake, while group-B shows an underestimation of CO2 uptake with respect to observations. This disagreement is accompanied by a large standard deviation, showing some inter-model differences in magnitudes (Fig. 2d-f)." Group A shows an overestimation of the CO2 uptake with respect to observations, while group-B shows an underestimation. The large standard deviation indicates considerable differences among the models (Fig. 2d‐f).

"CMIP5 models have a general positive bias against observations during summer and/or autumn with the exception of group-A models in the Sub-Antarctic zone." CMIP5 models have a generally positive bias relative to observations during summer and/or autumn, with the exception of group-A models in the Subantarctic zone.

change "an" to "the", delete "the" before "MPI-ESM"

change "harmony" to "consistency"

How can the observations be biased? Relative to what?

286-287 I would just delete everything after "representative".

293-295 This seems to imply that the sign change (-ve to +ve) is the same in spring and fall.

not clear what "amplitude" means in this context (see also 466)

delete "the" before "pCO2"

"It shows group A models (HadGEM-ES, NorESM2 and MPI-ESM) at the bottom of Fig. 6, indicating that these models are mainly DIC driven." It shows that group A models (HadGEM-ES, NorESM2 and MPI-ESM) are mainly DIC driven (Fig. 6). (I would double check all the model names here)

change "rich-CO2" to "CO2-rich"

"In the Antarctic zone CMIP5 models are largely coherent at the onset of MLD deepening (February), however significantly variable at the winter maximum depth." In the Antarctic zone, modelled MLD is quite consistent across models at the onset of MLD deepening (February), but more variable at the winter maximum.

[Figure]

"models cluster together in all the basins of the Sub-Antarctic zone with the only exception of CESM1-BGC in the Atlantic" models cluster together in all basins of the Subantarctic zone, with the exception of CESM1-BGC in the Atlantic change "scattered" to "divided"

delete "preceding"

remove ()'s around "biased"

add "convective" before "mixing"

change "is contrary to" to "differs from"

I would delete "add to the entrainment rate and"

416-424 this passage seems repetitive and could be shortened.

delete "flux"

"This is important because though annual means are useful" This is important because, although annual means are useful

"In the Antarctic zone, all selected CMIP5 models are in general agreement, as well as with observations on the dominant role of DIC in regulating FCO2 seasonal cycle" In the Antarctic zone, all of the CMIP5 models analyzed are in general agreement, and consistent with observations, on the dominant role of DIC in regulating the seasonal cycle of air-sea CO2 flux

"models and observations show little inter-basin differences in the seasonal cycle of FCO2, suggesting that mechanisms driving FCO2 are less localized" models and observations show weak interbasin differences in the seasonal cycle of FCO2, suggesting that mechanisms driving FCO2 are less spatially variable

"emergence of basin specific spatial characteristics of FCO2 might be inhibited by lack of observational coverage" basin-specific features may be obscured by incomplete data coverage in the observational data products delete "analyzed"

change "proposed" to "hypothesized"

469-470 "As evident in Fig. 8, bottom or subsurface DIC could not be the main driver of these amplified DIC amplitudes because the Antarctic zone shows comparable lower entrainment fluxes (Fig. 8)." This assertion does not make a lot of sense, and the figure legend and caption are ambiguous: I can't really tell which panels are for which regions. Anyway, what exactly are they trying to say here? That entrainment is shown to be a second order effect because rates are comparable in the Antarctic and Subantartic? The text as written makes it sound like the former is being contrasted with itself. The authors need to be clear about what exactly is being asserted here, and make sure that the assertion is actually supported by the data cited.

"this is consistent with Rosso et al., 2017 findings" this is consistent with Rosso et al. (2017)

"the seasonal cycle of chlorophyll in Fig. 9 show coherence (symmetric $\sim$ negative correlation) with the analyzed rate of change of DIC" the seasonal cycle of chlorophyll is anticorrelated with the rate of change of DIC (Fig. 9)

"may have an important role as suggested by Rysgaard et al., 2011 and Rosso et al 2017 and they should be investigated as part of a future study" may have an important role as suggested by Rysgaard et al. (2011) and Rosso et al. (2017) and should be investigated further add "flux" after "heat"

"influence the surface heat regulation" I can't tell what this means

"Therefore these analyzed temperature bias pose an important predicament with respect to our ability to predict future earth system changes, particularly the carbon cycle. We propose this bias as an important consideration to the model developing community as it relates to future biogeochemical and CMIP ESM development." These temperature biases pose an important problem with respect to our ability to model future changes in the carbon cycle. We suggest that diagnosing and reducing this bias should be a priority for future model development.

not clear what "exaggerated" means in this context change "on" to "of"

"We find that though some models exhibit comparable chlorophyll magnitudes with observations" We find that, although some models exhibit chlorophyll concentration comparable to observations (see also 526-527)

delete "the presentation of"

602, 687 references are incomplete

---

## Referee Comment (RC3) · Anonymous Referee #3 · 6 Nov 2017

Overall Statements:

The manuscript "Mechanisms of the Air-Sea CO2 Flux Seasonal Cycle biases in CMIP Earth Systems Models in the in the Southern Ocean" by N.P. Mongwe, M. Vichi, and P.M.S Monteiro is a valuable contribution to the inspection of regional and seasonal carbon related fluxes in global CMIP5 models. The authors show that even when annual carbon fluxes accord to data-derived fluxes, model results can differ substantially and show biases compared to data-derived fluxes on the seasonal and regional scale. The authors use diagnostic tools to attribute model result characteristics regarding the efficiency of the simulated biological- and solubility-carbon pump. These diagnostic

tools are based on empirical relations. The author's approach is thus not fully based on model results and leaves me not fully convinced. Are simulated strong DIC variations really due to high biological activities, and is the subsequent respiration really the cause for outgassing? This appears highly speculative.

On the other hand model characteristics which may imply a strong or even overestimated simulated biological pump are not shown or discussed. But this would help to understand the basic differences between the models.

For these two reasons it is necessary

to give more information on model characteristics (Tab. 1), and

to study additional model results (e.g., export production) which are available for CMIP5 models.

This manuscript is written very sloppily. This has to be improved substantially.

Detailed remarks:

Line 1: better "Earth System Models"

Line 24 -31: This part appears as repetition of the preceding part. You can highlight here some specific results (from Table 2).

Line 37: Define here the extent of the Southern Ocean as you use it within the manuscript and give the ratio of Southern Ocean to Global Ocean surface area.

Line 42: Ref missing

Line 44: Leung et al., 2015 Ref missing

Line 44: Roy et al., 2011 Ref missing

Line 45: Segschneider and Bendtsen, 2013 Ref missing

Line 45: Sarmiento Ref missing

Line 47: Le Quere et al 2007 Ref missing

Line 47: Son and Gerber 2010 Ref missing

Line 47: Thompson et al., 2011 Ref missing

Line 48: Landschutzer et al., 2015 Ref missing

Line 48: Zickfeld et al., 2008 Ref missing

Line 52: Please say once that all seasons you mention correspond to the Australian annual cycle.

Line 55: Taylor et al., 2012 Ref missing

Line 57: "almost all agree ..". Table 2 shows one exception.

Line 58: Anav et al., 2013 Ref missing

Line 60: Thomalla et al., 2011 Ref missing

Line 62 and Line 69 should be adapted

Line 62 ff: don't use different writings: in-gassing, outgassing, in gassing.

Line 63: "Ref" ???

Line 73: Sabine et al., 2004 Ref missing

Line 76: cooler than ?

Line 77: Marinov et al., 2006 Ref missing

Line 77: Metzl, 2009 Ref missing

Line 79: Matear and Lenton, 2008 Ref missing

Line 88: Rodgers et al., 2014 Ref missing

Line 88: Visineli et al., 2016 Ref missing

Line 99: Reference missing

Line 101: There are large differences between FCO2 used in this manuscript and values in Mongwe et al. (2016). Explain why you changed the data source. Why is it better than the former one?

Line 102: Landschützer et al (2014) Ref missing

Line 114: Pierrot and Wallace (2006) Ref missing

Line 115: Which equilibrium constants are used?

Line 129-130: Dunne et al (2013) Ref missing

Line 129-130: The hor. resolution of the MPI model is not correct

Line 129-130: Give additional information on model characteristics:

Parametrization of air-sea heat fluxes (e.g., dz(1); which layers are treated? ..)

Parametrization of air-sea flux of CO2 (e.g., Wanninkhof, 1992; etc)

Is an ice model included?

Which nutrients are included?

Line 136: Wanninkhof et al. (2009) Ref missing

Line 140: Ref missing

Line 150: Ref missing

Line 169: Does delta MLD(T+1) refer to delta MLD within line 168?

Line 184: Ref missing

Line 184-187 could be shifted into the introduction

Line 186: Give mean latitudinal extent of the two zones

[Figure]

Line 192-193 adapt PF and PFZ

Line 194: Four models do not broadly capture these features.

Line 218: Discuss also the RMSE

Lin 220: PCC(CESM1-BGC) = 0.47

Line 224: Adapt units to Tab. 2

Line 237: You mean annual standard deviation?

Line 237: Is the standard deviation also applied over the ensemble members?

Line 243: The letters a-f are missing

Line 244 "weakening of uptake or an increase of outgassing"

Line 249: There are more zonal differences in the model results than in the observations

Line 260: it's rather May

Line 268: I see an overestimation of group A, but no underestimation of group B

Line 273-275: Say it more straightforward: positive (red) – flux too high due to an overestimation of oceanic pCO2 and negative (blue) – flux too flow due to an underestimation of oceanic pCO2

P277: Only Pacific basin shown (compare headlines in Fig 4)

L285: "are relatively better" this important statement should be made more precisely. Could you evaluate Fig 4 numerically to do so?

Line 333 What's about the entrainment of alkalinity? This would damp the DIC effect on pCO2

Line 346: The simulated entrainment is much larger.

[Figure]

Line 386: "biological .." also other reasons for deltaDIC could exist

Line 390: Why should group B models do so? Maybe some additional characteristics (Tab. 1) could help to explain this.

Line 405: "Fig. 9d-f"

Line 406: The wording "anticipated" shows the high degree of speculation. By the way, I see only two type A models doing so.

Line 488: Ref missing

Line 457: Also other models show similar deltaT values

Line 515: It is not the pCO2 which is soluble

Line 584-585: New extra line

Line 654: really 2009? In the text it's 2007

Line 667: really 1995? In the text it's 2004

Line 697 ff: allign the Sallee references according to the year of publishing

Line 734: Vichi et al., 2007 or 2009?

Line 744: Figures and inscriptions too small

Line 759: It would help when the legend also provides the membership (group A – group B)

Line 792: Letter a-f missing.

Line 792: Only Pacific Ocean appears displayed

Line 860-861: Headings of the sub figures incorrect

Line 862: You mean "mol kg-1 month-1"?

Line 862 ff: Use "a"-"f"

Line 868-869: The y-axe for Chl is logarithmic, not the units

Line 870: You mean "mumol kg-1 month-1"?

Line 891: Give units

Line 891: Indicate group A and group B members

---

## Author Comment (AC1) · 24 Jan 2018

**1  Reviewer 1 (RC1)**

We thank the reviewer for the careful thought and suggestions, which we think have strengthened the clarity and rigor of our manuscript. We hope that we have been able to address all these points to the reviewer's satisfaction below and in the revised manuscript.

**General comments**

All required grammar corrections were done as suggested, with additional minor changes where necessary.  All are detailed in an annotated version of the document.

**Reviewer**: Unfortunately, I find the title beginning "Mechanisms..." to be a disappointing overreach as rather than including quantification of the solubility, transport, and biological pump mechanisms, the authors rely entirely on the empirical seasonal relationship correlation of dSST/dt and cursory analysis of mixed layer entrainment as metrics of model mechanisms

**Response**:  We apologize for giving this impression.  This study is based on a recently published mechanistic framework that used the seasonal cycle of $dpCO_2$ and $FCO_2$ as a mode to diagnose mechanistic differences between models and observations (Mongwe et al., 2016).  We, regretfully, neglected to provide a detailed description of how we separated the terms contributing to the total

DIC surface layer changes and how we compare these to temperature.  We have clarified this part in the revised manuscript. The total rate of change of DIC $\left(\frac{\partial DIC}{\partial t}\right)_{Tot}$ in the surface layer consists of the contribution of air-sea exchanges, biological, vertical and horizontal transport-driven changes (eq. 1).

$$\left(\frac{\partial DIC}{\partial t}\right)_{Tot} = \left(\frac{\partial DIC}{\partial t}\right)_{air-sea} + \left(\frac{\partial DIC}{\partial t}\right)_{Bio} + \left(\frac{\partial DIC}{\partial t}\right)_{Vert} + \left(\frac{\partial DIC}{\partial t}\right)_{Hor} \qquad \text{(eq.1)}$$

Because we used zonal means from medium resolution models, we assume that the horizontal terms are negligible.

Furthermore, in order to constrain the contribution of temperature on changing $pCO_2$ and $FCO_2$ we derived a DIC equivalent term $\left(\frac{\partial DIC}{\partial t}\right)_{SST}$ defined as the magnitude of DIC change that would correspond to a change in $pCO_2$ driven by a particular temperature change.  In this way the $\Delta pCO_2$, driven solely by modelled or observed temperature change, is converted into equivalent DIC units, which allows its contribution to be scaled against the observed or modelled DIC change (Eq.1).

This calculation is done in two steps: firstly, the temperature impact on $pCO_2$ is calculated using the

Takahashi et al., (1993) empirical expression that linearizes the temperature dependence of the equilibrium constants.

$\left(\frac{\partial pCO_2}{\partial t}\right)_{SST} = 0.0423 \times pCO_2 \times \frac{\partial SST}{\partial t}$                     (eq. 2)

Though this relationship between dSST and $dpCO_2$ is based on a linear assumption (Takahashi et al.,

1993), this formulation has been shown to hold and has been widely used in literature (e.g. Bakker et al., 2014; Feely et al., 2004; Marinov and Gnanadesikan, 2011; Takahashi et al., 2002; Wanninkhof et al., 2010). We show in the supplementary material that the extension of this expression into polar temperature ranges (SST < 2$^o$C) only introduces and additional uncertainty of 4 -5%.

Secondly, the temperature driven change in $pCO_2$ is converted to an equivalent DIC using the Revelle factor $\left(\gamma_{DIC} = \frac{DIC}{pCO_2}\frac{\partial pCO_2}{\partial DIC}\right)$.

$\left(\frac{\partial DIC}{\partial t}\right)_{SST} = \frac{DIC}{\gamma_{DIC} \times pCO_2}\left(\frac{\partial pCO_2}{\partial t}\right)_{SST}$              (eq. 3)

Although we used a fixed nominal polar Revelle factor of 14, we show in the supplementary material that this does not alter the phasing or magnitude of the relative controls of temperature or DIC on the seasonal cycle of $pCO_2$ (Fig. 1).

[Figure]

**Figure. 1** Seasonal cycle of the rate change of surface total DIC $\left(\frac{\Delta DIC}{\Delta t}\right)$ *black line*, and the estimated solubility DIC driven rate of change $\left(\frac{\partial DIC}{\partial t}\right)_{SST}$ *shaded area*, for monthly data given in µmol kg$^{-1}$

month$^{-1}$ at the Sub-Antarctic zone i.e. Pacific Ocean (first column), Atlantic Ocean (second column)

and Indian Ocean (third column). The dotted line shows the uncertainty boundaries for the Revelle factor in the Southern Ocean ($\gamma_{DIC}$ = 12, 15.5).

This methodology (Mongwe et al., 2016) is now fully described in the revised manuscript, before proceeding to compare directly the influence of temperature with the estimated total surface DIC

changes (eq. 3), The anomaly of the temperature contribution to $pCO_2$ change to total modelled or observed DIC change, expressed in equivalent DIC units, is set out in Eq. 3) below where a positive anomaly points to $\Delta pCO_2$ being thermodynamically controlled and a negative anomaly points to DIC

control:

This isolation of the role of solubility is the first step in our analysis, we then proceed to also look at the role of vertical DIC (entrainment) due to changes in the mixed layer depth and biological processes: now, in addition to the initial inclusion of biomass (chlorophyll), we have expanded it to include Net Primary Production (NPP), carbon export and oxygen to examine how DIC changes are driven by biological process. The main caveat is that we focus on processes that drive $CO_2$ variability on the vertical scale and, for now, have neglected the horizontal scale fluxes.  This assumption is thought to be reasonable given the large-scale zonal averages that we work with where the seasonal flux variability is likely to be dominated by vertical length scales as well as and medium resolution models we used for this analysis.

We were sorry to read that our analysis of the estimated DIC change at the base of the mixed layer to examine surface DIC changes driven by subsurface/bottom DIC variability was perceived as cursory.

These estimates are based on annual mean DIC profiles and seasonal MLD due to the availability of three-dimensional DIC data in the standard set of CMIP5 variables.  In order to reduce this uncertainty, we also ran an additional model (section 2.4) with comparable spatial resolution and verified that our conclusions were valid despite the use of annual means of the DIC distribution.

Thus, we re-emphasize that our approach is not correlation based, but does examine the variability of the main drivers of $CO_2$ at the seasonal scale i.e. DIC and temperature, which we thought could be useful in showing first order sources of the apparent CMIP5 $FCO_2$ seasonal cycle biases. Using this approach, we were able to show that overestimated warming and cooling rates were the main bias in group B CMIP5 models, while exaggerated primary production is the main bias in group A models.

This finding is an important consideration for the ability of Earth System Model to predict long-term changes in the oceanic $CO_2$ sink. It indicates that this ability is likely dependent on the model's capability to represent realistic seasonal changes in temperature and not just the mean state and ranges, because this has marked implications on DIC and $pCO_2$ solubility. This is especially important during the spring season where primary production (uptake) and solubility (surface warming) have an opposing effect on the direction of the $CO_2$ flux. The relative rates are critical to understanding the climate sensitivity of the model in respect of air-sea $CO_2$ fluxes.  We recognize that these points were not clear enough in our submitted manuscript. To better match with the revised content of the paper, we also propose to change our title to "The Seasonal Cycle of $CO_2$ fluxes in the Southern

Ocean: Diagnosing Anomalies in CMIP5 Earth Systems Models".

**Reviewer**: I suggest the authors quantify the role of SST change on DIC solubility to be able to confidently assess whether the role of temperature in the temperature correlated.

**Response**: This analysis does in fact quantify the role of SST change on $pCO_2$, which is then converted to DIC equivalents as explained above.  Strictly speaking, SST cannot change DIC in a closed system so we proposed the use of the DIC equivalent, which reflects the magnitude by which DIC would have to change if it were to make the same change as SST on $pCO_2$. We are here referring to an equivalent

DIC change resulting from $pCO_2$ change by solubility, this is to scale up the solubility component to total DIC changes (eq.1). The reviewer comments, however, highlight that our methodology was not adequately explained. In the revised manuscript we provide an expanded explanation of our methodology (see also the response above), which clarifies this point among other improvements.

**Reviewer**: The authors should include some quantification of model biases relative to observational uncertainty, and include several of the available observationally constrained products to assess that uncertainty

**Response**: Since our analysis is based on the Landschützer et al., 2014 data product, we have used the mean monthly $FCO_2$ (1998 – 2011) to compute the standard deviation of the seasonal cycle of

$FCO_2$. This is not strictly a measure of data uncertainty but more an estimate of the interannual variability that was used to compare against CMIP5 models variability in Fig. 2 & 3 of the original manuscript. For the annual means in Table 1, we use the uncertainty magnitude provided by

Landschützer et al., 2014 ($\pm0.31$ Pg C $yr^{-1}$) when comparing observations estimates against models.

We examine the uncertainty further by adding more data products, as suggested. We now compare

Landschützer et al., (2014) with the more recent Gregor et al (2017) data product, which uses

Support Vector Regression (SVR) and Random Forest Regression (RFR), as well as Takahashi et al (2009) for $pCO_2$ climatology in the supplementary material. We use $pCO_2$ instead of $FCO_2$ firstly, because Gregor et al., (2017) only provided fugacity and pCO$_2$, and also being mindful that the choice of wind product and tranfer veolocity constant in computing FCO$_2$ would increase the level of uncertinty (Swart et al., 2014). Secondly, while the focus of the paper is on the evaluation of FCO$_2$

biases, the major part of our diagnostic analysis is based on pCO$_2$, which determines the direction and part of the magnitude of the fluxes.

Fig. 2 below shows the seasonal cycle of pCO$_2$ in the Sub-Antarctic zone and Antarctic zone with interannual standard deviation between 1998 – 2011 and their corresponding FCO$_2$ climatology. All three datasets mostly agree in the phasing of the seasonal cycle of pCO$_2$ in the Sub-Antarctic, but show differences in the magnitude. Tatakahashi et al. (2009) shows an amplified impact of primary production in summer. We see this as a bias in the Takahashi et al., (2009) dataset arising from a period when the space – time coverage of pCO$_2$ observations was still limited and strongly biased towards summer. In the Antarctic zone these three observationally-based datasets agree in both phasing and amplitude. At this stage it is not clear whether this agreement is due to all the methods being equally exposed to the same few observations or it is due to a more marked CO$_2$ seasonal cycle in the Antarctic zone (relative to Sub-Antarctic) that can be captured with less observations.

[Figure]

**Figure. 2** pCO$_2$ (µatm) spatial (climatology) and seasonal cycle differences in Landschützer et al (2014),  Gregor et al (2017), Takahashi et al (2009) datasets in the Southern Ocean. The seasonal cycle climatology of pCO$_2$ in the Sub-Antarctic and Antarctic zone is based on the period 1998 – 2011.   The shaded areas show the standard deviation of the interannual variability of the seasonal cycle for this period. The uncertninity in the correlation coeffecient is based on the correlation coeffient of the mean plus stardard deviations seasonal cycle(s).

**Reviewer**: Interior budgets could be constructed, leading the authors to developing a simple box model of mixed layer DIC to be able to reproduce the various GCM results through the combination of gas exchange, thermal, transport, and biological mechanisms. While such a more mechanistically based box model analysis could prove very valuable in uncovering the mechanistic differences between the models, it is probably outside the scope of the present manuscript

**Response**:  This is an excellent suggestion to isolate different drivers of FCO$_2$, but it will be a separate study as it is out the scope of this analysis. Nevertheless, we hope that our revised manuscript clarifies our approach and its usefulness as an analysis of the mechanisms of the main drivers of CO$_2$

at the seasonal scale.

**Specific Comments**

We hope that all grammatical and spelling challenges have been addressed with minor changes where necessary.

**Reviewer**: 135 The assertion that "The seasonal cycle of the ocean-atmosphere CO2 gradient dpCO2

is considered to be the main driver of the seasonal variability of FCO2" …… Is true in a regional sense but is certainly not true in a temporal sense in most regions where wind variability can dominate like in the equatorial Pacific: The delta pCO2 argument was that is you average over large enough scales, the mixed layer equilibration time of CO2 was short enough (about a year) that CO2 fluxes were determined by the net balance of biology and thermal factors rather than the wind. On a seasonal scale, ignoring the role of wind seems like a fatal flaw. Rather, the authors should argue that the wind variability in this region is small before disregarding it. This is likely true in the Southern Ocean where winds are strong in all seasons.

**Response**:  This an important point, which may have other implications elsewhere outside the

Southern Ocean.  Here, while it is correct that winds provide the variability in kinematic forcing for sea-air CO$_2$ interactions, the weak seasonal cycle in wind stress in the Southern Ocean(Young, 1999)

means that the impact on FCO$_2$ is largely in the intra-seasonal (synoptic) scales.  The impact of wind in the mixed layer dynamics can also be amplified or suppressed depending on the mesoscale and sub-mesoscale characteristics of the surface ocean(Mahadevan et al., 2012; du Plessis et al., 2017)

which may also play a role on the onset of and variability of primary production and entrainment (and thus $FCO_2$).  In contrast,  $\Delta pCO_2$, which sets the direction and also contributes to part of the magnitude of the flux, is regulated by the strong seasonal modes of solar warming, which drives SST

and mixed layer depth (MLD) that influences the seasonal extremes of spring-summer productivity and winter convective entrainment in the Southern Ocean.

However doing this analysis has some complexity because different wind products result in different

$FCO_2$ responses (Swart et al., 2014), which both highlight a strong sensitivity of $pCO_2$ to winds and a challenge for choosing reliable wind product. For this analysis, as the reviewer pointed out, because winds do not have a strong seasonal variability in the Southern Ocean, we don't anticipate a strong seasonal impact on $FCO_2$ and it was excluded from the main text. We make this point in revised manuscript. Nevertheless we recognize that evaluating the impact of winds on $FCO_2$ in both setting the mean-state and inducing fine scale dynamics important $CO_2$ at the seasonal scale remains an important aspect and will be considered for a future study.

**Reviewer:** 181 – While I am glad the authors are considering mixed layer entrainment, it seems remiss here to ignore the biological and other circulation terms such as upwelling and consider them all lumped together as "DIC" terms.

**Response:** Once more, we recognize that the description of the separated terms was short, and they were all referred to as DIC drivers together. We also only relied on surface chlorophyll, which is indeed a measure of standing stock, to explain the biological CO2 uptake.  In the revised manuscript and in the answer to the first main comment above, we have clarified this point by showing all the terms and how we consider them to contribute to the surface layer changes of DIC (eq. 1).  We also explain that we neglect the horizontal term as we make a regional average over the whole sub-

Antarctic and Antarctic regions. While we don't provide an explicit estimation of vertical transport, we use the discretized DIC changes at the base of the mixed layer to provide an estimate of surface

DIC changes driven by winter convective entrainment.

In the revised manuscript, we added net primary production (NPP), surface oxygen and carbon export to help constrain the role of biological DIC changes from entrainment fluxes. The addition of

NPP and Carbon export improved our separation of the biological terms for DIC changes from entrainment.

**Reviewer:** 184 A brief description of the Orsi definition should be provided here.

**Response:** A brief description of Orsi definition was added,

"In this study, we partition the Southern Ocean into 2 zones using the criteria proposed by Orsi et al., (1995). It is defined as the ocean south of the Sub-tropical front (STF: 11.3$^o$C isotherm at 100 m)

[Orsi et al., 1995] and divided into two main domains, the Sub-Antarctic Zone between the STF and the Polar Front (PF, 2$^o$C isotherm at 200 m) the Antarctic Zone south of the PF. We further partition the domain into the three main basins of the Southern Ocean i.e. Pacific, Atlantic and the Indian

Ocean."

**Reviewer:** 249 – The statement that the models "do not capture any of the basin-specific features" is a fairly strong, but non-quantitative statement. This should be much more specific – like, the observational reanalysis shows a stronger flux in the Atlantic than the Indian and Pacific while the models show similar fluxes in each basin.

**Response:** We thank the reviewer for this suggestion. As suggested in the main comments above, we have now made the analysis more quantitative in general, and here in particular, we added a measure of how different the seasonal cycles from the various models are from the observational data products by using the correlation coefficient (Fig.2, revised Figure 4 in the new manuscript). We have also corrected the sentence as suggested.

[Figure]

**Figure. 2** Sea-Air $CO_2$ Flux mean seasonal and annual biases with respect to observations (gC m$^{-2}$ yr$^{-1}$) for the

Sub-Antarctic and Antarctic zones in the Pacific Ocean (first column, a and d), Atlantic Ocean (second column, b and e) and Indian Ocean (third column, c and f). $CO_2$ out-gassing biases are in red, while blue color intensity shows in-gassing biases. The models are sorted according to the annual mean bias presented in the last column (Amean).  The quantitative assessment of the difference between the seasonal cycle of the data product and models is based on the correlation coefficient and its confidence interval.

**Reviewer:** 319 – "justifies our a priori separation" comes across as inappropriate self- congratulation.

The salient point is that the separation quantifies the separation be- tween the two classes of models in terms of the relative dominance of SST-Flux correlation.

**Response:** We thank the reviewer for pointing out this issue and suggesting a proper sentence. We changed the text as follows;

"The model groupings that emerge from the calculated $M_{T-DIC}$ for CMIP5 models and observations quantify in terms of the relative role of temperature the a priori separation between group A and group B fluxes in the Sub-Antarctic zone that we proposed in section 3.2. It shows that the CO2 flux in group A models (HadGEM2-ES, NorESM2 and MPI-ESM) is mainly biologically driven while all group

-B models show a stronger temperature control (solubility is the main mode driving $pCO_2$ seasonal variability), particularly in the Sub-Antarctic zone (Fig. 6a-c). In the Antarctic zone $M_{T-DIC}$ magnitudes are relatively more similar between CMIP5 models and observations, which is supported by $FCO_2$

seasonality."

**References**

Bakker, D. C. E., Pfeil, B., Smith, K., Hankin, S., Olsen, a., Alin, S. R., Cosca, C., Harasawa, S., Kozyr, a., Nojiri, Y., O'Brien, K. M., Schuster, U., Telszewski, M., Tilbrook, B., Wada, C., Akl, J., Barbero, L., Bates, N. R., Boutin, J., Bozec, Y., Cai, W. J., Castle, R. D., Chavez, F. P., Chen, L., Chierici, M., Currie, K., De Baar, H. J. W., Evans, W., Feely, R. a., Fransson, a., Gao, Z., Hales, B., Hardman-Mountford, N. J., Hoppema, M., Huang, W. J., Hunt, C. W., Huss, B., Ichikawa, T., Johannessen, T., Jones, E. M., Jones, S. D., Jutterström, S., Kitidis, V., Körtzinger, a., Landschützer, P., Lauvset, S. K., Lefèvre, N., Manke, a. B., Mathis, J. T., Merlivat, L., Metzl, N., Murata, a., Newberger, T., Omar, a. M., Ono, T., Park, G. H., Paterson, K., Pierrot, D., Ríos, a. F., Sabine, C. L., Saito, S., Salisbury, J., S. Sarma, V. V. S., Schlitzer, R., Sieger, R., Skjelvan, I., Steinhoff, T., Sullivan, K. F., Sun, H., Sutton, a. J., Suzuki, T., Sweeney, C., Takahashi, T., Tjiputra, J., Tsurushima, N., C. Van Heuven, S. M. a, Vandemark, D., Vlahos, P., Wallace, D. W. R., Wanninkhof, R. and Watson, a. J.: An update to the surface ocean CO2 atlas (SOCAT version 2), Earth Syst. Sci. Data, 6(1), 69–90, doi:10.5194/essd-6-69-2014, 2014.

Feely, R. A., Wanninkhof, R., McGillis, W., Carr M. E and Cosca, C.: Effects of wind speed and gas exchange parameterizations on the air-sea $CO_2$ fluxes in the equatorial Pacific Ocean, J. Geophys. Res., 109(C8), C08S03, doi:10.1029/2003JC001896, 2004.

Gregor, L., Kok, S. and Monteiro, P. M. S.: Empirical methods for the estimation of Southern Ocean CO2: Support Vector and Random Forest Regression, Biogeosciences Discuss., (June), 1–18, doi:10.5194/bg-2017-215, 2017.

Landschützer, P., Gruber, N., Bakker, D. C. E. and Schuster, U.: Recent variability of the global ocean carbon sink, Global Biogeochem. Cycles, 28(9), 927–949, doi:10.1002/2014GB004853, 2014.

Mahadevan, A., D'Asaro, E., Lee, C. and Perry, M. J.: Eddy-driven stratification initiates North Atlantic spring phytoplankton blooms, Science (80-. )., 336(6090), 54–58, doi:10.1126/science.1218740, 2012.

Marinov, I. and Gnanadesikan, a.: Changes in ocean circulation and carbon storage are decoupled from air-sea CO2 fluxes, Biogeosciences, 8(2), 505–513, doi:10.5194/bg-8-505-2011, 2011.

Mongwe, N. P., Chang, N. and Monteiro, P. M. S.: The seasonal cycle as a mode to diagnose biases in modelled CO2 fluxes in the Southern Ocean, Ocean Model., 106, 90–103, doi:10.1016/j.ocemod.2016.09.006, 2016.

Orsi, A. H., Whitworth, T. and Nowlin, W. D.: On the meridional extent and fronts of the Antarctic

Circumpolar Current, Deep. Res. Part I, 42(5), 641–673, doi:10.1016/0967-0637(95)00021-W, 1995.

du Plessis, M., Swart, S., Ansorge, I. J. and Mahadevan, A.: Submesoscale processes promote seasonal restratification in the Subantarctic Ocean, J. Geophys. Res. Ocean., 122(4), 2960–2975, doi:10.1002/2016JC012494, 2017.

Swart, N. C., Fyfe, J. C., Saenko, O. A. and Eby, M.: Wind-driven changes in the ocean carbon sink,

Biogeosciences, 11(21), 6107–6117, doi:10.5194/bg-11-6107-2014, 2014.

Takahashi, T., Olafsson, J., Goddard, J. G., Chipman, D. W. and Sutherland, S. C.: Seasonal variation of

CO2and nutrients in the high-latitude surface oceans: A comparative study, Global Biogeochem.

Cycles, 7(4), 843–878, doi:10.1029/93GB02263, 1993.

Takahashi, T., Sutherland, S. C., Sweeney, C., Poisson, A., Metzl, N., Tilbrook, B., Bates, N.,

Wanninkhof, R., Feely, R. a, Sabine, C., Olafsson, J. and Nojiri, Y.: Global sea – air CO 2 flux based on climatological surface ocean p$CO_2$ , and seasonal biological and temperature effects, Deep Sea Res.

Part II Top. Stud. Oceanogr., 49(9–10), 1601–1622, 2002.

Wanninkhof, R., Park, G. H. and Risien C.M: Impact of small-scale variability on air–sea CO2 fluxes,

Gas Transf. …, 431–444 [online] Available from:

ftp://wombat.coas.oregonstate.edu/pub/chelton/papers/wanninkhof_etal_2011_GTWS.pdf, 2010.

Young, I. R.: Seasonal variability of the global ocean wind and wave climate, Int. J. Climatol., 19(9),

931–950, doi:10.1002/(SICI)1097-0088(199907)19:9<931::AID-JOC412>3.0.CO;2-O, 1999.

---

## Author Comment (AC2) · 24 Jan 2018

**Reviewer 3 (RC3)**

We thank the reviewer for the examination of our manuscript, provided reviews were helpful in improving the manuscript. We hope we have addressed all comments satisfactory below. The manuscript has been improved with respect to the submitted version, below we detail some of the improvements specifically requested.

**General comments**

**Reviewer:** These diagnostic tools are based on empirical relations. The author's approach is thus not fully based on model results and leaves me not fully convinced. Are simulated strong DIC variations really due to high biological activities, and is the subsequent respiration really the cause for outgassing? This appears highly speculative.

**Response:** We understand the reviewer's concern and we are confident that they are mostly due to our too brief explanation of the methodology. We relied excessively on a previous published paper by Mongwe et al. (2016) where the methodology was explained in full detail and we have now improved this part in the Method section. This was also compounded by some missing references, including Mongwe et al., (2016). Having used referencing software, I neglected a thorough check of the bibliography, which was a mistake and I apologize for this inexcusable mistake; we have corrected all references in the revised manuscript.

The approach is fully based on model process analysis, though it employed the widely used Takahashi et al, (1993) empirical relationship that linearized the temperature dependence of pCO$_2$.  We also neglected to provide a description of how we separated the terms contributing to the total DIC surface layer changes, as pointed out by another reviewer. We have clarified this part in the revised manuscript. The total rate of change of DIC $\left(\frac{\partial DIC}{\partial t}\right)_{Tot}$ in the surface layer consists of the contribution of air-sea exchanges, biological, vertical and horizontal transport- driven changes (eq. 1).

$\left(\frac{\partial DIC}{\partial t}\right)_{Tot} = \left(\frac{\partial DIC}{\partial t}\right)_{air-sea} + \left(\frac{\partial DIC}{\partial t}\right)_{Bio} + \left(\frac{\partial DIC}{\partial t}\right)_{Vert} + \left(\frac{\partial DIC}{\partial t}\right)_{Hor}$            (eq.1)

Because we used zonal means from medium resolution models, we assume that the horizontal terms are negligible.

Furthermore, in order to constrain the contribution of temperature on changing pCO$_2$ and FCO$_2$ we derived a new DIC equivalent term

$\left(\frac{\partial DIC}{\partial t}\right)_{SST}$ defined as the magnitude of DIC change that would correspond to a change in pCO$_2$ driven by a particular temperature change.  In this way the ΔpCO$_2$, driven solely by modelled or observed temperature change, is converted into equivalent DIC units, which allows its contribution to be scaled against the observed or modelled DIC change (Eq.1).

This calculation is done in two steps: firstly, the temperature impact on pCO$_2$ is calculated using the Takahashi et al., (1993) empirical expression that linearizes the temperature dependence of the equilibrium constants.

$\left(\frac{\partial pCO_2}{\partial t}\right)_{SST} = 0.0423 \times pCO_2 \times \frac{\partial SST}{\partial t}$            (eq. 2)

Though this relationship between dSST and dpCO$_2$ is based on a linear assumption (Takahashi et al., 1993), this formulation has been shown to hold and has been widely used in literature (e.g. Bakker et al., 2014; Feely et al., 2004; Marinov and Gnanadesikan, 2011; Takahashi et al.,

2002; Wanninkhof et al., 2010). We show in the supplementary material that the extension of this expression into polar temperature ranges (SST < 2$^o$C) only introduces and additional uncertainty of 4 -5%.

Secondly, the temperature driven change in pCO$_2$ is converted to an equivalent DIC using the Revelle factor $\left( \frac{\ln (pCO_2)}{\ln (DIC)} = \gamma_{DIC} \right)$.

$$\left( \frac{\partial DIC}{\partial t} \right)_{SST} = \frac{DIC}{\gamma_{DIC} \times pCO_2} \left( \frac{\partial pCO_2}{\partial t} \right)_{SST} \qquad\qquad \text{(eq. 3)}$$

Although we used a fixed nominal polar Revelle factor of 14, we show in the supplementary material that this does not alter the phasing or magnitude of the relative controls of temperature or DIC on the seasonal cycle of pCO$_2$ (Fig. 1).

Using this approach, we were able to link temperature (solubility) issues to the carbon flux discrepancies diagnosed in a group of models that was named Group B, as opposed to the other models in group A. Overestimated warming and cooling rates in the transitional seasons were found to be the source of the CO$_2$ flux bias in group B models. Our further analysis demonstrated that the estimated physical processes associated to changes in the MLD are not enough to explain the behavior of group A models, and the only process that was evident, was exaggerated primary production (and possibly associated shallow respiration).  Therefore, the DIC variations and the related CO$_2$ fluxes are linked to biological processes only for one class of the analyzed models (group A).

We then proceed to investigate the role of the other terms in eq. 1, namely the biological rate and the vertical DIC entrainment at the base of the mixed layer (we now make clear that the horizontal circulation term is neglected due to the regional averaging). For a more explicit analysis of the biologically linked DIC changes we have now added Net Primary Production (NPP), carbon export fluxes and oxygen concentrations in addition to the biomass (chlorophyll) that was used in the submitted original manuscript. This approach has improved our analysis of the biological influenced DIC variability; in particular it helped  constrain the role of primary production on the seasonal cycle of

$pCO_2$. It's true that the linked role of respiration in the DIC changes remain a speculation as we have not explicitly quantified respiration, nevertheless we now include more information supporting this speculation i.e. NPP and surface oxygen. Because of the lack of appropriate model output variables, the role of respiration is not fully explored here, however we maintain the speculation in the Discussion that the role of near surface respiration on DIC changes remain an important question for future research.

**Reviewer:** Give additional information on model characteristics: Parameterization of air-sea heat fluxes (e.g., dz(1); which layers are treated?

..) Parametrization of air-sea flux of CO2 (e.g., Wanninkhof, 1992; etc)  Is an ice model included?

**Response:**  We have interpreted this question as a request to report the specific parameterizations (or numerical schemes) used in the coupled models to exchange heat and gas between the atmosphere and the ocean. We can confirm that all the models used the Wanninkhof (1992)

parameterization for the air-sea carbon fluxes. As regards to the heat fluxes, this information is unfortunately not simple to retrieve as not explicitly written in the publications or in the technical reports. It is specific to the physical and numerical coupling between ocean and atmosphere models used in each ESM. We could only find it for a few models we have direct experience of. For instance, in the NEMO-based models, the heat flux is parameterized as a boundary condition for the vertical turbulent diffusion equation. We believe this is the standard method for all the other models but could not verify it.

Given the fact that all models tend to show the same kind of bias in the simulation of the rate of change of sea surface temperature except for

NorESM1, we would not ascribe it to a different choice of the heat flux parameterization.

All the climate models participating to CMIP5 have an active sea ice model. We have included more information in the table so that the reader can trace the different components used in the ESMs i.e. ocean, sea ice and atmospheric model, nutrient state variables, vertical coordinates and the various spatial resolutions.

| Full name and Source | Model Name | Ocean Resolution | Atmospheric Resolution | Nutrients | Sea ice model | Veridical Coordinate & Levels | Ocean Biology | Reference |
|---|---|---|---|---|---|---|---|---|
| **Canadian Centre for Climate Modelling and Analysis, Cananda** | CanESM2 | CanOM4 $0.9^{o}$ x$1.4^{o}$ | $2.8125^{o}$ x $2.8125^{o}$ | N (accounts for Fe limitation) | CanSIM1 | z 40 levels | NPZD | Zahariev et al., 2008 |
| **Centro Euro-Mediterraneo Sui Cambiamenti Climatici, Italy** | CMCC-CESM | OPA8.2 $0.5$-$2^{o}$x$2^{o}$ | $3.8^{o}$ x $3.7^{o}$ | P, N, Fe, Si | CICE4 | z 21 levels | PELAGOS | Vichi et al., 2007 |

| | | | | | | | | |
|---|---|---|---|---|---|---|---|---|
| **Centre National de Recherches Météorologiques-Centre Européen de Recherche et de Formation Avancée en Calcul Scientifique, France** | CNRM-CM5 | NEMOv3.3 $1^o$ | $1.4^o$ | P, N, Fe, Si | GELATO5 | z 42 levels | PISCES | Séférian et al., 2013 |
| **Institut Pierre-Simon Laplace, France** | IPSL-CM5A-MR | NEMO2.3 $0.5\text{-}2^o$ x $2^o$ | $2.58^o$ x $1.25^o$ | P, N, Fe, Si | LIM2 | z 31 levels | PISCES | Séférian et al., 2013 |
| **Max Plank Institute for Meteorology, Germany** | MPI-ESM-MR | MPIOM $1.41^o \times 0.89^o$ | $1.875^o$ x $1.875^o$ | P, N, Fe, Si | MPIOM | z 40 levels | HAMOCC 5.2 | Ilyina et al., 2013 |
| **Community Earth System Model, USA** | CESM1-BGC | $0.3^o$ x$1^o$ | $0.9^o$ x $1.25^o$ | (P), N, Fe, Si | | z 60 levels | BEC | Moore et al., 2004 |
| **Norwegian Earth System Model,** | NorESM1-ME | MICOM $0.5^o$ x $0.9^o$ | $2.5^o$ x $1.9^o$, | P, N, Fe, Si | CICE4.1 | $\rho$ 53 levels | HAMOCC | Tjiputra et al., 2013 |

| | | | | | | | | 69 |
|---|---|---|---|---|---|---|---|---|
| **Norway** | | | | | | | | |

**Specific Comments;**

**Reviewer:** There are large differences between $FCO_2$ used in this manuscript and values in Mongwe et al. (2016). Explain why you changed the data source. Why is it better than the former one?

**Response:** Our switch to Landschützer et al (2014) data product was based on the fact that it is based on more observations (SOCAT2, ~ 15

million surface measuremets ) and provided as a $1^o$x$1^o$ global gridded product , compared to $5^o$x $4^o$ in Takahashi et al., 2009, ~ 3 million surface measuremets. This increased the number of grid points per region (Sub-Antarctic and Antarctic) thus most likely making it a more appropriate basis to compare against CMIP5 models. Nevetherless, Landschützer et al (2014) is also subject to uncertainties arising from the empirical

Nearest-Neighbour interpolations and we are mindful to refer to this dataset as observational data product. We also provide an uncertainty evaluation in the supplementary material of the revised manuscript, as requested by another reviewer.  Since our analysis is based on the

Landschützer et al., 2014 data product, we used the mean monthly $FCO_2$ (1998 – 2011) to compute the standard deviation of the seasonal cycle of $FCO_2$. to compare against CMIP5 models variability in Fig. 2 & 3 of the original manuscript.

**Reviewer:** Line 115: Which equilibrium constants are used?

**Response:** K1, K2 from Mehrbach et al., 1973 refitted by Dickson and Millero, 1987, added to the revised manuscript.

**Reviewer:** Line 237: Is the standard deviation also applied over the ensemble members?

**Response:** Yes, we applied the standard deviation over the 10 years.

**Reviewer**: Line 244 "weakening of uptake or an increase of outgassing"

**Response:** Here we mean weakening of uptake since the net $CO_2$ flux remains in the negative phase of the seasonal cycle (net ingassing flux).

**Reviewer:** Line 249: There are more zonal differences in the model results than in the observations

**Response:** Though the CMIP5 models show some differences in the seasonal cycle between the basins, the $FCO_2$ seasonal cycle phasing mostly remain the same between basins in CMIP5 models, except for CESM1-BGC. The observational data product, in contrast, show almost different seasonal cycle phasing in each basin. We clarify this in the revised manuscript.

**Reviewer:** Line 268: I see an overestimation of group A, but no underestimation of group B

**Response:** The reviewer is correct to point this out, this sentence was not well explained. Here we meant underestimation with respect to observations $CO_2$ uptake. During the summer season (DJF) group A models ($FCO_2$ ~ -0.12 gC m$^{-2}$ day$^{-1}$) show an overestimated uptake, while group B ($FCO_2$ ~ 0 gC m$^{-2}$ day$^{-1}$) underestimates the $CO_2$ uptake relative to observations ($FCO_2$ ~ -0.03 gC m$^{-2}$ day$^{-1}$).  We rephrased this passage to try to make this point clearer.

**Reviewer**: Line 333 what's about the entrainment of alkalinity? This would damp the DIC effect on pCO2

**Response:** In the supplementary material of the revised manuscript (referenced to in the method description) we show that the DIC changes due to salinity and total alkalinity are much smaller relative to the total surface DIC changes and temperature driven DIC changes (solubility);

As shown in Fig. 2 & Fig. 3, the surface total DIC change $\left(\left(\frac{\partial DIC}{\partial t}\right)_{Tot}\right)_{average} \approx 10\frac{\mu mol}{kg\ month}$, while $\left(\left(\frac{\partial DIC}{\partial t}\right)_{Sal}\right)_{maximum} \approx 0.6\frac{\mu mol}{kg\ month}$ and

$\left(\left(\frac{\partial DIC}{\partial t}\right)_{Talk}\right)_{maximum} \approx 0.4\frac{\mu mol}{kg\ month}$. Thus $\left(\frac{dDIC}{dt}\right)_{Sal} \ll \left(\frac{\Delta DIC}{\Delta t}\right)$ & $\left(\frac{dDIC}{dt}\right)_{Talk} \ll \left(\frac{\Delta DIC}{\Delta t}\right)$,  these (salinity and Talk equivalent DIC changes) are about 10 orders of magnitude smaller than the equivalent solubility and total DIC change i.e. 0.5 µmol kg$^{-1}$ vs 5 µmol kg$^{-1}$ respectively.

Consequently, though salinity and Talk do play a role, they have been neglected from the analysis.

[Figure]

**Figure. 2** Shows the seasonal cycle of the rate equivalent DIC changes driven changes in Salinity and total alkalinity computed using Takahashi et al., (1993) formulations.

**Reviewer:** Line 406: The wording "anticipated" shows the high degree of speculation. By the way, I see only two type A models doing so.

**Response:** We agree with the reviewer that the phrasing of the sentences sounded speculative. This analysis was strengthened by the addition of NPP and carbon export in the revised manuscript (see also the answer to the main comments above).  It is indeed true that the role of respiration remains speculative here since we don't explicitly quantify the role respiration and we clarify this point in the manuscript. All three models in group A do show this behavior, though HadGEM2-ME only shows a modest effect, and this is now highlighted in the revised manuscript.

---

## Author Comment (AC3) · 24 Jan 2018

**Review 2 (RC2)**

We thank the reviewer for the detailed review of our manuscript. The review was indeed helpful in strengthening the manuscript. We hope we have addressed all comments satisfactorily.

All grammar corrections where done as suggested with minor changes where necessary.  All are detailed in an annotated version of the document.

**General comments**

**Reviewer:** I think the methodology could be better explained. Perhaps the logic of equations

1-3 is explained more thoroughly in the previous paper by Mongwe et al 2016, but this critical reference is missing from the reference list. Assuming they mean the paper in Ocean

Modelling 106: 90, I will agree that equations 1 and 2 can be derived from equations 3.2-3.4

in that paper. But the LHS of equation 2, which does not appear in the previous work, is physically speaking, a fairly nebulous quantity. The Takahashi et al 1993 estimate of 0.0423

$K^{-1}$ mainly expresses the change in partial pressure due to changing temperature for a given concentration of $CO_2$ ($[CO_2^*]$), with a small contribution from the partitioning of DIC

among $CO_2/HCO_3^-/CO_3^{2-}$ due to the temperature dependence of the equilibrium constants.

DIC does not change as a result of changes in temperature, except indirectly through gas exchange.

So what we have here is an observed change of $pCO_2$ with changing temperature, convoluted into a change of DIC by application of several highly empirical conversion fac- tors (more about this below), as an estimate of the changes in DIC not attributable to biological uptake/remineralization (and therefore primarily attributable to gas exchange).

This in itself might be inoffensive, but I would prefer if its relationship to actual physical processes were better explained. The equations that are taken as a starting point are highly empirical, and we should not invest rearrangements of these with an outsized significance.

0.0423 $K^{-1}$ is intended to be an average value for a broad range of ocean conditions, but it is stated to be valid for salinities from 34-36 and temperatures from 2-28$^o$C (Takahashi et al

1993). In the Southern Ocean one will encounter conditions outside, or on the far edges of, these ranges. What are the implications of this for the analysis shown here? This seems like something that could be evaluated. Similarly, the calculations assume a constant Revelle factor, but it should be quite straightforward to calculate Revelle factors from the model outputs, giving a range for the range of environmental conditions characteristic of the study area. The conclusions are probably robust to these assumption, I see no reason why not tested.

**Response:** The reviewer points out an overarching request to clarify our methodology, which indicates that it was not clearly presented. This is partly because the link to our previous paper (Mongwe et al., 2016), where we explained this methodology, was poorly made.  A more detailed synthesis from Mongwe et al., (2016) was necessary as suggested.

This was compounded by some inadvertently missing references, including the Mongwe et al., 2016. Having used referencing software, I neglected a thorough check of the bibliography, which was a mistake and I apologize for this mistake; all references in the revised manuscript have been corrected.

This reviewer's comment on the methodology is partly connected to the next comment asking how we ascribed the influence of temperature to DIC and the partitioning of the other DIC terms. We have deferred this part of the response to the next comment, in which this question is raised in a more specific way.  In this first part we focus on the question about the uncertainties linked to the constants of the Takahashi et al., (1993) empirical expression as well as the Revelle factor in polar waters.  We thank the reviewer for raising the question on limits to the validity of the empirical relationship used to calculate temperature-driven changes to $pCO_2$ observations and model data.

We examined the applicability of the Takahashi et al., (1993) linear approximation

$\left(\frac{1}{pCO_2}\frac{\partial pCO_2}{\partial SST} \approx 0.0423^o C^{-1}\right)$ in our region of study. In the Sub-Antarctic zone, surface water has a temperature range of ~$4^o$C – $12^o$C, which is within the limits ($2 - 28^o$C) provided by

Takahashi et al (1993).  Since surface temperatures go below $2^o$C in the Antarctic zone, we have tested whether this relationship can be extrapolated down to -$2^o$C. We do so by comparing the dependence of $pCO_2$ on temperature for a range of temperature values ($4^o$C

to -$2^o$ : $0.1^o$C intervals) using a carbonate equilibrium model and the Takahashi et al., (1993)

linearization.  Ancillary variables DIC, TAlk, phosphate, silicate and salinity are fixed and the carbonate equilibrium model CO2SYS (Pierrot et al., 2006) was used with K1, K2 from

Mehrbach et al., (1973) refitted by Dickson and Millero, (1987), which are the same parameterizations used in the majority of CMIP5 models. We used the mean climatological

$pCO_2$ (Landschützer et al., 2014) and the seasonal mean for nutrients (silicate and phosphate), salinity, TAlk and DIC from the same Antarctic region in GLODAP2 as was used in the models.

As shown in Fig. 1, the response of $pCO_2$ to temperature below 2$^o$C can still be described with the Takahashi et al., (1993) linear relationship. Thus we can extrapolate the Takahashi et al., (1993) linear dependence of $pCO_2$ to temperature for the estimation of temperature solubility changes to equivalent DIC changes in the Antarctic zone as explained above.

[Figure]

**Figure. 1** (S4 in the revised manuscript) in the revised supplementary material) Comparison between of dependence of $pCO_2$ to temperature changes according to the Takahashi et al., (1993) empirical the constant (0.0423$^o$C$^{-1}$) and the computed ratio of temperature dependence, from the carbonate system equations (CO2SYS, Pierrot, et al., 2006) using mean climatological data from GLODAP2

(Salinity, TAlk, DIC, silicate & phosphate) and $pCO_2$ from Landschützer et al (2014) for the Antarctic zone and a temperature range of -2.0$^o$C to 4$^o$C (0.1$^o$C intervals).

In a comparable way, we examined the uncertainties arising from a fixed polar Revelle factor in our calculations. We recomputed the Revelle factor in the Sub-Antarctic and Antarctic zones using annual mean climatologies of TAlk, salinity, surface temperature nutrients.

Firstly we examined DIC changes for the nominal range of $pCO_2$ change (340 − 399 µatm, 1

µatm intervals) and then used this dataset to derive the Revelle factor $\left( \gamma_{DIC} = \frac{DIC}{pCO_2} \frac{\partial pCO_2}{\partial DIC} \right)$.

The calculated Revelle factors in the Southern Ocean range between $\gamma_{DIC}$ ~ 12 − 15.5 with an average of $\gamma_{DIC}$ = 13.9±1.3. This justifies our use of $\gamma_{DIC}$ = 14 for the conversion of the equivalent solubility driven pCO$_2$ change to DIC throughout the analysis. As an addition, we now provide the uncertainty in this conversion as it translates into the temperature constraint, by using the upper and lower limits of the Revelle factor ($\gamma_{DIC}$ = 12 – 15.5) in the model framework. Below we show an example for observations in the Sub-Antarctic and

Antarctic zone, this shows that extremes of the Revelle factor values ($\gamma_{DIC}$ = 12 – 15.5) do not alter the phasing or magnitude of the relative controls of temperature or DIC on the seasonal cycle of pCO$_2$ (Fig 2).

[Figure]

**Figure. 2** Seasonal cycle of the rate change of surface total DIC for the Landschützer et al (2014) data product (black line) for the Landschützer et al (2014) data product and the estimated temperature driven DIC rate of change $\left(\frac{\partial DIC}{\partial t}\right)_{SST}$ shaded area, for monthly data given in

μmol kg$^{-1}$ month$^{-1}$ at the Sub-Antarctic zone i.e. Pacific Ocean (first column), Atlantic Ocean (second column) and Indian Ocean (third column). The dotted line shows the uncertainty boundaries for the Revelle factor extremes accounting to range in the Southern Ocean ($\gamma_{DIC}$ =

12 – 15.5).

**Reviewer** Finally, isn't the total DIC variability by definition the sum of the various components? So I'm not clear why the temperature driven component should ever be larger than the total. I find equation 3 and the discussion on 304-311 to be the most confusing part. We have an observed rate of change of DIC (which is never actually defined), which one would think would be the sum of the contributions from gas exchange, biological uptake

/remineralization and entrainment. But in this case, the index that is considered is that the total is either greater or less than one of these three components (whose physical meaning is nebulous). To confuse matters worse, we have a reference to "the total DIC seasonal cycle (dDIC/dt)" (306-307). Doesn't dX/dt imply an instantaneous rate-of-change that will itself vary over the annual cycle? I really do not understand what is being asserted here. (Also, the text should say something about exactly what sort of discretization was used in calculation of trends, e.g., does delta-X/delta-t for November represent a value for Nov. 1 based on a difference of October and November means, or is it something else? If this is the case, figure axes should indicate that calculated values are for the first of the month and not the mid- month.)

**Response**: We thank the reviewer for this comment, it was important to clarify this point.

We apologize for giving this impression. We, regretfully, neglected to provide an adequate description of how we separated the terms contributing to the total DIC surface layer changes. We have clarified this part in the revised manuscript. The total rate of change of

DIC $\left(\frac{\partial DIC}{\partial t}\right)_{Tot}$ in the surface layer consists of the contribution of air-sea exchanges, biological, vertical and horizontal transport-driven changes (eq. 1).

$$\left(\frac{\partial DIC}{\partial t}\right)_{Tot} = \left(\frac{\partial DIC}{\partial t}\right)_{air-sea} + \left(\frac{\partial DIC}{\partial t}\right)_{Bio} + \left(\frac{\partial DIC}{\partial t}\right)_{Vert} + \left(\frac{\partial DIC}{\partial t}\right)_{Hor} \qquad \text{(eq.1)}$$

In our method, we assumed that the horizontal term can be neglected because we used zonal means from medium resolution models

The discretized form of the total rate of change is written as:

$$\left(\frac{\partial DIC}{\partial t}\right)_{Tot} = \left(\frac{\Delta DIC}{\Delta t}\right)_{n,l} = \frac{DIC_{n+1,l} - DIC_{n,l}}{1\ month} \qquad \text{(eq. 2)}$$

where n is time in month, l is vertical level (in this case the surface, l=1). We here take the forward derivative such that November rate the is difference between December the 15[th]

and November the 15[th] , thus being centered at the interval between the months.

In order to compare the role of temperature in scalable terms with DIC in $pCO_2$ and $FCO_2$, we convert the instantaneous $pCO_2$ changes driven by solubility $\left(\text{using } \left(\frac{\partial pCO_2}{\partial DIC}\right)\frac{1}{pCO_2} = \right.$

$\left. 0.0423^o C^{-1} \text{ from Takahashi et al (1993)}\right)$ to an equivalent DIC change using the Revelle factor $\left(\frac{\ln(pCO_2)}{\ln(DIC)} \approx \gamma_{DIC}\right)$.

$$\left(\frac{\partial DIC}{\partial t}\right)_{SST} = \frac{DIC}{\gamma_{DIC} \times pCO_2}\left(\frac{\partial pCO_2}{\partial t}\right)_{SST} \qquad \text{(eq. 3)}$$

This equivalent DIC rate of change (eq. 2) driven by temperature (solubility) allow us to convert the influence of solubility into DIC units, which can then be directly compared with the other terms in eq. 1. We hope that this now clarifies the construction of our indicator metrics,

$M_{T-DIC} = \left| \left( \frac{\partial DIC}{\partial t} \right)_{SST} \right| - \left| \left( \frac{\partial DIC}{\partial} \right)_{Tot} \right|$         (eq. 4)

which is used to compare the estimated change of (equivalent) DIC as driven by temperature-controlled solubility with the actual DIC change simulated by the models and obtained from the observational data product. When $M_{T\text{-}DIC} > 0$ indicates that the $pCO_2$

variability is dominated by the rate of change of temperature and when $M_{T\text{-}DIC} < 0$ indicates that the $pCO_2$ variability is controlled mainly by DIC changes.

We have now revised manuscript to better reflect the underlying methodology.

**Reviewer**: The discussion of entrainment is also confusing and poorly connected to actual physical processes. Equation 4 does not have the units of a flux, but rather of a rate of change within the surface layer. The proper quantity here is not DIC concentration at MLD

(T+1) but rather the difference between DIC at MLD(T+1) and at MLD(T).

**Response:** The reviewer is correct in this regard and we made this correction in the revised text. The definition of entrainment at the base of the MLD has been improved using a more appropriate notation (please refer to the discretization in the answer above that is now included in the revised manuscript). Entrainment is physically considered as advection of preformed DIC at the base of the mixed layer. It is therefore based on the advection term.

$RE = U_e \left( \frac{\partial DIC}{\partial z} \right)_{MLD}$         (eq. 5)

$RE_n = \left( \frac{\Delta MLD_n}{\Delta t} \right) \left( \frac{\Delta DIC}{\Delta z} \right)_{n,MLD}$         (eq. 6)

$\Delta MLD = \frac{MLD_{n+1} - MLD_n}{1\,Month}$         (eq. 7)

$\left( \frac{\Delta DIC}{\Delta z} \right)_{n,MLD} = \frac{DIC_{n,MLD_{n+1}} - DIC_{n,MLD_n}}{\Delta z}$         (eq. 8)

In which Ue is an equivalent velocity based on the rate of change of the mixed layer depth.

This approximation of vertical entrainment is necessary as it is not possible to compute this term from the CMIP5 data because the vertical DIC distribution is only available as annual means. We clarified that we use the estimated rate of change of DIC at the base of the mixed to examine surface DIC changes driven by subsurface/bottom DIC changes. We also updated $F_{DIC}$ estimate with GLODAP version 2 dataset.

**Reviewer:** Why use only ten years of model output (124)? The results could be biased by internal variability; the more usual averaging period would be 20 or even 30 years. With a reference year of 2000, this would require using emissions scenarios, which is perhaps a reason not to do it, but the differences among scenarios are very small in 2005-2015

(because the scenarios are constructed precisely around the assumption that there is some inertia in human societies and abrupt changes are unlikely). I think the authors should (a)

pick one model, recalculate the results for 1990-2010 and 1985-2015, and estimate the potential error associated with aliasing of internal variability. And (b) if this error turns out to be large, repeat the calculation for the full suite of models.

**Response:**  The choice of the period was to match a period closest to the available observational data product (Landschützer et al (2014), 1998 – 2011). However, the reviewer is correct to highlight that we assumed that the seasonal cycle of $CO_2$ does not vary significantly on decadal timescales. We have now investigated this assumption for a few models (here we present HadGEM2-ES and CanESM2), comparing the mean seasonal cycle climatology of $FCO_2$ at the Sub-Antarctic zone for 30 years (1975 – 2005) and 10 years (1995

– 2005) of the historical scenario (Fig. 3).  It shows that the seasonal cycle of $FCO_2$ remain the small (R = 0.99) in both HadGEM2-ES and CanESM2 over 30 year.

We have now added this sentence in the method section: "The choice of the 10 year period was done to match the simulated $pCO_2$ values to the period of the observations. We tested the interannual variability of the seasonal cycle over a period longer than the reference 10

years for a few models and found no significant variation in the monthly standard deviation."

[Figure]

**Figure 3.** Compares the seasonal cycle of Sea-Air CO$_2$ fluxes over 30 year (1975 – 2005 and years (1995 – 2015). The shaded area shows the standard deviation.

**Reviewer:** The use of chlorophyll as a proxy is not really explained, when primary production and export production are generally available as model output fields. One might justify this by saying that observations are available only for chlorophyll, but this should be stated explicitly. There are also observation-based estimates of primary production available (see below Terminology).

**Response:** Our initial choice for chlorophyll was because of the availability of an observational data product for comparison and that it was available for most models (9 out of 10) in the CMIP5 portal. In the revised manuscript, we added net primary production (NPP), carbon export and oxygen for the models with available data. This addition of NPP, carbon export and oxygen was indeed useful in examining impact of biological driven DIC

changes in particular it helped isolate/constrain the role of primary production on the seasonal cycle of pCO$_2$.

**Reviewer**: I think the authors should acknowledge that the $FCO_2$ data product is not really 'observed' in the sense that $pCO_2$ is. I think they should compare modelled and observed $pCO_2$, and then discuss what this means for modelled estimates of $CO_2$ flux, without referring to the Landschuter $FCO_2$ estimates as observations. $CO_2$ fluxes in models and data products like this are actually quite different conceptually. When you estimate CO2 flux from observed pCO2, the errors in the flux are a linear function of errors in wind speed (or $u^2$, assuming a quadratic parameterization) and the piston velocity. In numerical models, $pCO_2$ and DIC self-regulate to dampen these errors when monthly averaged fluxes are considered, e.g., if both the wind speed and the DIC are too large, the enhanced outgassing flux will reduce the $pCO_2$ and DIC error. Higher wind speed or piston velocity will tend to drive $pCO_2$ towards atmospheric, and not necessarily towards the 'correct' value if over- or undersaturation exists, so there is no straightforward way to correct for this difference. But I think that the authors should acknowledge that it exists, and that in comparing modelled and 'observed' fluxes they are to some degree comparing apples and oranges. (With regard to point (4) above, if there is no observed primary production, there is no observed $CO_2$ flux either: both of these are extrapolated from the primary observed field using models of unknown accuracy.)

**Response**: It is indeed true that $FCO_2$ observations we use are estimates, data products derived from empirical methods of gap filling. We now refer to them in this answer and in the revised manuscript as "data products". We tried to make the point much clearer in the revised manuscript and also provided the uncertainty as given by Landschützer et al (2014) and used the interannual standard deviation over 14 years when comparing the seasonal cycle. However, it also important to note that the majority of our analysis is based on $pCO_2$ estimates rather than $FCO_2$ in particular. Thus, we do acknowledge the role of wind in providing the kinematic forcing for sea-air fluxes and although the wind influences the magnitude of the fluxes, the direction is determined by delta $pCO_2$, which is here considered the primary driver of the seasonal cycle of $FCO_2$. We are mindful that differences in the parameterization of the Sea-Air interactions and wind products chosen are likely to affect the resulting $FCO_2$ (Feely et al., 2004; Swart et al., 2014).

The uncertainty in the data products used for the assessment is an important point that we neglected to highlight in our first version of the manuscript. We now aknowledge the limitations of the data products and have further addressed the issue of uncertainties as requested by other reviewers as well.

We have examined the uncertainty of our reference data set by comparing Landschützer et al., 2014 data product with the more recent Gregor et al (2017a) data product, which uses

Support Vector Regression (SVR) and Random Forest Regression (RFR), as well as Takahashi et al  (2009) to derive a seasonal climatology $pCO_2$ in the SAZ and AZ of the Southern Ocean in the supplementary material (Fig. 5). Part of the reason we focus on $pCO_2$ in these data products instead of $FCO_2$ is firstly, because Gregor et al., (2017a) only focuses on fugacity and $pCO_2$. Thus we mindful that the choice of wind product and tranfer velocity constant in computing $FCO_2$ is likely to increase the level of uncertainty for the compared data products (Swart et al., 2014).  Secondly, while we evaluate $FCO_2$ biases as the final aim of the paper, the major part of our diagnostic analysis is based on $pCO_2$ rather $FCO_2$. Fig. 4 below shows the climatology of the seasonal cycle of $pCO_2$ in the Sub-Antarctic zone and Antarctic zone with interannual standard deviation between 1998 – 2011. All three datasets mostly agree in the phasing of the seasonal cycle of $pCO_2$ in the Sub-Antarctic, but show significant differences in the magnitude. Takahashi et al. (2009) shows an amplified impact of primary production on $pCO_2$ in summer.  We see this as a bias in the Takahashi et al., (2009) dataset arising from a period when the space – time coverage of $pCO_2$ observations was still limited and strongly biased towards summer. In the Antarctic zone all three observationally-based data products agree in both phasing and amplitude. At this stage it is not clear whether this agreement is due to all the methods being equally exposed to the same limited observations or if it is due to a more marked $CO_2$ seasonal cycle in the Antarctic zone (relative to Sub-

Antarctic) that can be captured with less observations

[Figure]

**Fig. 4** pCO$_2$ (µatm) spatial (climatology) and seasonal cycle differences in Landschützer et al (2014) (L14), Gregor et al (2017) (G17), Takahashi et al (2009) (T09) datasets in the Southern Ocean. The seasonal cycle climatology of pCO$_2$ in the Sub-Antarctic and Antarctic zone is based on the period 1998 – 2011. The shaded areas show the standard deviation of the interannual variability of the seasonal cycle for this period. The uncertninity in the correlation coeffecient is based on the correlation coeffient of the mean plus stardard deviations seasonal cycle(s).

**Reviewer:** I would like to see some discussion of the possibility that the apparently greater temperature control in the Pacific sector (259-263) is a real effect that arises from iron limitation. Because terrestrial sources of iron are much greater in Atlantic sector and the western half of the Indian sector (see e.g., Graham et al 2015 DSR I 104: 9; Tagliabue et al., 2012 Biogeosciences 9: 2333), it seems logical that the effect of seasonal biological drawdown on pCO$_2$ would be greater than in the Pacific and in the eastern half of the Indian sector. These regions also overlap the regions where the wind speed and the amplitude of its annual cycle are greatest (e.g., Trenberth et al., 1990, JPO 20: 1742), which will also tend to reduce the influence of biological uptake

**Response:** Thank you for this important suggestion, it is indeed likely that differences in wind, iron supply, primary productivity and MLD across the basins are most likely responsible for contrasting variability in the three basins. We have added this discussion in the revised manuscript;

"The observed differences in the seasonal cycle of $FCO_2$ across the three basins is likely due to differences in the basin properties of the Southern Ocean.  Recent studies have highlighted significant basin scale differences in $pCO_2$ and $FCO_2$ ascribed to large-scale differences in temperature (Landschutzer et al., 2015), winds (Gregor et al., 2017b and primary production as reflected in surface ocean phytoplankton biomass (Thomalla et al.,

2011).  The relatively higher chlorophyll biomass (Graham et al., 2015; Thomalla et al., 2011)

in the Atlantic Ocean, is likely linked to lower wind speeds (Trenberth et al., 1990) and higher supplies of iron from continental shelves (Thomalla et al., 2011;  Boyd and Ellwood,

2010; (Tagliabue et al., 2014)2; 2014). This is likely associated with longer periods of shallower MLD (Dec – Mar, Fig. 7), which favor sustained primary production leading to a stronger $CO_2$ sink in the Atlantic Ocean with respect to the Indian and Pacific Ocean (Figs.

3a-c; 6a-c).  In contrast, shorter periods of shallow MLD and lower iron concentrations in the

Pacific Ocean, as pointed out by Tagliabue et al. (2012), likely account for lower chlorophyll biomass and stronger thermal control of the seasonal cycle of $pCO_2$ and $FCO_2$ (Fig. 6b, Fig. 3

here).  In the Indian Ocean stronger wind speeds are likely responsible for the early deepening of the MLD (Fig. 6c), and thus chlorophyll biomass are lower (Fig. 9). In the Indian

Ocean, stronger wind speeds (Trenberth et al., 1990) are likely responsible for the early deepening of the MLD (Fig. 7c), limiting primary production and lower rates of change of temperature (Fig. 5c), ultimately resulting in a relatively constant $FCO_2$ for about half the year (Dec – Jun). Our plots indicate that CMIP5 models mostly don't show these basin- specific features highlighted in observational products (Landschutzer et al., 2015; Gregor et al., 2017a and Thomalla et al., 2011) with the exception of three group B models (i.e.

CESM1-BGC, CanESM2 and CMCC-CESM) in the Indian Ocean (Fig. 2, 3 a-c).  This poses a challenge to the new generation of Earth Systems Models in CMIP6"

**References**

Dickson, A. G. and Millero, F. J.: A comparison of the equilibrium constants for the dissociation of carbonic acid in seawater media, Deep Sea Res. Part A, Oceanogr. Res. Pap., 34(10), 1733–1743, doi:10.1016/0198-0149(87)90021-5, 1987.

Feely, R. A., Wanninkhof, R., McGillis, W., Carr M. E and Cosca, C.: Effects of wind speed and gas exchange parameterizations on the air-sea $CO_2$ fluxes in the equatorial Pacific Ocean, J. Geophys. Res., 109(C8), C08S03, doi:10.1029/2003JC001896, 2004.

Graham, R. M., De Boer, A. M., van Sebille, E., Kohfeld, K. E. and Schlosser, C.: Inferring source regions and supply mechanisms of iron in the Southern Ocean from satellite chlorophyll data, Deep. Res. Part I Oceanogr. Res. Pap., 104, 9–25, doi:10.1016/j.dsr.2015.05.007, 2015.

Gregor, L., Kok, S. and Monteiro, P. M. S.: Empirical methods for the estimation of Southern Ocean CO2: Support vector and random forest regression, Biogeosciences, 14(23), 5551–5569, doi:10.5194/bg-14-5551-2017, 2017a.

Gregor, L., Kok, S. and Monteiro, P. M. S.: Interannual drivers of the seasonal cycle of CO2 fluxes in the Southern Ocean, Biogeosciences Discuss., (September), 1–28, doi:10.5194/bg-2017-363, 2017b.

Landschützer, P., Gruber, N., Bakker, D. C. E. and Schuster, U.: Recent variability of the global ocean carbon sink, Global Biogeochem. Cycles, 28(9), 927–949, doi:10.1002/2014GB004853, 2014.

Mehrbach, C., Culberson, C. H., Hawley, J. E. and Pytkowitz, R. M.: Measurement of the aparent dissociation constants of carbonic acid in seawater at atmospheric pressure, Limnol. Oceanogr., 18(6), 897–907, doi:10.4319/lo.1973.18.6.0897, 1973.

Mongwe, N. P., Chang, N. and Monteiro, P. M. S.: The seasonal cycle as a mode to diagnose biases in modelled $CO_2$ fluxes in the Southern Ocean, Ocean Model., 106, 90–103, doi:10.1016/j.ocemod.2016.09.006, 2016.

Pierrot, D. E. Lewis, and D. W. R. Wallace. 2006. MS Excel Program Developed for

CO2 System Calculations. ORNL/CDIAC-105a. Carbon Dioxide Information Analysis Center,

Oak Ridge National Laboratory, U.S. Department of Energy, Oak Ridge, Tennessee.

doi: 10.3334/CDIAC/otg.CO2SYS_XLS_CDIAC105a

Swart, N. C., Fyfe, J. C., Saenko, O. A. and Eby, M.: Wind-driven changes in the ocean carbon sink, Biogeosciences, 11(21), 6107–6117, doi:10.5194/bg-11-6107-2014, 2014.

Tagliabue, A., Mtshali, T., Aumont, O., Bowie, A. R., Klunder, M. B., Roychoudhury, A. N. and

Swart, S.: A global compilation of dissolved iron measurements: Focus on distributions and processes in the Southern Ocean, Biogeosciences, 9(6), 2333–2349, doi:10.5194/bg-9-2333-

2012, 2012.

Tagliabue, A., Sallée, J.-B., Bowie, A. R., Lévy, M., Swart, S. and Boyd, P. W.: Surface-water iron supplies in the Southern Ocean sustained by deep winter mixing, Nat. Geosci.,

7(March), 314–320, doi:10.1038/NGEO2101, 2014.

Takahashi, T., Sutherland, S. C., Wanninkhof, R., Sweeney, C., Feely, R. a., Chipman, D. W.,

Hales, B., Friederich, G., Chavez, F., Sabine, C., Watson, A., Bakker, D. C. E., Schuster, U.,

Metzl, N., Yoshikawa-Inoue, H., Ishii, M., Midorikawa, T., Nojiri, Y., Körtzinger, A., Steinhoff,

T., Hoppema, M., Olafsson, J., Arnarson, T. S., Tilbrook, B., Johannessen, T., Olsen, A.,

Bellerby, R., Wong, C. S., Delille, B., Bates, N. R. and de Baar, H. J. W.: Climatological mean and decadal change in surface ocean pCO2, and net sea-air CO2 flux over the global oceans,

Deep. Res. Part II Top. Stud. Oceanogr., 56(8–10), 554–577, doi:10.1016/j.dsr2.2008.12.009,

2009.

Thomalla, S. J., Fauchereau, N., Swart, S. and Monteiro, P. M. S.: Regional scale characteristics of the seasonal cycle of chlorophyll in the Southern Ocean, Biogeosciences,

8(10), 2849–2866, doi:10.5194/bg-8-2849-2011, 2011.

Trenberth, K. E., Large, W. G. and Olson, J. G.: The Mean Annual Cycle in Global Ocean Wind

Stress, J. Phys. Oceanogr., 20(11), 1742–1760, doi:10.1175/1520-

0485(1990)020<1742:TMACIG>2.0.CO;2, 1990.

Young, I. R.: Seasonal variability of the global ocean wind and wave climate, Int. J. Climatol.,

19(9), 931–950, doi:10.1002/(SICI)1097-0088(199907)19:9<931::AID-JOC412>3.0.CO;2-O,

1999.

---

## Author Response (AR1)

[revised manuscript text omitted]

V2 Changes 2018/3/6 6:53 PM
Formatted ... [37]
V2 Changes 2018/3/6 6:53 PM
V2 Changes 2018/3/6 6:53 PM
V2 Changes 2018/3/6 6:53 PM
Formatted ... [40]
V2 Changes 2018/3/6 6:53 PM
Formatted ... [42]
V2 Changes 2018/3/6 6:53 PM
Inserted Cells ... [43]
V2 Changes 2018/3/6 6:53 PM
V2 Changes 2018/3/6 6:53 PM
Formatted ... [41]
V2 Changes 2018/3/6 6:53 PM
Inserted Cells ... [44]
V2 Changes 2018/3/6 6:53 PM
Inserted Cells ... [45]
V2 Changes 2018/3/6 6:53 PM
Formatted ... [47]
V2 Changes 2018/3/6 6:53 PM
Formatted ... [48]
V2 Changes 2018/3/6 6:53 PM
Formatted ... [46]
V2 Changes 2018/3/6 6:53 PM
Formatted ... [49]
V2 Changes 2018/3/6 6:53 PM
Formatted ... [50]
V2 Changes 2018/3/6 6:53 PM
V2 Changes 2018/3/6 6:53 PM
Formatted ... [53]
V2 Changes 2018/3/6 6:53 PM
V2 Changes 2018/3/6 6:53 PM
Formatted ... [51]
V2 Changes 2018/3/6 6:53 PM
Formatted ... [52]
V2 Changes 2018/3/6 6:53 PM
Formatted ... [54]
V2 Changes 2018/3/6 6:53 PM
Formatted ... [55]
V2 Changes 2018/3/6 6:53 PM
Formatted ... [58]
V2 Changes 2018/3/6 6:53 PM
V2 Changes 2018/3/6 6:53 PM
Formatted ... [56]
V2 Changes 2018/3/6 6:53 PM
Formatted ... [57]
V2 Changes 2018/3/6 6:53 PM
Formatted ... [63]
V2 Changes 2018/3/6 6:53 PM
V2 Changes 2018/3/6 6:53 PM
Formatted ... [59]
V2 Changes 2018/3/6 6:53 PM
Formatted ... [60]
V2 Changes 2018/3/6 6:53 PM
Formatted ... [61]
V2 Changes 2018/3/6 6:53 PM
Formatted ... [62]
V2 Changes 2018/3/6 6:53 PM
Formatted ... [35]
V2 Changes 2018/3/6 6:53 PM
Formatted ... [36]

[revised manuscript text omitted]

---

## Author Response (AR2)

**Rebuttal report**

We thank the reviewers for the careful thought and suggestions. We hope that we have been able to address all these points below and in the text. All grammar corrections were addressed as suggested except for cases where the text was removed completed, all text changes are shown in the attached track changes document. The manuscript was also further refined in grammar and syntax as suggested.

**Reviewer #1**

**Reviewer:**

Table 1 – only 7 of the 10 models are described. Please add GFDL-ESM2M, HadGEM2-ES and MRI-ESM

**Response:**

We apologise for the missing models, it was a technical error on the submitted manuscript, all models are now added.

All technical and editorial suggestions were addressed.

**Reviewer #2**

**Reviewer:**

I think the authors have done a good job of responding to the specific comments made in the previous reviews. However, I still find the logic of the method difficult to follow in places, and the general tone of the paper is more pessimistic about the models' performance than I think the data warrant. I would like these authors to find a way to see the cup as half full rather than half empty; I don't think their results are as damning of the models' performance as their text implies (see e.g. point 2 below).

**Response:**

We thank the reviewer making these points. The methods section has been further clarified, particularly explaining more explicitly the basis for comparing the equivalent DIC $(dDIC/dt)_{SST}$ and total DIC $(FDIC/dt)_{Tot}$ changes. It should be emphasized that equivalent DIC $(dDIC/dt)_{SST}$ is a "synthetic" variable in the sense that it does not contribute to the "real" DIC budget $(FDIC/dt)_{Tot}$. It's purpose is to scale the temperature impact on solubility of CO2 in DIC-equivalent units.    Below we also clarify why lumping together the biological and entrainment terms when comparing the total DIC changes with the temperature equivalent DIC is valid  as well as  why this methodology should hold even for cases when the biological and entrainment terms are opposing each other equally.

We tried to be as objective as possible in interpreting of our findings, avoiding a negative perspective view of CMIP5 models. We do agree that CMIP5 models show a strong improvement from previous generations and, in pointing out current limitations and biases, we were not aiming  at showing how bad models are, but highlighting considerations that pointed  towards potential model improvements.

**Reviewer:**

1) I still find the logic underlying equation (5) opaque. If we have

$X\_TOT = X\_T + X\_B + X\_E$

where T, B, and E indicate temperature, biology and entrainment, it isn't obvious to me why

$|X\_T| - |X\_TOT| < 0$ or $> 0$

is a useful index of whether or not T, B, or E are the dominant term. And what happens if it is identically zero? For example, if $X\_B = -1$, and $X\_E = +1$, then $X\_TOT = X\_T$. If e.g.,

$X\_T = 3*X\_E$, then in this case $X\_T$ is clearly the dominant term. But this index does not consider it so, and if $X\_T$ were only slightly larger or smaller, it would flip the index into one realm or the other. As I see it, $X\_B$ will normally be negative and $X\_E$ positive, while

$X\_T$ can be either. This index appears to depend rather strongly on the sign of $X\_T$, which has nothing to do with the relative magnitudes of the terms. Why not just make an index that directly addresses the relative magnitudes, like $|X\_T|/|X\_B + X\_E|$? This at least has the desirable property of increasing symmetrically with either positive or negative $X\_T$.

**Response:**

We thank the reviewer for raising this point; it shows that a further clarification of our methodology was necessary.

The reviewer raises an interesting  alternative to present our results, however, we hope that we can, by further clarifying our thinking, show that this suggestion has its own complexities.

Firstly, the temperature-linked equivalent DIC term $(dDIC/dt)_{SST}$  is a not a contributing term to the total DIC changes Eq. 1, but a construct  of the temperature driven $pCO_2$

changes (solubility) converted to equivalent DIC units.  Since temperature does not influence DIC directly, but $pCO_2$, it was necessary to convert solubility-linked $pCO_2$

changes into comparable  units of DIC, and hence the DIC equivalence.  It is an abstract term but of great value because it scales the temperature impact on pCO2 to DIC units - essentially it reflects the amount by which DIC would have to change to have the same impact on pCO2 as the temperature change but does not contribute to the DIC budget in a direct sense.  This is different to the real impact of temperature on pCO2 which then results in changes to air-sea fluxes that impact on $DIC_{Total}$.  This is incorporated into the

$DIC_{Total}$ term. .

Consequently, our approach (Eq. 5) is independent of the sign and magnitude of the contributing terms in Eq. 1 (entrainment and biological terms with a much smaller /

slower contribution from air-sea fluxes).  We are quantifying the relative magnitudes of the temperature and total DIC changes as drivers of pCO2 variability.  This enables us to reflect on the relative dominance of the drivers.  This is important because it highlights a tipping point between SST and DIC control, which has implications for the climate sensitivity of the models.

**Reviewer:**

When I look at Supplemental Figure 8, it makes me think that the models are actually doing a fairly good job. The periods where X_T dominates are short and confined to the spring and fall transitions. The general seasonal pattern is reproduced by almost all of the models for almost all of the regions. Certainly there are models where the DIC- driven seasonal variation of X_TOT seems too large. But generally the models reproduce the observed pattern fairly well (although I admit I don't understand why X_TOT is represented as a single line with a sign for each month, and X_T as a cloud symmetrical around zero).

**Response:**
Indeed when comparing absolute magnitudes of X_T and X_TOT, the models reflect the
phasing of the variability reasonably well. . However, there are two issues that need to
be considered. Firstly, that SST or DIC control is not a gradual scale but a tipping point
problem. It's one or the other so even a relatively close magnitude may lead to a bias in
respect of how the model reflects the climate sensitivity of the carbon cycle. Secondly,
is that the spring/fall temperature dominance in CMIP5 models is critical to diagnosing
the $pCO_2$ and $FCO_2$, biases against observational product. This is exactly the point we are
making with this figure. Though Landschützer et al (2014) and CMIP5 models only differ
during this two seasons, the separations are significant in the seasonal cycle of pCO2
because of the marginal differences in SST and DIC control; it results in the out-gassing
$CO_2$ biases in late-summer (due to warming) and ingassing due to rapid early autumn
cooling. This, we think, causes the main bias against Landschützer et al (2014_) data
product.
The reason for shaded symmetry of X_T in Fig. S8 is because this figure aims to give two
pieces of information, (i.) to show where periods X_TOT >(or <) X_T and elucidate on a
possible driver of the instantaneous X_TOT changes. Because X_T is only driven by
temperature changes, X_T is binary so it can symmetric. However X_TOT can be driven
by biology (- primary production and + respiration) and entrainment, and thus the sign
of total the DIC changes provide useful information in diagnosing driver of the
instantaneous DIC changes.
**Reviewer:**
What would happen if you added a fourth line to Fig. 3 representing an ensemble of ALL
of the models? I suspect that it would agree with the observations much better than
either of these two somewhat arbitrary groups alone (see e.g., Lambert and Boer (2001)
Climate Dynamics 17:83).
**Response:**
The reviewer is correct, an additional fourth line representing the ensemble of all modes
show a seasonal cycle that agrees more with Landschützer et al (2014). However though
we did this test in our analysis, it was excluded in the manuscript mainly because our
aim here is to explain the mechanisms responsible for model-observations biases and
the ensemble of all models is not useful for this regard. Notwithstanding these, other
than the fact that it is known that an ensemble of all models is more comparable to available observations estimates (e.g. Anav et al., 2013; Boer and Lambert, 2001), mechanisms of this ensemble are hard to explain. It is also not clear whether the long- term simulation of FCO2 based this ensemble (all models) is reliable.  In particular it is not clear whether long-term biases of individual models will significantly distort the trend from the "true future trend" or alternatively continue to average out towards  the observed trend.  Thus a good comparison with observations might be misleading. his might be intuitively useful in some cases but because we are not making a point about how CMIP5 models ensembles converge to observations, but evaluating the mechanistic basis for the biases, we thought this fourth line was not necessary.

**Reviewer:**

The assumption that temperature-driven changes in $pCO_2$ can be converted to an

"equivalent DIC" change implies equilibration with the atmosphere by gas exchange. I

don't think this a bad assumption to make in context of this kind of analysis, but I still don't think that it is stated explicitly enough (e.g., 199-206). The change in $pCO_2$ with

SST is purely a result of temperature driven changes in solubility (and to small degree, speciation): DIC does not change as a result of these processes.

**Response:**

As communicated above, the DIC equivalent temperature we are referring to here is purely a scaling of the solubility $CO_2$ changes to DIC units, which does not contribute to the DIC budgets in a direct sense. So, in fact the opposite holds to the equilibration assumption that is proposed.  When $DIC_T$ is > $DIC_{Tot}$ and $pCO_2$ is controlled by temperature the net effect is a strengthening disequilibrium at the air-sea interface which results in a divergence between observed and modelled $pCO_2$.  As mentioned earlier, the air-sea flux adjustments are in the $DIC_{Tot}$ term and the reason why temperature driven disequilibrium persistes is because the rate at which temperature adjusts $pCO_2$ >> rate of air-sea flux equilibrium restoration.

**Reviewer:**

The assumption that the horizontal terms are negligible is probably OK as a first approximation, but I would recommend that the authors qualify this assumption by noting that, at least in the subantarctic zone, (a) there is a latitudinal gradient in DIC

concentration, (b) there is a wind-driven equatorward flow (Ekman transport), and (c)

in some regions there is a strong seasonal cycle in the wind stress. a+b+c implies that there is a seasonal cycle in the divergence of the horizontal transport.

**Response:**

Thank you  for the point, in the revised manuscript with clarified as suggested in the revised manuscript.

**Terminology**

All terminology corrections were addressed as suggested.

**Reviewer 3**

**Reviewer:**

The authors added a valuable synthesis chapter. There they have strengthened a second item, which focuses on the simulated homogeneity of the fluxes in the different basins differing from observations. It is common practice to answer every question of remark of the reviewer.  On  the other hand the text still does not give information about model characteristics which may be responsible for exaggerated SST or DIC sensitivities on

FCO2 in the different models. This has not been done. In some cases the item has been ignored, in some cases it has been accounted for. Please discuss with the reviewer and, in case if text change, give a note there and how the test has been improved. The manuscript is still written very sloppily

**Response:**

We apologize that not all remarks were addressed in the submitted manuscript. The manuscript has been improved in presentation for grammar and flow of structure.

We have added a new paragraph at the end of the discussion section that explicitly addresses the question on what model characteristics may be responsible for exaggerated seasonal cycles for the rates of change of SST .

" The cause of differences in the seasonal rates of SST change in group-SST models remains a subject of ongoing research. The Southern Ocean is a part of the global ocean (upwelling) where earth systems models show a persistent warming  SST bias (Hirahara et al., 2014). Several studies point to highlight potential explanations but the main reasons remain uncertain.   For example, CMIP5 models differences in the magnitude and meridional location of the peak of wind speeds in the Southern Ocean (Bracegirdle et al., 2013) and MLD (Meijers, 2014; Sallée et al., 2013) may be such that the net effect on surface turbulence and mixing leads to these amplified surface temperature rates.

Other known CMIP5 model' biases that may contribute include; heat fluxes and storage (Frölicher et al., 2015) as well as sea-ice dynamics (Turner et al., 2013).

Notwithstanding these, investigation of the reasons for sources of these dSST/dt biases is out of the scope of this study. Our aim here is to show that understanding biases in the drivers of $pCO_2$ (DIC and SST) at the seasonal scale is necessary to understand differences in the seasonal cycle of $FCO_2$ between models and observational products.

However we recommend that the mechanistic basis for the differences the seasonal rates of warming and cooling be a matter of urgent investigated further"

**Reviewer:**

Fig. 10 shows new patterns compared with the old manuscript, why?

Fig. 10 legend: still wrong units

**Response:**

Figure 10 was corrected from the previous version hence the some differences in the putterns,, there was an error on the equation we used to calculate the fluxes. This was pointed out by reviewer two, and thus we made a correction and instead calculate the rates of change of DIC at the base of the MLD (Eq. 6), this we now we use to infer entrainment.

The units in Fig. 10 are however correct based on Eq. 6 - 8.

Fig. 10a-f shows the estimated entrainment of rate given in umol/kg month  and Fig.

10d-l shows the vertical DIC gradients given in umol/kg m

**Terminology**

All terminology and reference corrections were addressed as suggested.

[revised manuscript text omitted]

V2 Changes 2018/4/17 11:06 AM
Formatted ... [65]
V2 Changes 2018/4/17 11:06 AM
Formatted Table ... [66]
V2 Changes 2018/4/17 11:06 AM
Formatted ... [67]
V2 Changes 2018/4/17 11:06 AM
Formatted ... [68]
V2 Changes 2018/4/17 11:06 AM
Formatted ... [69]
V2 Changes 2018/4/17 11:06 AM
Formatted ... [79]
V2 Changes 2018/4/17 11:06 AM
Formatted ... [70]
V2 Changes 2018/4/17 11:06 AM
Formatted ... [71]
V2 Changes 2018/4/17 11:06 AM
Formatted ... [76]
V2 Changes 2018/4/17 11:06 AM
Formatted ... [72]
V2 Changes 2018/4/17 11:06 AM
Formatted ... [73]
V2 Changes 2018/4/17 11:06 AM
Formatted ... [74]
V2 Changes 2018/4/17 11:06 AM
Formatted ... [75]
V2 Changes 2018/4/17 11:06 AM
Formatted ... [77]
V2 Changes 2018/4/17 11:06 AM
Formatted ... [78]
V2 Changes 2018/4/17 11:06 AM
Formatted ... [80]
V2 Changes 2018/4/17 11:06 AM
Formatted ... [81]
V2 Changes 2018/4/17 11:06 AM
Formatted ... [82]
V2 Changes 2018/4/17 11:06 AM
Formatted ... [83]
V2 Changes 2018/4/17 11:06 AM
Formatted ... [84]
V2 Changes 2018/4/17 11:06 AM
Formatted ... [85]
V2 Changes 2018/4/17 11:06 AM
Formatted ... [86]
V2 Changes 2018/4/17 11:06 AM
Formatted ... [87]
V2 Changes 2018/4/17 11:06 AM
Formatted ... [88]
V2 Changes 2018/4/17 11:06 AM
Formatted ... [89]
V2 Changes 2018/4/17 11:06 AM
Formatted ... [90]
V2 Changes 2018/4/17 11:06 AM
Formatted ... [91]
V2 Changes 2018/4/17 11:06 AM
Formatted ... [92]
V2 Changes 2018/4/17 11:06 AM
Formatted ... [93]
V2 Changes 2018/4/17 11:06 AM
Formatted ... [94]
V2 Changes 2018/4/17 11:06 AM
Formatted ... [95]
V2 Changes 2018/4/17 11:06 AM
Formatted ... [96]
V2 Changes 2018/4/17 11:06 AM
Formatted ... [97]
V2 Changes 2018/4/17 11:06 AM
Formatted ... [98]
V2 Changes 2018/4/17 11:06 AM
Formatted ... [99]
V2 Changes 2018/4/17 11:06 AM
Formatted ... [100]
V2 Changes 2018/4/17 11:06 AM
Formatted ... [101]
V2 Changes 2018/4/17 11:06 AM
Formatted ... [102]
V2 Changes 2018/4/17 11:06 AM
Formatted ... [103]

[revised manuscript text omitted]

V2 Changes 2018/4/17 11:06 AM
V2 Changes 2018/4/17 11:06 AM
V2 Changes 2018/4/17 11:06 AM
Formatted ... [170]
V2 Changes 2018/4/17 11:06 AM
V2 Changes 2018/4/17 11:06 AM
V2 Changes 2018/4/17 11:06 AM
V2 Changes 2018/4/17 11:06 AM
Formatted ... [171]
V2 Changes 2018/4/17 11:06 AM
V2 Changes 2018/4/17 11:06 AM
V2 Changes 2018/4/17 11:06 AM
V2 Changes 2018/4/17 11:06 AM
V2 Changes 2018/4/17 11:06 AM
V2 Changes 2018/4/17 11:06 AM
V2 Changes 2018/4/17 11:06 AM
V2 Changes 2018/4/17 11:06 AM
Formatted ... [172]
V2 Changes 2018/4/17 11:06 AM
V2 Changes 2018/4/17 11:06 AM
Formatted ... [173]
V2 Changes 2018/4/17 11:06 AM
V2 Changes 2018/4/17 11:06 AM
Formatted ... [174]
V2 Changes 2018/4/17 11:06 AM
V2 Changes 2018/4/17 11:06 AM
V2 Changes 2018/4/17 11:06 AM
Formatted ... [175]
V2 Changes 2018/4/17 11:06 AM
V2 Changes 2018/4/17 11:06 AM

[revised manuscript text omitted]

V2 Changes 2018/4/17 11:06 AM
Formatted ... [227]

V2 Changes 2018/4/17 11:06 AM

V2 Changes 2018/4/17 11:06 AM
Formatted ... [228]

V2 Changes 2018/4/17 11:06 AM

V2 Changes 2018/4/17 11:06 AM
Formatted ... [229]

V2 Changes 2018/4/17 11:06 AM

V2 Changes 2018/4/17 11:06 AM
Formatted ... [230]

V2 Changes 2018/4/17 11:06 AM

V2 Changes 2018/4/17 11:06 AM
Formatted ... [231]

V2 Changes 2018/4/17 11:06 AM

V2 Changes 2018/4/17 11:06 AM
Formatted ... [232]

V2 Changes 2018/4/17 11:06 AM

V2 Changes 2018/4/17 11:06 AM
Formatted ... [233]

V2 Changes 2018/4/17 11:06 AM

V2 Changes 2018/4/17 11:06 AM
Formatted ... [234]

V2 Changes 2018/4/17 11:06 AM

V2 Changes 2018/4/17 11:06 AM
Formatted ... [235]

V2 Changes 2018/4/17 11:06 AM

V2 Changes 2018/4/17 11:06 AM
Formatted ... [236]

V2 Changes 2018/4/17 11:06 AM
Formatted ... [237]

V2 Changes 2018/4/17 11:06 AM

V2 Changes 2018/4/17 11:06 AM
Formatted ... [238]

V2 Changes 2018/4/17 11:06 AM

V2 Changes 2018/4/17 11:06 AM
Formatted ... [239]

V2 Changes 2018/4/17 11:06 AM

V2 Changes 2018/4/17 11:06 AM
Formatted ... [240]

V2 Changes 2018/4/17 11:06 AM

V2 Changes 2018/4/17 11:06 AM

V2 Changes 2018/4/17 11:06 AM
Formatted ... [241]

V2 Changes 2018/4/17 11:06 AM
Formatted ... [242]

V2 Changes 2018/4/17 11:06 AM

V2 Changes 2018/4/17 11:06 AM
Formatted ... [243]

V2 Changes 2018/4/17 11:06 AM

V2 Changes 2018/4/17 11:06 AM
Formatted ... [244]

V2 Changes 2018/4/17 11:06 AM

V2 Changes 2018/4/17 11:06 AM
Formatted ... [245]

V2 Changes 2018/4/17 11:06 AM

V2 Changes 2018/4/17 11:06 AM
Formatted ... [246]

V2 Changes 2018/4/17 11:06 AM

[revised manuscript text omitted]

V2 Changes 2018/4/17 11:06 AM
Formatted ... [285]
V2 Changes 2018/4/17 11:06 AM
V2 Changes 2018/4/17 11:06 AM
Formatted ... [286]
V2 Changes 2018/4/17 11:06 AM
V2 Changes 2018/4/17 11:06 AM
Formatted ... [287]
V2 Changes 2018/4/17 11:06 AM
V2 Changes 2018/4/17 11:06 AM
Formatted ... [288]
V2 Changes 2018/4/17 11:06 AM
V2 Changes 2018/4/17 11:06 AM
Formatted ... [289]
V2 Changes 2018/4/17 11:06 AM
V2 Changes 2018/4/17 11:06 AM
Formatted ... [290]
V2 Changes 2018/4/17 11:06 AM
V2 Changes 2018/4/17 11:06 AM
Formatted ... [291]
V2 Changes 2018/4/17 11:06 AM
V2 Changes 2018/4/17 11:06 AM
Formatted ... [292]
V2 Changes 2018/4/17 11:06 AM
V2 Changes 2018/4/17 11:06 AM
Formatted ... [293]
V2 Changes 2018/4/17 11:06 AM
V2 Changes 2018/4/17 11:06 AM
Formatted ... [294]
V2 Changes 2018/4/17 11:06 AM
V2 Changes 2018/4/17 11:06 AM
Formatted ... [295]
V2 Changes 2018/4/17 11:06 AM
V2 Changes 2018/4/17 11:06 AM
Formatted ... [296]
V2 Changes 2018/4/17 11:06 AM
V2 Changes 2018/4/17 11:06 AM
Formatted ... [297]
V2 Changes 2018/4/17 11:06 AM
V2 Changes 2018/4/17 11:06 AM
Formatted ... [298]
V2 Changes 2018/4/17 11:06 AM
V2 Changes 2018/4/17 11:06 AM
Formatted ... [299]
V2 Changes 2018/4/17 11:06 AM
V2 Changes 2018/4/17 11:06 AM
Formatted ... [301]
V2 Changes 2018/4/17 11:06 AM
V2 Changes 2018/4/17 11:06 AM
Formatted ... [302]
V2 Changes 2018/4/17 11:06 AM
V2 Changes 2018/4/17 11:06 AM
Formatted ... [303]
V2 Changes 2018/4/17 11:06 AM
V2 Changes 2018/4/17 11:06 AM
Formatted ... [304]
V2 Changes 2018/4/17 11:06 AM
V2 Changes 2018/4/17 11:06 AM

[revised manuscript text omitted]

V2 Changes 2018/4/17 11:06 AM
Formatted ... [350]

V2 Changes 2018/4/17 11:06 AM

V2 Changes 2018/4/17 11:06 AM
Formatted ... [351]

V2 Changes 2018/4/17 11:06 AM

V2 Changes 2018/4/17 11:06 AM
Formatted ... [352]

V2 Changes 2018/4/17 11:06 AM

V2 Changes 2018/4/17 11:06 AM
Formatted ... [353]

V2 Changes 2018/4/17 11:06 AM

V2 Changes 2018/4/17 11:06 AM
Formatted ... [354]

V2 Changes 2018/4/17 11:06 AM

V2 Changes 2018/4/17 11:06 AM
Formatted ... [355]

V2 Changes 2018/4/17 11:06 AM

V2 Changes 2018/4/17 11:06 AM
Formatted ... [357]

V2 Changes 2018/4/17 11:06 AM

V2 Changes 2018/4/17 11:06 AM
Formatted ... [358]

V2 Changes 2018/4/17 11:06 AM

V2 Changes 2018/4/17 11:06 AM
Formatted ... [359]

V2 Changes 2018/4/17 11:06 AM

V2 Changes 2018/4/17 11:06 AM
Formatted ... [360]

V2 Changes 2018/4/17 11:06 AM

V2 Changes 2018/4/17 11:06 AM
Formatted ... [361]

V2 Changes 2018/4/17 11:06 AM

V2 Changes 2018/4/17 11:06 AM
Formatted ... [362]

V2 Changes 2018/4/17 11:06 AM

V2 Changes 2018/4/17 11:06 AM
Formatted ... [363]

V2 Changes 2018/4/17 11:06 AM

V2 Changes 2018/4/17 11:06 AM
Formatted ... [364]

V2 Changes 2018/4/17 11:06 AM

V2 Changes 2018/4/17 11:06 AM
Formatted ... [349]

[revised manuscript text omitted]